# SIMULATION-FREE STRUCTURE LEARNING FOR STOCHASTIC DYNAMICS

## ABSTRACT

Modeling dynamical systems and unraveling their underlying causal relationships is central to many domains in the natural sciences. Various physical systems, such as those arising in cell biology, are inherently high-dimensional and stochastic in nature, and admit only partial, noisy state measurements. This poses a significant challenge for addressing the problems of modeling the underlying dynamics and inferring the network structure of these systems. Existing methods are typically tailored either for structure learning or modeling dynamics at the population level, but are limited in their ability to address both problems together. In this work, we address both problems simultaneously: we present STRUCTUREFLOW, a novel and principled simulation-free approach for jointly learning the structure and stochastic population dynamics of physical systems. We showcase the utility of STRUCTUREFLOW for the tasks of structure learning from interventions and dynamical (trajectory) inference of conditional population dynamics. We empirically evaluate our approach on high-dimensional synthetic systems, a set of biologically plausible simulated systems, and an experimental single-cell dataset. We show that STRUCTUREFLOW can learn the structure of underlying systems while simultaneously modeling their conditional population dynamics — a key step toward the mechanistic understanding of systems behavior.

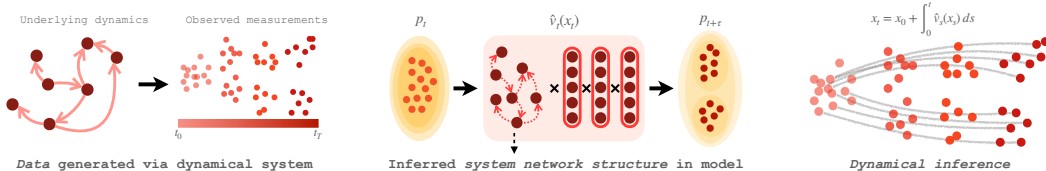

Figure 1: **Overview of STRUCTUREFLOW for joint structure learning and dynamical inference.**

## 1 INTRODUCTION

Unraveling the underlying structure of physical systems from their stochastic dynamics, given noisy and partial measurements, is a central problem in many areas of science. Many physical systems, notably in cell and molecular biology, operate in out-of-equilibrium regimes, are high-dimensional, and are subject to intrinsic stochasticity (Elowitz et al., 2002; Schiebinger et al., 2019). This poses a significant challenge for the task of deciphering the underlying system structure and modeling the resulting dynamics. Effectively addressing this problem is a crucial step toward gaining a mechanistic understanding of systems and the ability to predict their behavior under natural and perturbational conditions. This would, in turn, provide practitioners with a map for how to control and guide a system's response towards desirable states (Dixit et al., 2016; Norman et al., 2019). In cellular and molecular biology, such a tool would have significant implications in advancing our ability to interpret and model cell fate, development, response to disease (Rizvi et al., 2017; Binnewies et al., 2018; Gulati et al., 2020; Molè et al., 2021), predict perturbational responses in tissues and patients (Ji et al., 2021; Bunne et al., 2023; Peidli et al., 2024; Atanackovic et al., 2024), and facilitate experimental design (Zhang et al., 2023; Huang et al., 2024).

Currently, there exist two broad classes of approaches for deciphering the stochastic dynamics of physical systems from *data*. The first class, which we refer to as **dynamical inference**[1], commonly

---

[1]This task is also commonly labeled and referred to as *trajectory inference* (Hashimoto et al., 2016; Weinreb et al., 2018; Schiebinger et al., 2019; Tong et al., 2020; Neklyudov et al., 2022). Since the concept of "trajectory"

deals with constructing an estimate of the underlying vector field from partial and noisy measurements. From a good estimate of the vector fields, qualitative properties of dynamics can be interpreted, valuable for understanding cell dynamics and cell fate (Qiu et al., 2022; MacArthur, 2022). In this work, we focus on the preceding task of learning good estimates of vector fields from partially observed population snapshot data.

The second (and in our work, concurrent) class, **structure learning**[2], aims to reconstruct the *directional* relationships between each of the variables of a given system. Existing methods are typically designed to tackle either task in isolation and are limited in their ability to address both problems together (Zheng et al., 2018; Huynh-Thu and Geurts, 2018; Gao et al., 2018; Brouillard et al., 2020; Atanackovic et al., 2023; Zhang, 2024). We posit that knowledge of the underlying vector field is advantageous for this task, and thus we address both problems simultaneously under a single-model.

To achieve this, we propose STRUCTUREFLOW, a principled and *simulation-free* method for joint dynamical inference and structure learning. Building on recent advances in score and flow matching ([SF]²M) (Tong et al., 2024) and entropy-regularized optimal transport (EOT) (Cuturi, 2013; Shi et al., 2023), STRUCTUREFLOW learns a probability flow ordinary differential equation (PF-ODE) from snapshot data. This allows it to model the continuous evolution of a system's population while simultaneously inferring the underlying network structure embedded directly within the model's parameterization.

The simulation-free framing of STRUCTUREFLOW is critical for high-dimensional systems common in cellular biology, as it avoids computationally expensive trajectory simulations during training. To best facilitate the joint learning task, we introduce a novel parameterization: an autonomous (time-independent) vector field represented by a Neural Graphical Model (NGM) (Bellot et al., 2022; Dinh and Ho, 2020) which captures the stationary system structure, while a time-dependent score function captures the evolving stochastic dynamics. We present an overview of our framework in Figure 1 and outline our core contributions below:

- We formulate joint dynamical inference and structure learning as a multi-timepoint Schrödinger Bridge (SB) problem.
- We introduce STRUCTUREFLOW, a novel simulation-free approach tailored for simultaneously learning the underlying network structure and conditional population dynamics of a stochastic dynamical system from noisy and partial observations.
- We construct a comprehensive empirical evaluation for the joint inference task over an assortment of systems: (i) on high-dimensional synthetic systems, (ii) an collection of biologically plausible simulated systems, and (iii) an experimental single cell dataset with genetic interventions. We use our evaluation pipeline to showcase the application of STRUCTUREFLOW for the joint structure learning and dynamical inference tasks, all while building an extensive benchmark of state-of-the-art methods.

## 2 BACKGROUND AND PRELIMINARIES

### 2.1 PROBLEM: MODELING STOCHASTIC POPULATION DYNAMICS

We consider a dynamical model in $\mathbb{R}^d$ described by the stochastic differential equation (SDE):

$$\mathrm{d}\boldsymbol{x}_t = \boldsymbol{v}_t(\boldsymbol{x}_t)\,\mathrm{d}t + \boldsymbol{\sigma}\,\mathrm{d}\boldsymbol{B}_t, \tag{1}$$

where $\boldsymbol{x}_t \in \mathbb{R}^d$ is the state at time $t$, $\boldsymbol{v}_t : [0,1] \times \mathbb{R}^d \to \mathbb{R}^d$ is the drift, $\mathrm{d}\boldsymbol{B}_t$ are Brownian motion increments, and $\boldsymbol{\sigma} \in \mathbb{R}^{d \times d}$ is the diffusion coefficient matrix. For simplicity, we shall assume that $\boldsymbol{\sigma} = \sigma\mathbf{I}$ for some $\sigma > 0$, although the concepts we introduce can all be generalized to the setting of anisotropic noise.

Given $\boldsymbol{x}_0 \sim p_0$, where $p_0$ is a density over $\mathbb{R}^d$, the dynamics of equation 1 gives rise to a family of *marginals* $(p_t)_{t \in [0,1]}$, where $p_t$ denotes the marginal probability distribution of the random variable $\boldsymbol{x}_t$ at time $t$, which are characterized by the accompanying *Fokker-Planck equation* (FPE):

$$\partial_t p_t = -\nabla \cdot (p_t(\boldsymbol{x}_t)\boldsymbol{v}_t(\boldsymbol{x}_t)) + \tfrac{1}{2}\nabla \cdot (\sigma^2 \nabla p_t(\boldsymbol{x}_t)). \tag{2}$$

The solution to the FPE at time $t$, $p_t(\boldsymbol{x})$, gives the probability density of the population at time $t$, starting from $p_0(\boldsymbol{x})$. Importantly, equation 2 can be reformulated as an equivalent *probability flow* ordinary differential equation (ODE) (Song et al., 2021; Maoutsa et al., 2020):

$$\partial_t p_t(\boldsymbol{x}) = -\nabla \cdot (p_t(\boldsymbol{x})\boldsymbol{u}_t(\boldsymbol{x})), \qquad \boldsymbol{u}_t(\boldsymbol{x}) = \boldsymbol{v}_t(\boldsymbol{x}) - \tfrac{1}{2}\sigma^2 \nabla_{\boldsymbol{x}} \log p_t(\boldsymbol{x}). \tag{3}$$

---

has been frequently used in the literature to refer to several non-equivalent things, in this work, we prefer to name this task *dynamical inference*.

[2]Also commonly referred to in literature as *network inference*, *causal discovery*, and *system identification*.

In the above, $\boldsymbol{u}_t$ is the *probability flow field* and is related to the drift of the SDE equation 1 up to the addition of a term involving the score, $\boldsymbol{s}_t(\boldsymbol{x}) = \nabla_{\boldsymbol{x}} \log p_t(\boldsymbol{x})$. As will be apparent in Section 2.2, the probability flow formulation of the FPE equation 2, and hence the SDE equation 1, opens up an avenue for addressing dynamics inference without the need for costly simulation-based methods.

**Inference problem setup.** We consider a setting where empirical snapshot observations are available at multiple timepoints $t_0, \ldots, t_{T-1}$ under different conditions $c \in \mathcal{C}$, yielding a set of empirical marginal distributions $\{\hat{p}_{t_i}^{(c)}\}_0^{T-1}$ over $\boldsymbol{x}_{t_i} \in \mathbb{R}^d$ for each condition $c$. Each marginal comprises $N_i$ i.i.d. samples assumed to arise from some unknown SDE of the form equation 1. We assume the noise level $\boldsymbol{\sigma}$ is known, while the vector field $\boldsymbol{v}_t$ is unknown[3]. From here, our objective is twofold: (**1;** *dynamical inference*) approximate the (*conditional*) vector field $\boldsymbol{v}_t$ of the underlying stochastic dynamics, and (**2;** *structure learning*) recover a directed graph $\boldsymbol{A} \in \mathbb{R}^{d \times d}$ that captures dependencies among $d$ variables and represents the underlying data generative process of the system. We formulate this joint inference task through the lens of Schrödinger Bridges.

We highlight orthogonal fields of work, such as Kipf et al. (2018) and Frishman and Ronceray (2020) which jointly learn interactions and dynamics, but require fully observed trajectories, making them inapplicable to single cell time-series experiments (which produce population snapshots only). Further, neural relational inference (Kipf et al., 2018) assumes deterministic trajectories, whereas we consider the setting of trajectories adhering from stochastic dynamics.

## 2.2 Simulation-Free Schrödinger Bridges via Score and Flow Matching

The Schrödinger Bridge Problem (SBP) is concerned with finding the most likely stochastic evolution transporting a source density $q_0$ to a target density $q_1$, given a reference process that encodes prior knowledge of the dynamics. In its dynamical form, the SBP is typically formulated in terms of *laws of stochastic processes*, i.e. probability measures on the path space $C([0,1], \mathbb{R}^d)$ that describe the distribution of entire sample trajectories. Writing $\mathbb{P}$ to be the law of a process transporting $q_0$ to $q_1$ and $\mathbb{Q}$ to be the reference measure, we seek a stochastic process $\mathbb{P}^*$ satisfying:

$$\mathbb{P}^* = \underset{\mathbb{P}:p_0=q_0,p_1=q_1}{\arg\min} \ \text{KL}(\mathbb{P}\|\mathbb{Q}) \tag{4}$$

for marginals $p_t$ of $\mathbb{P}$, and $\text{KL}(\mathbb{P}\|\mathbb{Q}) = \int d\mathbb{P} \log(d\mathbb{P}/d\mathbb{Q})$ is the Kullback-Leibler divergence. When the reference process $\mathbb{Q}$ is the Brownian motion, a key result from Schrödinger Bridge theory (Föllmer, 1988) is that the solution of equation 4 takes the form of a mixture of Brownian bridges $\mathbb{Q}_{xy}$ with respect to a *coupling* $\pi$ of the distributions $(q_0, q_1)$:

$$\mathbb{P}^\star = \int \mathbb{Q}_{xy} \, d\pi(x, y). \tag{5}$$

Furthermore, the SBP coupling $\pi$ amounts to the solution of an entropic optimal transport problem (Léonard, 2014):

$$\pi^\star = \arg\min_{\pi \in \Pi(q_0,q_1)} \tfrac{1}{2}\mathbb{E}_\pi\|x - y\|_2^2 + \varepsilon\text{KL}(\pi|q_0 \otimes q_1) \tag{6}$$

with regularization parameter $\varepsilon = \sigma^2 > 0$. In practice when $(q_0, q_1)$ are discrete distributions, this can be solved extremely efficiently using the Sinkhorn algorithm (Cuturi, 2013). While the coupling $\pi$ is easy to compute from samples, finding a solution to the dynamical SBP in continuous time is desirable for modelling continuous dynamics. With a notable exception being the case of Gaussian measures (Bunne et al., 2022) where analytical expressions are available, the dynamical SBP equation 4 does not admit a straightforward solution for $\boldsymbol{v}_{\text{SB}}(t, \boldsymbol{x})$, defined as the drift of the SDE, $d\boldsymbol{x}_t = \boldsymbol{v}_{\text{SB}}(t, \boldsymbol{x}_t)\,dt + \sigma\,d\boldsymbol{B}_t$, which underlies the Schrödinger Bridge process $\mathbb{P}$.

Numerous works aim to build approximations to $\boldsymbol{v}_{\text{SB}}$ and hence solutions to the dynamical SBP (Chen et al., 2022; Bortoli et al., 2021; Shi et al., 2023; Tong et al., 2024). In particular, Tong et al. (2024) proposes to use score matching (Hyvärinen, 2005) and flow matching (Lipman et al., 2023a) ([SF]$^2$M) as a natural methodology for building the approximation to $\boldsymbol{v}_{\text{SB}}$ in order to solve for $\mathbb{P}$. Note $\boldsymbol{v}_{\text{SB}}(t, \boldsymbol{x})$ simply corresponds to the drift $\boldsymbol{v}_t$ defined by the SBP. The key observation is to leverage the reciprocal process characterization of $\mathbb{P}$ (Léonard, 2014) as a mixture of Brownian bridges, and the fact that Brownian bridges conditioned on endpoints $(\boldsymbol{x}_0, \boldsymbol{x}_1) =: \boldsymbol{z}$ admit closed form expressions for their

---

[3]Throughout we *assume* this data-generation model holds for every condition $c$.

probability flow $\boldsymbol{v}_t^\circ(\boldsymbol{x}|\boldsymbol{z})$ and score $\nabla \log p_t(\boldsymbol{x}|\boldsymbol{z})$:

$$\boldsymbol{v}_t^\circ(\boldsymbol{x}|\boldsymbol{z}) = \frac{1-2t}{t(1-t)}(\boldsymbol{x}-(t\boldsymbol{x}_1+(1-t)\boldsymbol{x}_0))+(\boldsymbol{x}_1-\boldsymbol{x}_0); \quad \nabla \log p_t(\boldsymbol{x}|\boldsymbol{z}) = \frac{t\boldsymbol{x}_1+(1-t)\boldsymbol{x}_0-\boldsymbol{x}}{\sigma^2 t(1-t)}. \tag{7}$$

Together, these provide the probability flow characterisation of the Brownian bridge. With this in hand, and leveraging the fact that the dynamical SBP can be constructed as a mixture of Brownian bridges (equation 5), Tong et al. (2024) propose to construct a neural approximation to the SB flow field $\boldsymbol{v}^\theta(t, \boldsymbol{x}) \approx \boldsymbol{v}(t, \boldsymbol{x}) := \boldsymbol{v}_t^\circ(t, \boldsymbol{x}) - \frac{\sigma^2}{2}\nabla \log p_t(\boldsymbol{x})$. The flow field $\boldsymbol{v}^\theta(t, \boldsymbol{x})$ is trained together with a score field $\boldsymbol{s}^\theta(t, \boldsymbol{x})$ using the conditional score and flow matching loss

$$\mathcal{L}_{[\mathrm{SF}]^2\mathrm{M}}(\theta) = \mathbb{E}_{t,\boldsymbol{z},\boldsymbol{x}}\left[\|\boldsymbol{v}_t^\theta(\boldsymbol{x}) - \boldsymbol{v}_t^\circ(\boldsymbol{x}|\boldsymbol{z})\|^2 + \lambda(t)\|\boldsymbol{s}_t^\theta(\boldsymbol{x}) - \nabla \log p_t(\boldsymbol{x}|\boldsymbol{z})\|^2\right]. \tag{8}$$

In the above, $\lambda(t) > 0$ weighs the score as a function of time $t$, and $\boldsymbol{z} := (\boldsymbol{x}_0, \boldsymbol{x}_1) \sim \pi(\boldsymbol{x}_0, \boldsymbol{x}_1)$ where $\pi$ is the entropic OT coupling in equation 5 obtained via the Sinkhorn algorithm. Following Proposition 3.4 of Tong et al. (2024), minimising $\mathcal{L}_{[\mathrm{SF}]^2\mathrm{M}}(\theta)$ then approximates the solution to the SBP in equation 4 under mild conditions.

In the above, $\lambda(t) > 0$ weighs the score as a function of time $t$, and $\boldsymbol{z} := (\boldsymbol{x}_0, \boldsymbol{x}_1) \sim \pi(\boldsymbol{x}_0, \boldsymbol{x}_1)$ where $\pi$ is the entropic OT coupling in equation 5 obtained via the Sinkhorn algorithm. Following Proposition 3.4 of Tong et al. (2024), minimising $\mathcal{L}_{[\mathrm{SF}]^2\mathrm{M}}(\theta)$ then approximates the solution to the SBP in equation 4 under mild conditions. Additionally, Chen et al. (2023); Theodoropoulos et al. (2025); Rohbeck et al. (2025) model a multi-marginal SBP, where successive time-marginal couplings are not computed independently, but rather globally across all timepoints.

## 3 STRUCTUREFLOW: JOINT INFERENCE OF STRUCTURE AND DYNAMICS

Our objective is to jointly infer the network structure and population dynamics of the underlying stochastic system. STRUCTUREFLOW achieves this in a novel simulation-free manner. In this section we introduce the central components of our framework, namely: **(i)** improved parameterization tailored for the joint task, **(ii)** modeling conditional stochastic population dynamics from interventional data, and **(iii)** simulation-free training for learning conditional stochastic population dynamics.

### 3.1 STRUCTURAL VECTOR FIELD PARAMETERIZATION

We consider a time-independent (or *autonomous*) vector field, which models the underlying structure of a system, and a time-dependent score function accounting for stochasticity. We assume that the underlying system structures (graphs) are stationary between adjacent marginals $(\hat{p}_{t_i}^{(c)}, \hat{p}_{t_{i+1}}^{(c)})$. We posit that this assumption yields an easier *joint* inference problem, allowing us to parameterize *a single* structure within the autonomous vector field.

**Autonomous vector field and time-dependent score parameterization.** From equation 1 we seek to recover the autonomous vector field $\boldsymbol{v}$, while the objective in (Tong et al., 2024) targets the SB flow $\boldsymbol{v}_t^\circ$. The quantities of the ground truth process equation 1 are related via $\boldsymbol{v}_t^\circ(\boldsymbol{x}_t) = \boldsymbol{v}(\boldsymbol{x}_t) - \frac{\sigma^2}{2}\nabla_{\boldsymbol{x}_t}\log p_t(\boldsymbol{x}_t)$, where $\boldsymbol{v}_t^\circ, \nabla \log p_t$ are the probability flow field and the score of the Brownian bridge pinned at $(t_i, x_i), (t_{i+1}, x_{i+1})$ respectively. We therefore make the parametrization choice $\hat{\boldsymbol{v}}_t(\boldsymbol{x}_t) = \boldsymbol{v}^\theta(\boldsymbol{x}_t) - \frac{\sigma^2}{2}\boldsymbol{s}_t^\phi(\boldsymbol{x}_t)$. Since time-independent vector fields, $\boldsymbol{v}$, are a function of state, and not spurious temporal correlations, they are better suited for modeling underlying structures which are stationary over time, as we assume. This leads to a more principled modeling approach.

**Modeling dynamic structural dependencies via neural graphical vector fields.** We consider a neural structural model following the definition of Bellot et al. (2022), where we assume there exists functions $\boldsymbol{v}_1, \ldots, \boldsymbol{v}_d$ such that for $j = 1, \ldots, d$, and $\boldsymbol{v}_j : \mathcal{X}^d \to \mathbb{R}$, where $\mathcal{X}^d$ is a bounded subset of $\mathbb{R}^d$. We can rewrite equation 1 under the neural dynamic structural model formalism as

$$\mathrm{d}x_j(t) = \boldsymbol{v}_j(\boldsymbol{x}(t))\,\mathrm{d}t + \boldsymbol{\sigma}_j\,\mathrm{d}B_j(t), \quad \boldsymbol{x}(0) \sim p_0, \tag{9}$$

where $\mathrm{d}x_j(t)$ denotes the structural dependency for the instantaneous rate of change of the $j^{\mathrm{th}}$ variable dependent on all state observations $\boldsymbol{x}$ at time $t$.[4] We use an NGM (Bellot et al., 2022) to parameterize the dynamic structural relationships defined in equation 9.

NGMs are a class of neural graphical model parameterizations designed to represent complex variable (features) dependencies in a computationally efficient manner (Shrivastava and Chajewska, 2023).

---

[4]We note that we are using $\boldsymbol{x}(t)$ interchangeably with $\boldsymbol{x}_t$.

Bellot et al. (2022) extend this formulation for the structure learning of continuous-time dynamical systems, motivating their application in our framework. The NGM lets us parameterize the structural relationships of a system with $d$ variables directly within the *autonomous* vector field $\boldsymbol{v}(\boldsymbol{x}_t)$. We use an NGM to approximate the *autonomous* component of the ground truth process described in equation 1 ($\boldsymbol{v}(\boldsymbol{x}_t)$ instead of $\boldsymbol{v}_t(\boldsymbol{x}_t)$). The NGM parameterization of the *autonomous* vector field is defined as:

$$\boldsymbol{v}^{\theta_j}(\boldsymbol{x}_t) = \psi(\cdots \psi(\psi(\boldsymbol{x}_t \theta_j^A)\theta_j^1)\cdots)\theta_j^K, \quad j = 1,\ldots,d, \tag{10}$$

where $\theta_j^A \in \mathbb{R}^{d \times h}$ denotes a weight matrix (or graph layer) representing the dynamic dependencies between $\mathrm{d}x_j(t)$ and $\boldsymbol{x}(t)$, $\theta_j^k \in \mathbb{R}^{h \times h}$ for $k = 1,\ldots,K-1$ are the weight matrices of each corresponding hidden layer, $\theta_j^K \in \mathbb{R}^{h \times 1}$ is the final layer's weight matrix yielding a scalar output, and $\psi(\cdot)$ is an activation function. We assume variable dependencies defined by $\theta_j^A$ are sparse and enforce sparsity on $\theta_j^A$ using Group Lasso regularization during training. We include further details regarding the NGM and its optimization for dynamic structure learning in Appendix A.1, as well as principled theoretical justification for recovering the underling network structure in Appendix A.3.

### 3.2 Modeling Conditional Population Dynamics

STRUCTUREFLOW explicitly learns conditional stochastic dynamics (trajectories) using both interventional and observational data and simultaneously infers the underlying system structure. We model interventions as ideal *knockouts*, i.e. where for a given intervention, a variable is entirely removed from the system, and the intervention does not directly affect other variables of the system. For an intervention $c$, we define a binary mask $\boldsymbol{M}^{(c)} \in \{0,1\}^{d \times d}$ as:

$$M_{ji}^{(c)} = \begin{cases} 0 \text{ if } i = c, j \neq c \\ 1 \text{ otherwise} \end{cases} \tag{11}$$

Where $i$ is the possible source of influence and $j$ is the target. The mask is structured to represent the severed outgoing influences of the variable c. In the observational setting, no mask is applied. Under condition $c$, the vector field's parameters become $\theta_j^{A(c)}$, obtained by an element-wise product with $\boldsymbol{M}^{(c)}$: $\theta_j^{A(c)} = \boldsymbol{M}^{(c)} \odot \theta_j^A$. The drift is then evaluated as:

$$\boldsymbol{v}_c^{\theta_j}(\boldsymbol{x}_t|c) = \psi(\cdots \psi(\psi(\boldsymbol{x}_t \theta_j^{A(c)})\theta_j^1)\cdots)\theta_j^K. \tag{12}$$

We can then recover our underlying matrix estimate $\boldsymbol{A} \in \mathbb{R}^{d \times d}$ as $A_{ji} = \|(\theta_j^A)_{i,:}\|_2$. Similarly, we condition the score $\boldsymbol{s}_t^\phi(\boldsymbol{x}_t)$ on c, i.e., $\nabla_{\boldsymbol{x}_t} \log p_t(\boldsymbol{x}_t|\boldsymbol{k}^{(c)})$ where $\boldsymbol{k}^{(c)} \in \{0,1\}^d$ is a conditional input vector with $k_i^{(c)} = 1$ if $i$ is perturbed under $c$, otherwise $k_i^{(c)} = 0$.

### 3.3 Simulation-free end-to-end Training

STRUCTUREFLOW builds on the $[\text{SF}]^2\text{M}$ framework introduced by Tong et al. (2024) to learn a neural approximation of $\boldsymbol{v}$ without simulation. We use Entropic Optimal Transport (EOT) to pair points drawn from temporally adjacent snapshots $\hat{p}_{t_i}^{(c)}$ and $\hat{p}_{t_{i+1}}^{(c)}$. We use the Sinkhorn algorithm (Cuturi, 2013) to estimate the probabilistic EOT couplings between pairs of distributions only a single time before training starts, and do not need to be recomputed. From each coupling, we draw paired samples $(x_0, x_1)$ across $[t_i, t_{i+1}]$ and compute the corresponding target drift of the probability flow $\boldsymbol{v}_t^\circ$ and target score $\boldsymbol{s}_t = \nabla_{\boldsymbol{x}} \log p_t(\boldsymbol{x} \mid c)$. We parametrize the drift of the probability flow as:

$$\hat{\boldsymbol{v}}_{t,c}(\boldsymbol{x}_t; \Theta|\boldsymbol{M}^{(c)}, \boldsymbol{k}^{(c)}) = \boldsymbol{v}_c^\theta(\boldsymbol{x}_t|\boldsymbol{M}^{(c)}) - \frac{\sigma^2}{2} \boldsymbol{s}_t^\phi\left(\boldsymbol{x}_t|\boldsymbol{k}^{(c)}\right). \tag{13}$$

for conditional probability flow parametrization $\hat{\boldsymbol{v}}_{t,c}$ under condition $c$. We optimize model parameters $\Theta = (\theta, \phi)$ by regressing to the targets $\boldsymbol{v}_t^\circ$ and $\boldsymbol{s}_t$ via the STRUCTUREFLOW loss

$$\mathcal{L}_{\text{SF}}(\Theta) = \sum_c \mathbb{E}\left[(1-\alpha)\,||\hat{\boldsymbol{v}}_{t,c}(\boldsymbol{x}_t; \Theta|\boldsymbol{M}^{(c)}, \boldsymbol{k}^{(c)})t - \boldsymbol{v}_t^\circ||^2 + \alpha||\boldsymbol{s}_{t,c}^\phi(\boldsymbol{x}_t|\boldsymbol{k}^{(c)}) - \boldsymbol{s}_t||^2\right], \tag{14}$$

where the expectation is over $t \sim \mathcal{U}(0,1)$ and $\boldsymbol{x}_t \sim p_t(\boldsymbol{x})$, and $0 \leq \alpha \leq 1$ weighs the vector field loss and score loss. During training, we apply Group Lasso on $\theta_j^A$ to encourage the model to learn sparse dependencies. We include detailed pseudo code for training STRUCTUREFLOW in Algorithm 1.

We note that we choose to solve the EOT problem to pair successive time marginals ($\hat{p}_{t_i}^{(c)}$ and $\hat{p}_{t_{i+1}}^{(c)}$) independently of the other marginals $\hat{p}_t \mid t \notin \{i, i+1\}$, not globally across all time marginals like is done in Chen et al. (2023); Theodoropoulos et al. (2025); Rohbeck et al. (2025). STRUCTUREFLOW is flexible however, in that one would simply swap the coupling algorithm to solve the global multi-

marginal problem instead with no additional changes to training. This could be used for other means as well, such as incorporating biological priors.

## 4 RELATED WORK

**Structure learning for continuous dynamics.** There exist several works for structure learning of continuous dynamics. Notably, Bellot et al. (2022) introduce a continuous-time framework for inferring structure of the underlying dynamical system from time-series data. Wang et al. (2023) extend this approach to the stochastic and Bayesian setting. However, these approaches rely in computationally expensive neural ODE solvers during training, Recent lines of work have introduced structure learning methods for continuous dynamics specifically tailored for biological systems and the inference of gene regulatory networks (GRNs) (Huynh-Thu et al., 2010; Huynh-Thu and Geurts, 2018; Gao et al., 2018; Tokumasu et al., 2023). However, all these methods assume a selection of fixed ODE parameterizations, linear dynamics, or perturbation-specific recovery, limiting their ability to model stochastic trajectories and complex network interactions. Symbolic regression-based approaches off a complimentary line of work (Brunton et al., 2016; Sun et al., 2023; Dakhmouche et al., 2025). Dakhmouche et al. (2025) also solve for continuous population dynamics (dynamical/trajectory inference) and provide an interpretable model via the learned symbolic equations, but do not explicitly recover a network structure. Brunton et al. (2016); Sun et al. (2023) also use symbolic regression to recover closed-form equations, but assume fully observed trajectories.

**Joint inference of structure and dynamics.** While both dynamical inference and structure learning are central to understanding stochastic systems, many existing methods treat these tasks in a decoupled fashion (Qiu et al., 2022; Sha et al., 2024). These procedures are centered around learning a dense vector field from the observed data, subsequently extracting structural information post-hoc. Moreover, they do not incorporate any prior on sparsity or explicitly model structure during training. Tong et al. (2024) provide a preliminary investigation into the use of the NGM architecture in conjunction with $[\text{SF}]^2\text{M}$ for jointly inferring structure and dynamics, but do so in a limited setting, i.e. don't consider interventions and assume access to a significant quantity of time-points. We demonstrate later (Section 5 Figure 3) that the trivial application of the NGM architecture under the $[\text{SF}]^2\text{M}$ setup does not achieve competitive performance. Lin et al. (2025) propose a neural ODE-based approach for the joint inference task, but do not model stochastic dynamics, only consider two time-points, and require simulation during training (akin to Bellot et al. (2022)). STRUCTUREFLOW addresses these limitations for the joint task.

## 5 EXPERIMENTS

We evaluate STRUCTUREFLOW on both synthetic and real-world datasets to assess its ability to recover underlying network structure and infer stochastic population dynamics. Our experiments focus on three key capabilities: structure learning of dynamical systems from population data, dynamical inference across left-out time-points, and prediction of responses to unseen interventions (knockouts). We also include a scaling study to highlight our method's computational efficiency. We provide further details for these experiments in Appendix A, B, and E.

### 5.1 EXPERIMENTS ON SYNTHETIC HIGH-DIMENSIONAL SYSTEMS

We evaluate STRUCTUREFLOW for the structure learning task across varying system dimensionality using randomly generated Erdos-Renyi graphs (Erdős and Rényi, 1959) and derive synthetic data via linear SDE simulation (see Appendix A.6 for details). We vary the dimensionality of the system $d$ from 10 to 500 variables

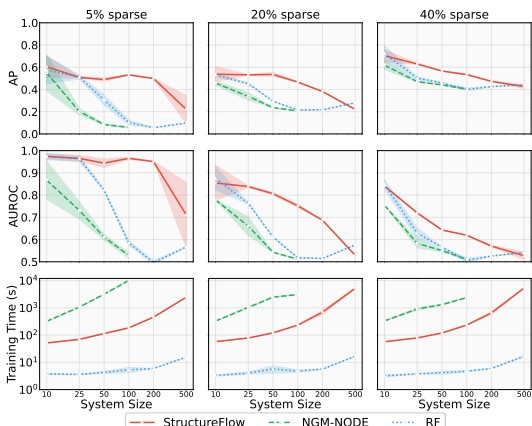

Figure 2: **STRUCTUREFLOW yields improved structure learning performance when scaled to high-dimensional systems.** We compare with NGM-NODE and RF on synthetic linear systems with varying dimensionality ($d$) and system (graph) sparsity levels (5%, 20%, 40%).

and the sparsity (proportion of non-zero edges) from 5% to 40% of the underlying graph which defines the data-generative process of the dynamical system. We consider a simulation-based approach, NGM-NeuralODE (NGM-NODE) (Bellot et al., 2022), and Reference Fitting (RF) (Zhang, 2024),

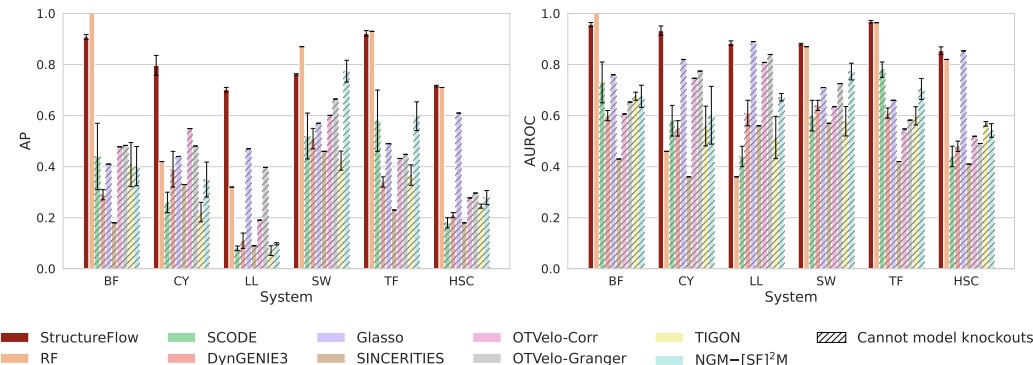

Figure 3: **STRUCTUREFLOW is consistently a top performing *structure learning* method across simulated biological systems.** Here, we use interventional (with *knockouts*) and observational (no *knockouts*) data, and report average precision (AP) and area under the ROC curve (AUROC) scores.

Table 1: **STRUCTUREFLOW outperforms baseline methods for *dynamical inference* on simulated biological systems.** Shown is a comparison of dynamical inference methods for learning conditional population dynamics across synthetic biological systems. We include $[\text{SF}]^2\text{M}$ for comparison, but note that $[\text{SF}]^2\text{M}$ does not infer underlying network structure (i.e. cannot address joint task). Colors indicate **1ˢᵗ best performer** and **2ⁿᵈ best performer** for models that perform the joint inference task.

| | TF | | CY | | LL | | HSC (Curated) | | BF | | SW | |
|---|---|---|---|---|---|---|---|---|---|---|---|---|
| | $W_2\downarrow$ | MMD$\downarrow$ | $W_2\downarrow$ | MMD$\downarrow$ | $W_2\downarrow$ | MMD$\downarrow$ | $W_2\downarrow$ | MMD$\downarrow$ | $W_2\downarrow$ | MMD$\downarrow$ | $W_2\downarrow$ | MMD$\downarrow$ |
| $[\text{SF}]^2\text{M}$ | 0.761 ± 0.014 | 0.134 ± 0.005 | 0.517 ± 0.015 | 0.153 ± 0.002 | 0.847 ± 0.053 | 0.190 ± 0.010 | 0.664 ± 0.011 | 0.083 ± 0.003 | 0.660 ± 0.007 | 0.140 ± 0.004 | 0.572 ± 0.008 | 0.215 ± 0.005 |
| RF | 1.376 ± 0.000 | 0.191 ± 0.000 | 2.086 ± 0.000 | 0.326 ± 0.000 | 2.120 ± 0.007 | 0.296 ± 0.000 | 0.950 ± 0.000 | 0.066 ± 0.000 | 1.495 ± 0.000 | 0.242 ± 0.000 | 1.371 ± 0.000 | 0.290 ± 0.000 |
| OTVelo | 1.874 ± 0.262 | 0.582 ± 0.039 | 1.972 ± 0.144 | 0.622 ± 0.034 | 2.878 ± 0.361 | 0.647 ± 0.015 | 1.942 ± 0.006 | 0.603 ± 0.002 | 1.847 ± 0.277 | 0.591 ± 0.042 | 1.940 ± 0.144 | 0.577 ± 0.022 |
| TIGON | 1.409 ± 0.294 | 0.063 ± 0.022 | 1.224 ± 0.053 | 0.044 ± 0.003 | 1.840 ± 0.293 | 0.066 ± 0.039 | 0.666 ± 0.363 | 0.012 ± 0.014 | 1.394 ± 0.294 | 0.066 ± 0.021 | 0.957 ± 0.190 | 0.017 ± 0.009 |
| STRUCTUREFLOW | 0.789 ± 0.012 | 0.135 ± 0.004 | 0.576 ± 0.012 | 0.140 ± 0.004 | 0.842 ± 0.031 | 0.196 ± 0.008 | 0.683 ± 0.011 | 0.068 ± 0.003 | 0.694 ± 0.017 | 0.142 ± 0.007 | 0.610 ± 0.010 | 0.220 ± 0.006 |

as baselines, since they are capable of the joint structure learning and trajectory inference task. We report results in Figure 2 showing area under the receiver operating characteristic curve (AUROC), average precision (AP), and training time (seconds). We observe that STRUCTUREFLOW exhibits favorable scaling performance as dimensionality of the system increases. While the baseline methods are competitive on small systems, we see that STRUCTUREFLOW consistently maintains strong performance in high-dimensional settings while exhibiting improved computational efficiency in terms of training time compared to NGM-NODE, especially for large $d$. We remark that RF is the most computationally efficient in this regard as it makes assumptions of linearity of the underlying system. Due to this, RF suffers from poor expressivity and exhibits poor performance on the second element of the joint task – dynamical (trajectory) inference. We show this in the following sections.

## 5.2 EXPERIMENTS ON SIMULATED BIOLOGICAL SYSTEMS

We consider six simulated biological systems: trifurcating (**TF**), cyclical (**CY**), long linear (**LL**), swirling (**SW**), bifurcating (**BF**), and a curated system mimics hematopoietic stem cell (**HSC**) differentiation. All datasets are simulated via BoolODE (Pratapa et al., 2020) under both observational and interventional (knockout) conditions across multiple time-points (see details in Appendix A). See Appendix E for extended synthetic system results and ablations. In this section, we empirically evaluate STRUCTUREFLOW for the joint structure learning and dynamics inference tasks.

**STRUCTUREFLOW effectively infers network structure of simulated biological systems.** To evaluate how well STRUCTUREFLOW recovers the underlying network structure, we extract the inferred graph from the first layer of the parameterized vector field (defined in equation 10) which interprets the learned weight matrix as a proxy for variable-variable (gene-gene) interactions. We use AP and AUROC to evaluate how closely the STRUCTUREFLOW inferred graphs recapitulate the ground truth structure used to simulate the respective synthetic biological systems.[5] We compare STRUCTUREFLOW to a variety of baseline methods (some of which are tailored for structure learning in biological systems): RF Zhang (2024), OTVelo Zhao et al. (2024), TIGON Sha et al. (2024), dynGENIE3 Huynh-Thu and Geurts (2018), SINCERITIES Gao et al. (2018), SCODE

---

[5]In this work, we do not threshold the learned graphs/structures. Instead, we use AP/AUROC metrics to evaluate structure learning performance across all possible thresholds. We leave extensions for learning thresholded structures for future work.

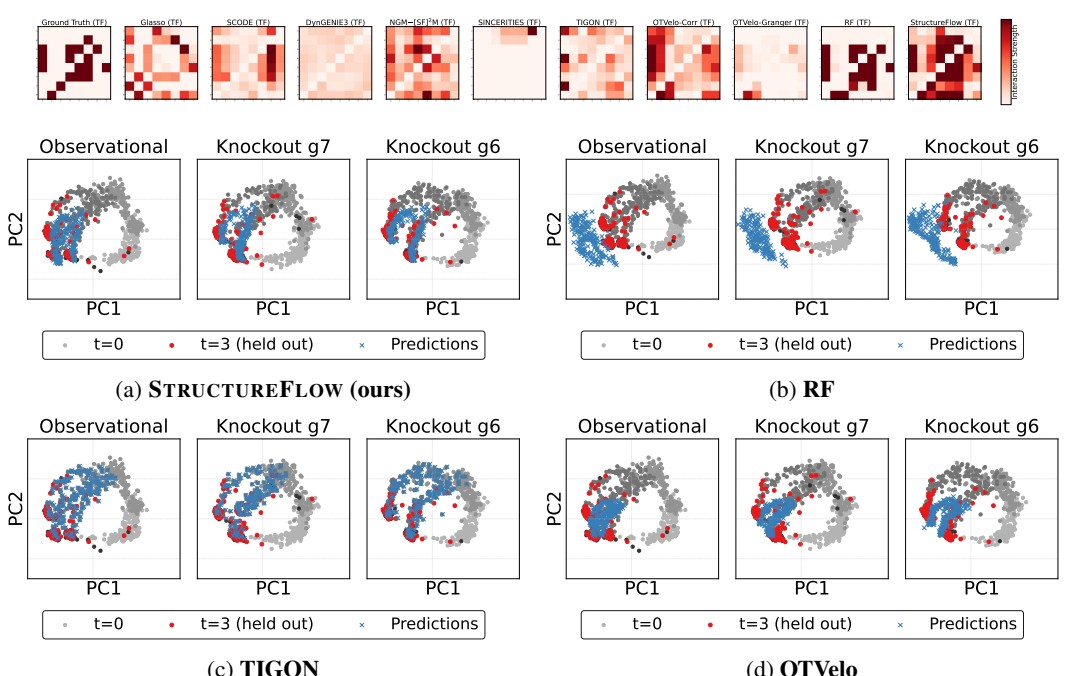

Figure 4: **Summary of inferred structure and visualization of model predicted dynamics.** (top) connectivity matrices (heatmaps) showing ground truth and inferred graphs for the **TF** (trifurcating) system. The x/y-axis labels correspond to system variables and shading indicates edge interaction strength. (bottom) visual comparison of dynamical inference using four different methods: STRUCTUREFLOW, RF, TIGON, and OTVelo, respectively. We show 2D plots of left-out timepoint prediction for the observational (wild-type) setting and for two *seen* interventions (knockouts) selected for their diverse trifurcating paths.

Matsumoto et al. (2017), graphical lasso (Glasso), and NGM-[SF]$^2$M Tong et al. (2024). All models are trained across five random seeds. We show the quantitative results for this experiment and evaluation in Figure 3 and visualizations of the inferred structures in Figure 4 (top). We observe that STRUCTUREFLOW exhibits higher consistency as a top performing method for inferring the underlying structure across all systems and both metrics. In comparison, certain baseline methods show favorable performance in specific systems and on one metric at a time, but struggle to consistently perform across the board.

**STRUCTUREFLOW effectively infers dynamics across left-out time-points in simulated biological systems.** To evaluate dynamical inference performance, we consider a leave-one-out cross-validation experiment across discrete timepoints, where we iteratively exclude each timepoint $t_k$ for $k \in \{1, \ldots, T-2\}$ during model training, then simulate via ODE from the ground truth data at $t_{k-1}$ to $t_k$. We use the Wasserstein-2 distance ($W_2$) and Maximum Mean Discrepancy (MMD) to evaluate the distributional distances between the predicted and ground truth distributions at $t_k$, evaluating STRUCTUREFLOW's ability to infer trajectory dynamics.

We consider RF, OTVelo, and TIGON as baselines since they are designed to address both inference task. We additionally include a comparison to [SF]$^2$M as a baseline method for dynamical inference, but we note [SF]$^2$M is not capable of jointly inferring the network structure. We show quantitative results for this experiment in Table 1 and visualization of the predictions in Figure 4a and Figure 4b. We observe that STRUCTUREFLOW outperforms both baselines RF and OTVelo, and a top performer relative to TIGON, while consistently out performing TIGON on the $W_2$. We remark that although TIGON yields strong performance on the dynamical inference task, it does not yield competitive results on the concurrent structure learning task (as shown in Figure 3).

**STRUCTUREFLOW infers structure and dynamics in the unbalanced setting.** We show a use case of STRUCTUREFLOW in the unbalanced setting. To extend

Table 2: **Unbalanced STRUCTUREFLOW**

|  | AUROC ↑ | AP ↑ | $W_2$ ↓ | MMD ↓ |
|---|---|---|---|---|
| TIGON | $0.540 \pm 0.014$ | $0.427 \pm 0.047$ | $2.355 \pm 0.591$ | $0.038 \pm 0.041$ |
| STRUCTUREFLOW | $0.735 \pm 0.005$ | $0.358 \pm 0.006$ | $1.672 \pm 0.532$ | $0.029 \pm 0.031$ |
| U-STRUCTUREFLOW | $\mathbf{0.838 \pm 0.010}$ | $\mathbf{0.509 \pm 0.025}$ | $\mathbf{1.670 \pm 0.438}$ | $\mathbf{0.027 \pm 0.033}$ |

STRUCTUREFLOW to the unbalanced setting (**U**-STRUCTUREFLOW), we replace the balanced Sinkhorn algorithm in our EOT procedure with the unbalanced Sinkhorn divergence and incorporate a growth rate field to model proliferation-induced density shifts. We evaluate our method on a growth-based bifurcation system. Following the setup of Zhang et al. (2025), we use the underlying gene regulatory network and boolean logic of the standard BoolODE bifurcation dataset, but we impose lineage specific proliferation rates where one of the attractors has a high proliferation rate and the other has a near-zero growth rate. We observe unbalanced-STRUCTUREFLOW yeilds the best performance for the joint task (Table 2). We discuss further details in Appendix B.3.

## 5.3 EXPERIMENTS ON REAL DATA (RENGE) SYSTEM

We consider an interventional time-series dataset of human induced pluripotent stem cells (iPSC) introduced by (Tokumasu et al., 2023). Interventions are conducted via clustered regularly interspaced short palindromic repeats (CRISPR) (knockouts) on 23 transcription factors (TFs). Cell states are then sequenced via single-cell RNA-seq across 4 time bins for each CRIPSR intervention. We process the dataset following the procedure of Zhang (2024) and select the 8 interventions with greatest observed change in population-level gene expression (see Appendix A.8).

**STRUCTUREFLOW recovers underlying structure of gene regulatory networks competitively to baselines.** We evaluate STRUCTUREFLOW on its ability to recover the $103 \times 103$ directed network of inferred TF-TF interactions from the processed real data. We remark that a central challenge in evaluating structure learning performance on real biological systems (datasets) is that there rarely exists a definitively known ground truth network. In this setting, we have access to an approximation of the ground truth network for 18 genes (TFs), which are determined using a CHIP-seq reference (see E.9 for details). Models

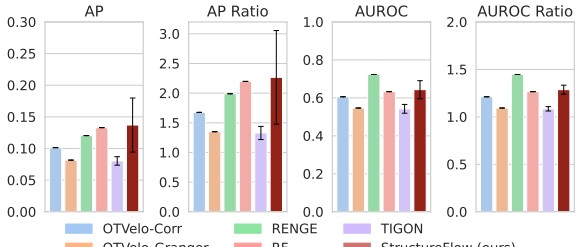

Figure 5: **STRUCTUREFLOW is capable of achieving better performance compared to baselines for the *structure learning* tasks in the real-data system.** We use AP/AUROC ratio denote structure recovery performance w.r.t. a random predictor.

are trained using all 103 genes, but evaluation in done over $18 \times 103$ *ground truth* networks. We show results in Figure 5 and observe that STRUCTUREFLOW yields competitive performance to counterpart baselines. While there is considerable variance across random seeds, the best-performing instances of STRUCTUREFLOW consistently outperform the top results from baseline methods. We include additional results in Appendix E.

**STRUCTUREFLOW effectively models cell trajectories and improves prediction of left-out interventions (knockouts) in real-data setting.** We report results for dynamical inference over left out time-points and left-out knockouts on the real biological system in Table 3 and Table 4, respectively. In the left-out intervention (knockout) setting, we withhold all time marginals $t_k, k = [0, ..., T-1]$ for a given interventional condition during training, then evaluate performance on predicting the final timepoint. Similar to the simulated biological systems, we use $W_2$ and MMD to evaluate performance. OTVelo and TIGON do not model interventions, and thus are not included in the left-out intervention evaluation. For left-out-time point dynamical inference (Table 3), we observe that STRUCTUREFLOW outperforms all baselines on $W_2$ and yields competitive results on MMD. We observe the STRUCTUREFLOW is the best performing joint method on the left-out intervention task. We remark that although STRUCTUREFLOW shows improved generalization performance on the left-out intervention task, implying some plausibly improved learning of underlying mechanisms, we make no claim that STRUCTUREFLOW truly models the *causal* dependencies of the underlying data generative process. We include an equivalent experiment for the simulated biological systems in Appendix E Table 7, but note these systems present an easier generalization setting relative to the real-data system. This is an interesting result that we leave to be explored further in future work.

## 6 CONCLUSION

In this work, we introduced STRUCTUREFLOW, a novel simulation-free method tailored for jointly learning the underlying network structure and the conditional population dynamics of high-dimensional stochastic systems. Our work highlights the benefits of addressing the joint inference task with a single model trained via score-and-flow matching and tailored for the problem — i.e. improved performance on both tasks. We demonstrate this through a suite of empirical experiments

Table 3: **STRUCTUREFLOW shows competitive performance for *dynamical inference* of left-out marginals on the real (Renge) biological dataset.** We show a comparison of dynamical inference methods for learning conditional population dynamics across left-out timepoints $k = 1, 2$ on the Renge dataset. We report results over withheld $k$ as well as the average of the two. Colors indicate **1ˢᵗ best performer** and **2ⁿᵈ best performer** for models that perform the joint inference task. Standard deviation in the last column is calculated over the averaged scores.

| | Timepoint 1 | | Timepoint 2 | | Average | |
|---|---|---|---|---|---|---|
| | $W_2\downarrow$ | MMD$\downarrow$ | $W_2\downarrow$ | MMD$\downarrow$ | $W_2\downarrow$ | MMD$\downarrow$ |
| [SF]²M | 5.673 ± 0.034 | 0.023 ± 6.9e-5 | 5.841 ± 0.030 | 0.015 ± 1.0e-4 | 5.757 ± 0.032 | 0.019 ± 8.6e-5 |
| RF | 10.187 ± 0.003 | 0.273 ± 2.6e-4 | 9.248 ± 0.008 | 0.221 ± 5.9e-4 | 9.718 ± 0.005 | 0.247 ± 4.2e-4 |
| OTVelo | 10.959 ± 0.000 | 0.711 ± 0.000 | 10.844 ± 0.000 | 0.706 ± 0.000 | 10.902 ± 0.063 | 0.708 ± 3.3e-3 |
| TIGON | 6.434 ± 0.041 | 0.004 ± 1.1e-3 | 6.401 ± 0.062 | 0.003 ± 9.2e-4 | 6.417 ± 0.016 | 0.003 ± 1.2e-3 |
| STRUCTUREFLOW | 5.589 ± 0.026 | 0.024 ± 4.6e-4 | 5.679 ± 0.029 | 0.017 ± 1.9e-4 | 5.634 ± 0.027 | 0.020 ± 3.3e-4 |

Table 4: **STRUCTUREFLOW shows competitive performance for *dynamical inference* of left-out interventions on the real (Renge) biological dataset.** We show a comparison of dynamical inference methods for learning conditional population dynamics on the Renge dataset for left-out interventions. We report results for 3 left-out gene knockout conditions, as well as the average performance across conditions. **Bold** indicates the best performing method that can perform the joint task.

| | NANOG | | POU5F1 | | PRDM14 | | Average | |
|---|---|---|---|---|---|---|---|---|
| | $W_2\downarrow$ | MMD$\downarrow$ | $W_2\downarrow$ | MMD$\downarrow$ | $W_2\downarrow$ | MMD$\downarrow$ | $W_2\downarrow$ | MMD$\downarrow$ |
| [SF]²M | 5.726 ± 0.015 | 0.011 ± 9.4e-5 | 5.930 ± 0.024 | 0.011 ± 1.8e-4 | 6.002 ± 0.043 | 0.017 ± 1.3e-4 | 5.886 ± 0.027 | 0.013 ± 1.3e-4 |
| RF | 9.860 ± 0.009 | 0.238 ± 3.2e-4 | 10.218 ± 0.003 | 0.260 ± 2.9e-4 | 9.884 ± 0.002 | 0.256 ± 3.0e-4 | 9.988 ± 0.005 | 0.251 ± 3.1e-4 |
| STRUCTUREFLOW | **5.517 ± 0.041** | **0.013 ± 3.5e-5** | **5.717 ± 0.021** | **0.012 ± 2.5e-4** | **5.816 ± 0.023** | **0.019 ± 3.7e-4** | **5.683 ± 0.028** | **0.015 ± 2.2e-4** |

and show that STRUCTUREFLOW exhibits improved performance on the joint inference task compared to existing approaches, which either tackle the individual tasks in isolation and/or are limited in their ability to address both problems. Moreover, we showcase the application of our method on a challenging real system for recovering network structure and simultaneously predicting the system response of left-out (*unseen*) interventions.

**Limitations & future work.** In this work, we focused on the problem of joint structure learning and dynamical inference for stochastic population dynamics. With this come non-standard challenges, e.g. evaluating structure learning performance requires knowledge of the ground truth directed graphs which define the data generative process of the dynamical system. To address this, we considered a suite of synthetic and biologically meaningful simulated systems (Pratapa et al., 2020), such that both tasks can be evaluated in unison, which emulate real systems. However, these simulated systems may be limited in their ability to fully reflect system behavior in real settings. For example, our simulated knockout conditions may not exactly reflect realistic CRISPR effects, such as imperfect knockouts. We believe exploring settings with imperfect intervention is a fruitful direction for future work.

In a similar vein, evaluating structure learning performance in real-data settings requires experimental validation (which was accomplished by Tokumasu et al. (2023) to produce what we label as the Renge dataset. Such experimental validation is costly to acquire and unrealistic in large systems. Thus, this experimental validation is limited to a small set of system variables, and is still at best an estimate of the ground truth network. Evaluating performance of structure learning methods in real-data settings remains an unsolved challenge. We also do not consider certain aspects of physical processes that are prevalent in natural systems, such as modeling population dynamics while considering interacting particles (Atanackovic et al., 2024) and/or structure learning of non-stationary networks (Lu et al., 2025), but view these directions as natural extensions for STRUCTUREFLOW. Incorporating biological priors (either through the choice of reference process (Petrović et al., 2025; Zhang and Stumpf, 2025) or by directly imposing knowledge of structural relationships on the network (Roohani et al., 2024)) is another interesting direction for future work. Lastly, using optimal couplings acquired using marginals across all timepoints is a directly applicable extension to STRUCTUREFLOW, and may help improve overall performance of both structure learning and dynamical inference (Chen et al., 2023; Theodoropoulos et al., 2025; Rohbeck et al., 2025).

We remark, STRUCTUREFLOW is designed to be built upon and extended to all aforementioned settings. In this work, we provide a methodological and experimental foundation amenable to further development.

REPRODUCIBILITY STATEMENT

To facilitate the reproducibility of our results and findings, all source code and datasets will be released upon publication.

We also provide the necessary mathematical preliminaries, background, and introduction to our methods in Section 2 and 3, respectively. We further outline experimental details in Appendix A, where we expand on the mathematical details of NGM and RF, provide the experimental details we used to train STRUCTUREFLOW, and clarify our data setups across synthetic (linear), BoolODE, and single-cell CRISPR perturbation settings. In Appendix B, we provide details on our implementation, including Algorithm 1, which outlines the core STRUCTUREFLOW training loop.

LLM USAGE STATEMENT

Large Language Models (LLMs) were not used during ideation or writing, and thus, are not regarded as a contributor to this work.

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

## A  EXPERIMENTAL DETAILS

### A.1  NEURAL GRAPHICAL MODELS FOR DYNAMICAL SYSTEMS

Unlike static graphical models, where variables are assumed to be independent and identically distributed, dynamical systems require modeling dependencies in continuous time.

Neural graphical models (NGMs) parameterize the vector field $\boldsymbol{v}(\boldsymbol{x}(t))$ using deep neural networks (Bellot et al., 2022). Specifically, a neural dynamic structural model (NDSM) is defined as:

$$dx_j(t) \ = \ v_j^\theta(\boldsymbol{x}(t)) \, dt \ + \ dw_j(t), \qquad \boldsymbol{x}(t_0) = \boldsymbol{x}_0$$

where $v_j^\theta \in \mathcal{F}$ are analytic functions parameterized by neural networks with trainable weights $\theta \in \Theta$. Each function $v_j^\theta$ can be expressed as:

$$v_j^\theta(\boldsymbol{x}) \ = \ \psi\Big(\psi\big(\cdots\psi(\boldsymbol{x}\,\theta_j^A)\,\theta_j^1\cdots\big)\,\theta_j^K\Big),$$

where $\theta_j^A \in \mathbb{R}^{d\times h}$ is the first (graph) layer, $\theta_j^k \in \mathbb{R}^{h\times h}$ for $k = 1,\ldots,K-1$ and $\theta_j^K \in \mathbb{R}^{h\times 1}$ are the deeper layers, and $\psi$ is an activation function. Directed edges in the inferred graph correspond to non-zero partial derivatives of these networks, and indicate direct causal influences between variables. The adjacency can be inferred if:

$$G_{ji} \neq 0 \quad \Longleftrightarrow \quad \big\| \partial_i \, v_j^\theta \big\|_{L_2}$$

Now to recover $G$, structure learning can be formulated as a penalized optimization problem:

$$\arg\min_{v^\theta} \ R_n(\theta) \ = \ \min_{v^\theta} \ \frac{1}{n}\sum_{m=1}^{n} \big\|\boldsymbol{x}(t_m) - \hat{\boldsymbol{x}}(t_m)\big\|_2^2,$$

$$\text{subject to} \ \ d\boldsymbol{x}(t) = \boldsymbol{v}^\theta(\boldsymbol{x}(t))\,dt \ \ \text{and} \ \ \rho(\theta) \leq \eta,$$

where $\rho(\theta)$ is a regularization term. In (Bellot et al., 2022) this is often implemented using group lasso (GL) or adaptive group lasso (AGL):

$$\rho_{\text{GL}}(\theta) \ = \ \lambda_{\text{GL}} \sum_{j=1}^{d}\sum_{i=1}^{d} \big\|(\theta_j^A)_{i,:}\big\|_2, \qquad \rho_{\text{AGL}}(\theta) \ = \ \lambda_{\text{AGL}} \sum_{j=1}^{d}\sum_{i=1}^{d} \frac{\big\|(\theta_j^A)_{i,:}\big\|_2}{\big\|(\hat{\theta}_j^A)_{i,:}\big\|_2^\gamma}.$$

Here, $\lambda_{\text{AGL}}, \lambda_{\text{GL}}$ control the regularization strength, and $(\hat{\theta}_j^A)_{i,:}$ is an initial estimate of the parameters.

Given an estimate of $\boldsymbol{v}^\theta$, NGMs can be extended to irregularly sampled and non-linear time series data by numerically computing the forward trajectory with an ODE solver:

$$\hat{\boldsymbol{x}}(t_1), \hat{\boldsymbol{x}}(t_2), \ldots, \hat{\boldsymbol{x}}(t_n) \ = \ \text{ODESolve}\big(\boldsymbol{v}^\theta, \, \boldsymbol{x}(t_0), \, t_1, t_2, \ldots, t_n\big)$$

enforcing the constraint $d\boldsymbol{x}(t) = \boldsymbol{v}^\theta(\boldsymbol{x}(t))\,dt$ while optimizing for $R_n(\theta)$ (Chen et al., 2018). While the NGM was originally developed to identify causal dependencies in deterministic systems governed by ODEs, our work extends their applicability to stochastic systems by using the probability flow formulation of the underlying SDE. Differently from Bellot et al. (2022), we train an NGM which parameterizes the probability-flow ODE, whose instantaneous drift is

$$\hat{\boldsymbol{v}}_t(\boldsymbol{x}; \Theta) \ = \ \boldsymbol{v}^\theta(\boldsymbol{x}) \ - \ \frac{\sigma^2}{2}\,\boldsymbol{s}_t^\phi(\boldsymbol{x}),$$

which is equivalent in marginal behavior to the original SDE but removes the stochasticity by incorporating the score function $\nabla \log p_t(\boldsymbol{x})$. This allows us to retain the NGM for structure discovery, while still modeling data generated from inherently noisy dynamics.

### A.2  REFERENCE FITTING (RF)

In contrast to STRUCTUREFLOW, Reference Fitting (Zhang, 2024) (RF) starts from the principle that the inferred couplings $\pi$ should minimize entropy with respect to a reference process, with the joint-optimization problem defined as:

$$\min_{A\in\mathbb{R}^{d\times d}, b\in\mathbb{R}^d} \ \min_{\pi\in\mathcal{C}(\mu,\mu')} \sigma^2 \text{KL}\left(\pi | K_{(A,b)}^\sigma\right) + \mathcal{R}\,(A,b)$$

where $\mathcal{R}$ is a regularizer term, $A$ and $b$ are the linear and constant terms, respectively, of an Ornstein-Uhlenbeck (OU) process of the form:

$$\mathrm{d}X_t = (AX_t + b)\,\mathrm{d}t \ + \sigma\mathrm{d}B_t$$

and $K^\sigma_{(A,b)}$ is the transition kernel of the OU process.

Perturbations can then be modeled in this set-up by the linear interaction matrix $A^{(g)}$, where $g$ represents a knocked-out gene, and thus the $g$th is zeroed out. Thus, for any gene intervention (knockout) $g$, the reference OU process takes the form:

$$\mathrm{d}X_t = \left( A \odot M^{(g)} \right) X_t^{(g)} \mathrm{d}t + \sigma \mathrm{d}B_t$$

where $M_{ij}^{(g)} = \mathbf{1}_{i \neq g}$ is the masking matrix for intervention $g$. This allows the RF process to learn interactions which may have very low signal in the observational (wild-type) data, and thus cannot be learned from the observed dynamics alone. The transition kernel, $K^\sigma_{(A,b)}$, is then approximated by separating the drift and noise terms to model the kernel's mean and covariance, respectively:

$$\mu_t = e^{tA} x_0, \Sigma_t = \sigma^2 t I$$

This yields a transition kernel of the form:

$$K_t(x, x') \propto \exp\left( -\frac{||e^{tA}x - x'||_2^2}{2\sigma^2 t} \right)$$

Running an alternating optimization over the couplings $\pi$ and reference dynamics yields both the reference process itself and couplings which correspond to this process.

**Trajectory fitting (ODE view).** Given $v^\theta$, irregularly sampled trajectories can be obtained by numerical integration,

$$\hat{\boldsymbol{x}}(t_1), \ldots, \hat{\boldsymbol{x}}(t_n) = \mathrm{ODESolve}\left( v^\theta, \boldsymbol{x}(t_0), t_1, \ldots, t_n \right),$$

and one may minimize a data misfit $R_n(\theta) = \frac{1}{n} \sum_{i=1}^n \|\boldsymbol{x}(t_i) - \hat{\boldsymbol{x}}(t_i)\|_2^2$ with the above sparsity penalty (Chen et al., 2018; Bellot et al., 2022).

**Stochastic dynamics via probability flow.** While NGMs were introduced for deterministic ODEs, we extend them to stochastic systems by parameterizing the *probability-flow ODE* associated with an SDE. Let the instantaneous probability-flow drift be

$$\hat{\boldsymbol{v}}_t(\boldsymbol{x}; \Theta) = \boldsymbol{v}^\theta(\boldsymbol{x}) - \frac{\sigma^2}{2} \boldsymbol{s}_t^\phi(\boldsymbol{x}),$$

where $\Theta = \{\theta, \phi\}$, $\boldsymbol{v}^\theta$ is the NGM (autonomous) drift defined above, and $\boldsymbol{s}_t^\phi(\boldsymbol{x}) = \nabla_{\boldsymbol{x}} \log p_t(\boldsymbol{x})$ is a score network. This PF-ODE induces the same marginals $p_t$ as the original SDE but removes stochasticity by absorbing it into the score term. We therefore retain the NGM's FC1 layer $\theta^A$ for *structure discovery* (via GL/AGL on $A_{ji} = \|(\theta_j^A)_{i,:}\|_2$) while modeling data generated by noisy dynamics.

### A.3 ON THEORETICAL GUARANTEES FOR STRUCTURE RECOVERY FROM POPULATION DYNAMICS VIA NGMs

**Edge Recovery in the NGM and STRUCTUREFLOW.** Although our vector field is parameterized by a neural network, using a sparse weight matrix as the first layer of the network provides a principled approach to encode, and likewise to recover, the underlying interaction structure of the non-linear system (Bellot et al., 2022). For continuous-time dynamics Bellot et al. (2022) denote directed edge(s) $x_k \to x_j$ whenever the drift $f_j$ has a **functional dependence** on $x_k$, i.e. $\partial_k f_j \neq 0$. They show that for any analytic neural network parameterization of $f_j$, every true functional dependency must pass through the input layer and conversely, if $\partial_k f_j = 0$, there exists an equivalent parameterization where the entire k-th column of the first-layer weight matrix is zero. This justifies the application group lasso solely to the first-layer weights, because zeros in that layer correspond exactly to absent edges in the underlying continuous system. Bellot et al. (2022) show that with enough data, adaptive group lasso recovers the correct sparsity pattern. STRUCTUREFLOW relies on the same principle because our autonomous vector field makes use of the same formulation of the NGM.

**On identifiability of underlying structure via the NGM and flow matching.** In general, the question of identifiability for directed structure is challenging, even in the simplest possible case of linear dynamics (see e.g. Wang et al. (2024)). Similarly, Zhang (2024) were not able to provide an identifiability result but only show well-posedness of their optimization problem.

To our knowledge, identifiability in the joint trajectory inference and structure learning problem from population snapshots is an open problem and is particularly challenging due to the interdependence of the learned structure. We thus leave a general theoretical study of identifiability to future work. In what follows, we reason that the NGM with Group Lasso recovers the true underlying structure in

the idealized setting where the drift function and score can be reconstructed exactly. We consider autonomous dynamics, where the drift field $v$ is given by a NGM model with parameters $\theta_0$

$$dX_t = v(X_t; \theta_0)dt + \sigma dB_t.$$

Equivalently, this can be written as a probability flow, with $u_t(x) = v(x) - D\nabla \log p_t$ and $D = \frac{1}{2}\sigma\sigma^\top$:

$$\partial_t p_t = -\nabla \cdot (p_t(x)u_t(x)).$$

Consider the case where samples of the flow $u_t(x)$ and score $\nabla \log p_t(x)$ can be accessed exactly, where $(t, x) \sim p_t(x)$. Writing $\hat{v}_t(x; \theta)$ to be the reconstructed vector field, parameterized as a NGM with parameter $\theta$. The corresponding probability flow is

$$\hat{u}_t(x) = \hat{v}(x) - D\nabla \log p_t.$$

The flow matching objective with target $u_t(x)$ is thus

$$\min_\theta \mathbb{E}_{t\sim[0,1], x\sim p_t} \|\hat{u}_t(x) - u_t(x)\|_2^2 \implies \min_\theta \mathbb{E}_t \mathbb{E}_{x\sim p_t} \|\hat{v}(x; \theta) - v(x; \theta_0)\|^2.$$

In this idealized setting, flow matching amounts to least-square regression on the target vector field $v(x, \theta_0)$. Let $\{x_i\}_{i=1}^N$ a sample of size $N$ drawn following $t \sim [0, 1]$ and $x \sim p_t$. Then, consider the empirical and population risks $R_N, R$

$$R_N(\theta) = \min_\theta \frac{1}{N} \sum_{i=1}^N \|\hat{v}(x_i; \theta) - v(x; \theta_0)\|^2,$$

$$R(\theta) = \mathbb{E}_{t,x} \|\hat{v}(x; \theta) - v(x; \theta_0)\|^2.$$

Our goal is to characterize identifiability of the structural graph from samples. Then this falls into the framework considered by Dinh and Ho (2020). Specifically, the results of [Dinh and Ho (2020), Theorem 3.6] guarantees local consistency of the learned graph from minimizing (1) in the limit where $n \to \infty$.

### A.4 EXPERIMENT DETAILS

**Model architectures.** The learned flow model is a NGM as described above. The score and residual models are standard multilayer perceptrons with ReLU activations. These support time-varying and conditionally encoded inputs (knockout vectors). All networks are initialized using Gaussian weights: $\mathcal{N}(0, 0.1)$ for weights and $\mathcal{N}(0, 1 \times 10^{-2})$ for biases.

**Trajectory simulation.** Simulations are run between pairs of discrete timepoints $(t_k, t_{k+1})$ where $k = 0, \ldots, T - 1$ using 100 Euler steps. We consider two rollout strategies: (a) stochastic differential equation (SDE) sampling and (b) probability flow ODE (PF-ODE) sampling. Both start from $p_{t_k}$, the empirical distribution of cells at timepoint $t_k$, but differ in how the dynamics are integrated.

ODE SIMULATION The probability flow ODE (PF-ODE), given by

$$\dot{x}_t = \boldsymbol{v}(x_t) - \frac{\sigma^2}{2}\boldsymbol{s}_t^\phi(x_t), \quad x_0 \sim p_{t_k}, \tag{15}$$

describes the deterministic evolution of the density $p_t$ using our learned score $\boldsymbol{s}_t^\phi$ and flow $\boldsymbol{v}(x_t)$. The score corrects the flow to match the evolving density, while the deterministic rollout removes sample-path noise, yielding smoother trajectories but potentially underestimating variability.

**Evaluation metrics.** We report both Wasserstein-2 (W2) distance and squared Maximum Mean Discrepancy (MMD$^2$) between predicted and ground truth distributions. W2 is computed using the exact Earth Mover's Distance from the POT library with a squared Euclidean cost matrix.

MMD$^2$ is computed using a radial basis function (RBF) kernel. For each evaluation, we average over five kernel bandwidths. Each bandwidth is defined via a kernel scale parameter $\sigma \in \{0.01, 0.1, 1, 10, 100\}$. The final MMD$^2$ score is the mean of the values computed under each of these five kernel settings.

For structure learning evaluation, we compute the Area Under the Receiver Operating Characteristic curve (AUROC) and Average Precision (AP) of the learned graph against the ground-truth. AUROC measures the trade-off between true positive and false positive rates across all classification thresholds. AP summarizes the precision-recall curve as the weighted mean of precisions achieved at each threshold.

**Experiment protocols.**

- **Leave-one-out (timepoint):** For each dataset, we iteratively exclude data from a single discrete timepoint $t_k$ for $k \in \{1, \ldots, T-2\}$ from training (we only hold out the interior timepoints). The model is trained on the remaining timepoints $\{t_j : j \in \{0, \ldots, T-1\} \setminus \{k\}\}$ and then used to simulate the transition from timepoint $t_{k-1}$ to $t_k$. We report the average performance across all held-out timepoints by training a separate model for each excluded $t_k$.

- **Leave-one-out (knockout):** For each dataset, we select the first three gene knockouts and exclude one per trained model. Each model is then used to simulate transitions from $t_k$ to $t_{k+1}$ conditioned on the held-out inter across discrete timepoints $k = 0, \ldots, T-2$. A separate model is trained for each held-out knockout, and results are averaged across the three excluded interventions.

**Seed values.** All experiments are repeated across fixed seed values. For structure learning experiments, we use 5 seeds: $\{1, 2, 3, 4, 5\}$. For computationally intensive experiments (leave-one-timepoint-out and leave-one-knockout-out) we use 3 seeds: $\{1, 2, 3\}$

**Hardware.** All experiments were implemented in PyTorch and run on a MacBook Pro with an Apple M1 Pro CPU and 32GB RAM. No GPU is needed. All models train in under 10 minutes.

A.5   HYPERPARAMETERS

We report all hyperparameters used in training our models. Unless otherwise specified, we use the AdamW optimizer with a fixed weight decay of $1 \times 10^{-2}$.

**Synthetic datasets.** Models are trained for 15,000 steps using a batch size of 64 and a learning rate of $3 \times 10^{-3}$. Regularization and model-specific settings are provided below:

Table 5: Hyperparameters for synthetic datasets.

| Hyperparameter | Value |
|---|---|
| Training steps | 15,000 |
| Batch size | 64 |
| Learning rate | $3 \times 10^{-3}$ |
| Weight decay | $1 \times 10^{-2}$ |
| $\alpha$ (Score term influence) | 0.1 |
| $\ell_2$ regularization | $5 \times 10^{-6}$ |
| Residual regularization | $1 \times 10^{-3}$ |
| Group Lasso regularization | 0.04 |
| Knockout hidden layers | [100] |
| Score model hidden layers | [100, 100] |
| Residual model hidden layers | [64, 64] |
| SDE noise scale $\sigma$ | 1.0 |

**Real data.** For the real data experiments, some hyperparameters are overriden:

Table 6: Overriden Hyperparameters for real data (RENGE).

| Hyperparameter | Value |
|---|---|
| Training steps per fold | 10,000 |
| Learning rate | 0.07 |
| $\ell_2$ regularization | $1 \times 10^{-9}$ |
| $\alpha$ (Score term influence) | 0.73 |
| Group Lasso regularization | 0.0008 |
| Knockout hidden layers | [128] |
| Score model hidden layers | [128, 128] |
| Residual model hidden layers | [128, 128] |

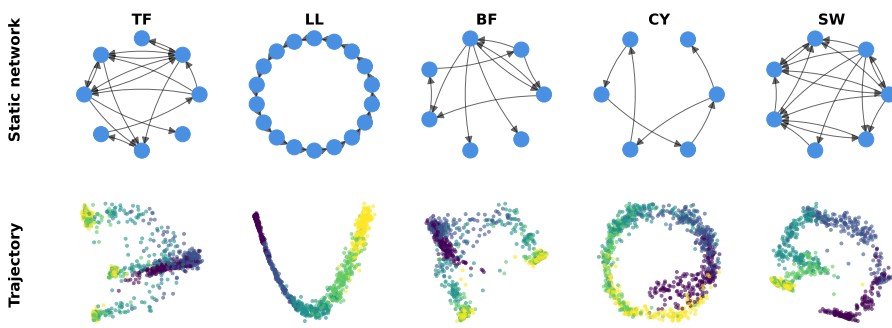

Figure 6: **BoolODE synthetic biological systems.** Network topologies and example trajectory visualizations for the six synthetic biological systems used in our experiments: trifurcating (TF), cyclical (CY), long linear (LL), swirling (SW), bifurcating (BF), and hematopoietic stem cell (HSC) systems. Figure is sourced from Zhang (2024).

We used Optuna to perform hyperparameter optimization, and selected configurations that maximized the area under the precision-recall curve (AUPR). The search space included key hyperparameters such as the learning rate, regularization strength, group lasso regularization, batch size, and model capacity (e.g., number of hidden units). For each trial, model performance was evaluated using cross-validation, and the trial achieving the highest mean AUPR across folds was selected as the optimal configuration. We note that the score term was of higher importance here. This is likely due to the fact that scRNA-sequencing data is much noisier than the synthetic systems.

## A.6 SYNTHETIC LINEAR SYSTEM

We generate synthetic linear systems to evaluate structure learning performance across varying dimensionality and sparsity levels. The underlying network structure is generated using Erdos-Renyi random graphs (Erdös and Rényi, 1959), where edges are placed independently with probability $p$ to achieve the desired sparsity level (5%, 20%, or 40% of possible edges).

For a system with $d$ variables, we construct a random adjacency matrix $\boldsymbol{A} \in \mathbb{R}^{d \times d}$ where non-zero entries are sampled to avoid weak interactions. Specifically, edge weights are drawn from the union of two uniform distributions: $\mathcal{U}(-1.0, -0.5) \cup \mathcal{U}(0.5, 1.0)$, ensuring all edges have magnitude $\geq 0.5$. Positive and negative edges occur with equal probability, representing activating and inhibiting interactions respectively. The linear SDE governing the system dynamics is:

$$d\boldsymbol{x}_t = \boldsymbol{A}\boldsymbol{x}_t dt + \sigma d\boldsymbol{B}_t \tag{16}$$

where $\boldsymbol{x}_t \in \mathbb{R}^d$ represents the system state, $\boldsymbol{A}$ encodes the linear interactions, and $\sigma = 0.1$ is the noise level. We simulate trajectories over $T = 5$ time points, generating $N = 1000$ samples per time point. The initial conditions are sampled from a multivariate Gaussian distribution $\mathcal{N}(\boldsymbol{0}, 0.5\boldsymbol{I})$.

## A.7 BOOLODE

BoolODE was employed to simulate 1000 cells independently for each trajectory. To obtain time-resolved snapshots, we divided the simulation into T=5 discrete intervals. Expression outputs from simulations were log-transformed for use in subsequent analysis. Knockout trajectories were produced by altering Boolean rules so that the targeted gene retained only self-activation, while its initial expression level was set to zero.

To evaluate performance across different conditions, we considered gene expression values across across the following conditions for each dataset type:

- HSC: Gata1, Fli1, Fog1, Eklf, Scl, Gfi1, EgrNab, cJun
- TF, BF, CY, SW, and LL: observational (wild-type), g2, g3, g4, g5, g6, g7, g8

Since we considered 1000 cell expression trajectories, each measured across 800 continuous time and pseudo-time values, and divided across eight conditions, we were left with 100,000 cell measurements per condition. This was then further discretized into T=5 time bins.

## A.8 SINGLE-CELL CRISPR PERTURBATION TIME-SERIES

Following Zhang (2024), we use raw count data from Tokumasu et al. (2023), retrieved from the Gene Expression Omnibus database (accession GSE213069). Zhang et al. use the following procedure:

- Columns corresponding to gRNAs were removed.
- Counts were normalized using the `scanpy.pp.normalize_total` function with default options.
- The data was log-transformed.
- Prior to dimensionality reduction, highly variable genes were selected using scanpy.pp.highly_variable_genes.

We considered the set of 103 transcription factors that Zhang (2024) uses and included only cells that received a single knockout. To construct the ChIP-seq reference, we obtained experimental binding information from ChIP-Atlas (Zou et al., 2024) for the following TFs (within a 1kb window): Chd7, Ctnnb1, Dnmt1, Foxh1, Jarid2, Kdm5b, Med1, Myc, Nanog, Nr5a2, Pou5f1, Prdm14, Sall4, Sox2, Tcf3, Tcf7l1, Ubtf, Znf398 with a 1kb window. We then further reduced the knockouts to a group of 8 most prominent ones as referenced in Zhang (2024). We show a visualization of the dateset in UMAP space in Figure 7.

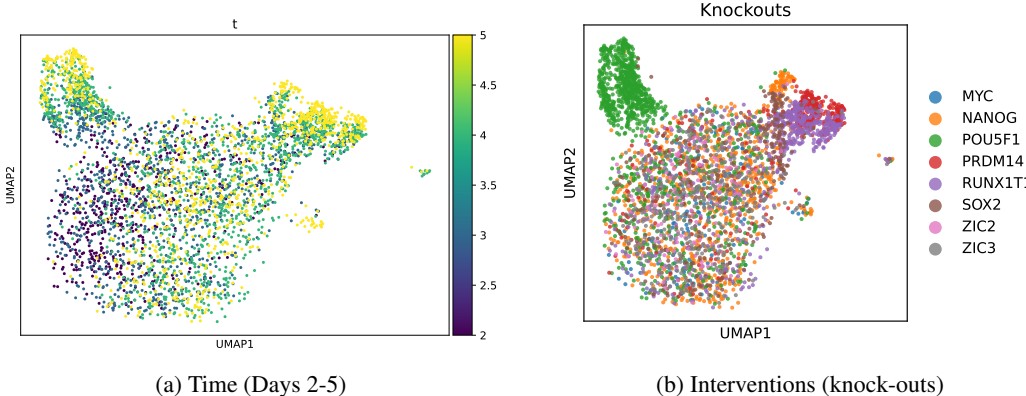

(a) Time (Days 2-5)                    (b) Interventions (knock-outs)

Figure 7: **Visualization of real (Renge) dataset.** (a) UMAP visualization showing temporal progression of wildtype (unconditional) data across days 2-5, with colors indicating time points. (b) UMAP visualization of intervention conditions showing trajectories under different gene knock-out perturbations, with colors indicating the specific knocked-out gene.

## B    IMPLEMENTATION DETAILS

### B.1    STRUCTUREFLOW ALGORITHM

---

**Algorithm 1** STRUCTUREFLOW Training Loop

---

**Require:** Training data across conditions $c$, hyperparameters $\alpha, \sigma, \lambda_{\text{GL}}$, number of steps $N$, $d$ variables
1: Initialize parameters $\Theta = \{\theta, \phi\}$, Optimizer $Opt$
2: **for** step $n = 1$ to $N$ **do**
3:      $\boldsymbol{M}^{(c)} \leftarrow M_{ji}^{(c)} = 0$ if $i = c, j \neq c$, $M_{ji}^{(c)} = 1$ otherwise **for all** $j, i \in 1, ..., d \times 1, ..., d$
4:      $\boldsymbol{k}^{(c)} \leftarrow k_j^{(c)} = 1$ if $j$ perturbed under $c$, else $k_j^{(c)} = 0$ **for all** $j$
5:      Sample $(\boldsymbol{x}_t, t, \boldsymbol{s}_t, \boldsymbol{v}_t^\circ)$ from SB for condition $c$
6:      $\boldsymbol{s}_t^\phi \leftarrow \boldsymbol{s}_t^\phi(\boldsymbol{x}_t | \boldsymbol{k}^{(c)})$
7:      $\hat{\boldsymbol{v}}_t \leftarrow \boldsymbol{v}^{\theta_j}(\boldsymbol{x}_t \mid \boldsymbol{M}^{(c)}) - \frac{\sigma^2}{2} \boldsymbol{s}_t^\phi(\boldsymbol{x}_t \mid \boldsymbol{k}^{(c)})$                    ▷ Using $\boldsymbol{M}^{(c)} \odot \theta_j^A$
8:      $\mathcal{L} \leftarrow \sum_c \mathbb{E}\left[(1-\alpha)\left\|\hat{\boldsymbol{v}}_t(\boldsymbol{x}_t; \Theta | \boldsymbol{M}^{(c)}, \boldsymbol{k}^{(c)}) - \boldsymbol{v}_t^\circ\right\|^2 + \alpha \left\|\boldsymbol{s}_t^\phi(\boldsymbol{x}_t | \boldsymbol{k}^{(c)}) - \boldsymbol{s}_t\right\|^2\right]$
9:      $\boldsymbol{g} \leftarrow \nabla_\Theta \mathcal{L}$
10:     Update parameters $\Theta \leftarrow Opt(\Theta, \boldsymbol{g})$
11:     $\theta_j^A \leftarrow \text{Prox}_{\eta \lambda_{\text{GL}} \|\cdot\|_{\text{group}}}(\theta_j^A)$                    ▷ group-wise proximal update on first layer
12: **end for**

---

## B.2 Gene Regulatory Network Inference Baselines

All baselines were trained with the same pre-processing steps, train-test splits, and evaluation metrics. In addition, we experimented with the hyper-parameters of the baseline methods. For methods that considered the same BoolODE dataset(s) (Zhang, 2024; Zhao et al., 2024), we used the respective hyper-parameters reported in the work. For other methods, we experimented with hyper-parameters settings, but found that the hyper-parameters set by the authors of the respective baseline methods performed generally well.

**SCODE:** Sparse COding for Differential Equations (SCODE) from Matsumoto et al. (2017) is a method for reconstructing GRNs from single-cell time series by fitting a linear ODE system to gene expression trajectories. The method infers a sparse gene interaction matrix such that simulated expression profiles recapitulate observed temporal dynamics. In the synthetic case, we take the transposed expression matrix and pseudotime vector that is given to us in both the observational and interventional settings and write them to disk in a format compatible with SCODE. We then execute the model via the Ruby + R wrapper developed by Matsumoto et al. (2017), and read the mean interaction matrix from the resulting output files, which gives us an adjacency matrix. SCODE does not work with interventional data so we do not expect an increase in performance.

**SINCERITIES:** Single-Cell Network Inference by Covariance Regression for Time Series, from Gao et al. (2018) is an R-based algorithm for inferring directed GRNs from time-course single-cell data. SINCERITIES combines information from lagged covariances and local linear regression to estimate causal effects, and it outputs signed edge weights. We use the R script developed by the authors and the given expression matrix and psuedotime matrix to create a `GRNPrediction.txt` file which gives a gene $\times$ gene matrix.

**GLASSO:** Graphical Lasso is a classical method for inferring sparse undirected graphical models from high dimensional data. GLASSO estiamtes the precision (inverse covariance) matrix under an $l_1$ penalty to enforce sparsity, such that nonzero entries correspond to conditional dependencies between genes. This method does not work with time-series data. We first standardize the gene expression matrix, and choose the regularization parameter via cross-validation. The estimated precision matrix is negated (higher values correspond to stronger interactions), and the diagonal is set to zero. Our result is an adjacency matrix with symmetric edge weights that represent partial correlations between genes.

**dynGENIE3:** This method from Huynh-Thu and Geurts (2018) is an extension of the GENIE3 algorithm (Huynh-Thu et al., 2010) to reconstruct GRNs from time-series single-cell or bulk gene expression data. The method frames network inference as a feature selection problem, where for each gene, a random forest regression model is trained to predict its expression at each time point as a function of all other genes' expressions at earlier time points. The regulatory strength is aggregated over all of the trees and time lags, and yields a directed adjacency matrix. We aggregated our expression matrix `adata.X`, which is an annData object, by time points (mean expression per time) and store it in a list X, and keep the corresponding points in `t`. The dynGENIE3 model is then called with these two parameters as input and we save the resulting adjacency matrix. By averaging across all cells at each time $t$ we effectively create pseudo-bulk expressions, then stack these per-timepoint means into a matrix so as to be a single trajectory. This approach is corroborated by Moscardó García et al. (2025) who use a nearly identical procedure for dynGENIE3.

**RENGE:** RENGE Tokumasu et al. (2023) can use both temporal and interventional data. Renge uses pseudotime trajectories from single-cell data, and CRISPR knockout information, with a two-step inference strategy, that involves a regression model to predict expression as a function of genes, and network aggregation to aggregate the inferred effects across all perturbations and time points. This gives us a gene $\times$ gene matrix based on both the endogenous dynamics and systematic perturbation present in the dataset. The main experiment from Tokumasu et al. (2023) was what was used in our paper and in Zhang (2024). We use the model that Tokumasu et al. (2023) provide in their codebase and do further experimentation, using that matrix as a baseline.

**TIGON:** Trajectory Inference with Growth via Optimal transport and Neural networks (TIGON) (Sha et al., 2024) models time-series scRNA-seq as a continuum of cell states whose density evolves under a reaction-transport dynamic. It represents a time-dependent cell-state density represented by two fields: a velocity field that models state transitions, and a growth field that accounts for birth/death and mass change. TIGON integrates the learned dynamics using ODE integration per time step, and learns to minimize a the Wasserstein-Fisher-Rao distance. At inference, TIGON learns to reconstruct the continuous trajectories and velocities for individual cells, estimates time-resolved

growth maps, and can learn a gene-regulatory structure from the Jacobian of its flow. TIGON uses either a PCA layer or an autoencoder to first encode the ambient data. This is a step for the "dimensionless-solver" used within TIGON, which is optimized to operate in 10 dimensional space. All of the other baselines, and STRUCTUREFLOW, are trained directly in the ambient space. Thus, in order to provide a comparative setting, we train TIGON in the ambient space as well and forgo the PCA/autoencoder step. We did not encounter challenges with applying TIGON to the high dimensional real (Renge) biological system. TIGON uses Gaussian mixture models to represent $p_t$ (equivalent to our Eq. 2). We also train TIGON in PCA space, and then convert to the ambient space using the following formula: $J_X = W^T J_z W$ where $J_z$ is the latent Jacobian. We note no difference in performance for structure learning.

**OTVelo:** OTVelo Zhao et al. (2024) estimates gene velocity from cell trajectories reconstructed via optimal transport. The method proceeds in two stages. The first stage is trajectory inference where it aligns cell across consecutive time-points using Fused Gromov-Wasserstein (FGW) optimal transport, which uses both expression state and local geometric structure. The second part uses OT pairings to estimate the gene velocities via two different approaches. We consider two versions of this model, OTVelo-Corr and OTVelo-Granger. OTVelo-Corr uses time-lagged correlation analysis that generalizes the standard time-lagged correlation to distributions of cells matched by the OT pairings. OTVelo-Granger, on the other hand, is a regularized linear regression model (Elastic Net) where the velocity of a target gene at time $t + 1$ is predicted from the velocities of regulating gene at time $t$. Given that we discretize all of our data into bins, and OTVelo tests their model on the same datasets, simulated using BoolODE, we apply the model from Zhao et al. (2024) directly in our pipeline.

### B.3 UNBALANCED STRUCTUREFLOW

In order to properly model cell proliferation and death, we formalize the extension of STRUCTURE-FLOW to biological systems where probability mass is not conserved. We generalize the probability flow formulation to include a growth term, derived from Unbalanced Optimal Transport (UOT).

**Unbalanced Optimal Transport** In the standard STRUCTUREFLOW framework, couplings $(x_0, x_1) \sim \pi(x_0, x_1)$ are acquired via a balanced Sinkhorn solver, which enforces strict mass conservation ($\sum \pi_i = \frac{1}{N}$). To model cell proliferation and death, we replace this with UOT. The universalness of our method allows us to easily incorporate unbalancedness into STRUCTUREFLOW by rescaling source and target measures Eyring et al. (2024). We compute the coupling $\pi^*$ between empirical snapshots $\hat{\rho}_{t_k}$ and $\hat{\rho}_{t_{k+1}}$ by minimizing the unbalanced Sinkhorn Divergence Wang et al. (2025); Chizat et al. (2019):

$$\pi^* = \arg \min_{\pi \in \mathbb{R}_+^{N \times M}} \sum_{i,j} C_{ij} \pi_{ij} + \epsilon H(\pi) + \rho[KL(\pi_{\#0}|\hat{\rho}_{t_k}) + KL(\pi_{\#1}|\hat{\rho}_{t_{k+1}})]$$

where $\epsilon H(x)$ is the entropic regularization coefficient, $\rho$ is the marginal relaxation parameter, and $\pi_{\#0}$ are row sums which control the amount of mass transported away from each starting cell. If $\pi_{\#0} < \hat{\rho}_{t_k}$ then the transport plan used less mass than was available and if $\pi_{\#1} > \hat{\rho}_{t_{k+1}}$ then the transport plan delivered more mass than was originally available relative to the source.

**Stochastic Dynamics with Growth** To account for cellular growth, we adopt the formulation proposed in Zhang et al. (2025), where the population density $\rho_t(x)$ evolves according to a Fokker-Planck equation with a source term:

$$\partial_t \rho_t(x) = -\nabla \cdot (\rho_t(\boldsymbol{x}_t) \boldsymbol{v}_t(\boldsymbol{x}_t)) + \tfrac{1}{2} \nabla \cdot (\sigma^2 \nabla \rho_t(\boldsymbol{x}_t)) + g_t(x) \rho_t(x)$$

where $\boldsymbol{v}_t(\boldsymbol{x})$ represents the drift term and $\tfrac{1}{2} \nabla \cdot (\sigma^2 \nabla \rho_t(\boldsymbol{x}_t))$ is the diffusion coefficient. Unlike eq. (2) where probability mass is conserved, the total mass $M_t = \int \rho_t(x) dx$ varies over time according to the net growth rate of the system. We model the joint evolution where the local change in log-density is governed by both the divergence of the flow and the local growth rate, defined as:

$$\frac{d}{dt} \log \rho_t(x_t) = g_t(x_t) - \nabla \cdot u_t(x_t)$$

where $u_t(x)$ is defined in eq. (3). This formulation then allows us to disentangle the vector field $v_t$ from the growth field, so that the density increase is due to the growth field, and the vector field only models transitions. Lastly to learn the growth field $g_\phi(x)$, we adopt the simulation-free strategy proposed in Wang et al. (2025). Instead of solving a differential equation for mass evolution during training, we interpret the row-sums of the unbalanced coupling matrix as static proxies for net proliferation accumulated along a trajectory. For a source sample $x_0^i$, the target growth $G_i$ is defined

by the log-marginal of the transport plan $\pi^*$:

$$G_i = \log\left(\sum_j \pi^*_{ij}\right)$$

This value represents the total probability mass at $t_1$ attributed to the lineage of $x_0^i$. If the sum is greater than 1, the cell has proliferated, otherwise the cell has undergone apoptosis (died). To learn the growth network, we regress the neural network $g_\phi(x,t)$ to the static target across the interpolated trajectory:

$$\mathcal{L}_{\text{growth}}(\phi) = \mathbb{E}_{t,(x_0^i,x_1^j)\sim\pi^*}[\|g_\phi(x_t) - G_i\|^2]$$

## C  NOVELTY STATEMENT

This work introduces key methodological and empirical contributions to joint structure learning and dynamics inference:

**Decomposition of Stochastic Dynamics.** We decompose the probability flow into an autonomous drift $v(x)$ capturing system structure and a time-dependent score function $s_t(x)$ modeling stochasticity. This aligns with biophysical reality where regulatory mechanisms are time-invariant.

**Conditional Dynamics via Interventions.** We incorporate interventional data through knockout conditioning masks, enabling learning from perturbational experiments and interventional trajectory inference on unseen perturbations.

**Simulation-Free Training for High-Dimensional Systems.** Our framework extends simulation-free training to conditional settings, enabling tractable learning on high-dimensional systems (up to 500 variables) while jointly inferring structure and dynamics.

**Comprehensive Evaluation.** We establish extensive benchmarks including scalability analysis, evaluation on six biologically plausible systems with interventional data, and validation on real single-cell CRISPR time-series data with realistic constraints (4-5 time points).

## D  BROADER IMPACT

The genome is the fundamental identity of every living thing. The expression of DNA dictates life. It is a multi-dimensional representation of the state of any living thing. The ultimate goal of our work is to enable improve our understanding of how an organism's genes interact, allowing us to predict the trajectory of its life, and ultimately to computationally develop interventions that can positively change that trajectory. The stochastic dynamical systems we consider here are used ubiquitously in the biophysics and modelling literature. In a biological context, our vector field $v$ can be related to the metaphorical "Waddington's landscape".

## E  ADDITIONAL RESULTS

### E.1  STRUCTURE LEARNING ON SYNTHETIC SYSTEMS WITH OBSERVATIONAL DATA

In 8, we observe that STRUCTUREFLOW achieves competitive results in most systems across both AP and AUROC metrics. Whereas other methods tend to fluctuate widely in performance across these synthetic systems, STRUCTUREFLOW underperforms all other methods only on the SW system, while beating most other benchmarks otherwise.

### E.2  DYNAMICAL INFERENCE OF CONDITIONAL POPULATION DYNAMICS ON SYNTHETIC SYSTEMS USING [SF]$^2$M

We note in Figure 11 that [SF]$^2$M predictions cluster around the synthetic systems' ground-truth trajectories across both observational and gene knockout settings. This is a result of training across both unconditional and conditional data.

### E.3  PREDICTION TO UNSEEN INTERVENTIONS (KNOCK-OUTS) ON SYNTHETIC SYSTEMS

Table 7 shows performance of STRUCTUREFLOW and baselines on the leave-one-knockout-out trajectory inference task. STRUCTUREFLOW outperforms RF on the TF and CY systems, and is competitive on the Curated (HSC) system.

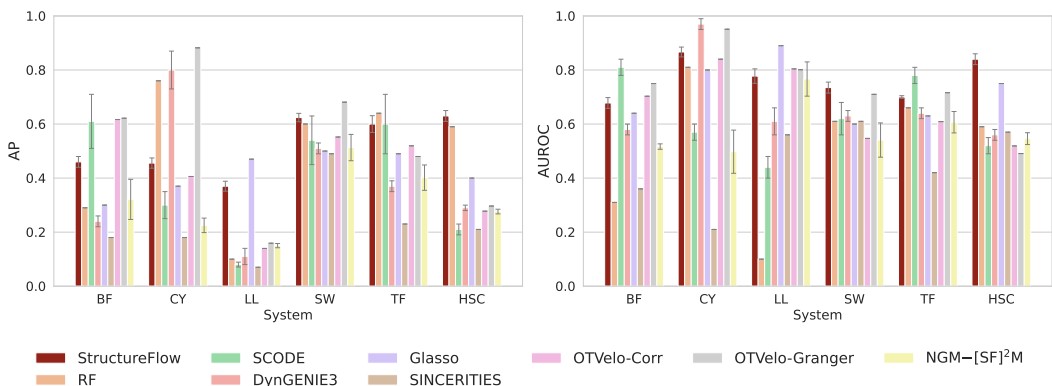

Figure 8: **Structure learning performance across synthetic systems using only observational data**. Shown are average precision (AP) and area under the ROC curve (AUROC) scores for models trained on only observational data

Table 7: **Comparison of left-out intervention (knock-out) prediction on simulated biological systems.** For each dataset, we report Wasserstein-2 ($W_2$) and MMD metrics. Lower is better.

| | TF | | CY | | LL | | HSC | | BF | | SW | |
|---|---|---|---|---|---|---|---|---|---|---|---|---|
| | $W_2\downarrow$ | MMD↓ | $W_2\downarrow$ | MMD↓ | $W_2\downarrow$ | MMD↓ | $W_2\downarrow$ | MMD↓ | $W_2\downarrow$ | MMD↓ | $W_2\downarrow$ | MMD↓ |
| $[SF]^2M$ | 0.964 ± 0.016 | 0.103 ± 0.003 | 0.967 ± 0.028 | 0.277 ± 0.017 | 1.082 ± 0.095 | 0.157 ± 0.010 | 0.886 ± 0.011 | 0.104 ± 0.003 | 1.138 ± 0.031 | 0.220 ± 0.005 | 1.404 ± 0.051 | 0.358 ± 0.015 |
| RF | 1.062 ± 0.000 | 0.098 ± 2.8e-8 | 1.889 ± 0.000 | 0.329 ± 1.5e-8 | 1.887 ± 0.005 | 0.232 ± 6.5e-4 | 0.984 ± 3.4e-10 | 0.065 ± 1.7e-8 | 1.176 ± 0.000 | 0.134 ± 1.1e-8 | 1.285 ± 0.000 | 0.284 ± 3.4e-8 |
| STRUCTUREFLOW | 1.012 ± 0.010 | 0.110 ± 4.1e-3 | 0.924 ± 0.036 | 0.250 ± 1.8e-2 | 1.123 ± 0.037 | 0.141 ± 6.1e-3 | 0.852 ± 0.011 | 0.120 ± 4.2e-3 | 1.079 ± 0.016 | 0.213 ± 7.1e-3 | 1.159 ± 0.032 | 0.330 ± 1.1e-2 |

### E.4 OT-CFM ABLATION ON SYNTHETIC (TRIFURCATING) SYSTEM

In Table 15 we remove the stochastic noise term from the conditional flow matching objective, reducing the formulation to an ODE that models the drift component. This corresponds to the classical flow matching objective Lipman et al. (2023b), where the dynamics align deterministic trajectories. STRUCTUREFLOW is trained to model the process as a Schrödinger Bridge, where the forward-backward SDE explicitly models both drift and diffusion. From the results, we observe that discarding the stochastic component (OT-CFM) substantially degrades both classification (AUROC/AUPR) and distributional alignment metrics ($W_2$), highlighting that noise modeling contributes meaningfully to the structure learning task. For dynamical systems such as cellular trajectories observed across time points, modeling the process as a Schrödinger bridge rather than an ODE provides a more accurate representation of developmental variability and uncertainty.

Table 8: **Comparison of STRUCTUREFLOW with and without stochastic component on the TF synthetic system.** Results show AUROC, AUPR, and distributional distance metrics ($W_2$, MMD).

| | AUROC ↑ | AUPR ↑ | $W_2 \downarrow$ | MMD ↓ |
|---|---|---|---|---|
| **STRUCTUREFLOW** | **0.968 ± 0.005** | **0.932 ± 0.011** | **0.795 ± 0.007** | 0.136 ± 0.002 |
| **STRUCTUREFLOW (OT-CFM)** | 0.791 ± 0.013 | 0.634 ± 0.037 | 0.846 ± 0.022 | **0.062 ± 0.004** |

### E.5 SCALING AND RUNTIME OF VARIOUS BASELINES AND OUR METHOD

Here we provide an additional experiment comparing training time of all methods on the BoolODE **TF** system. For this experiment, all methods are run on the same device (GPU, Quadro RTX 6000), but we note STRUCTUREFLOW can be efficiently trained using a CPU. We only report paramter sizes for neural network-based methods. For STRUCTUREFLOW and the flow matching baseline, we use (·) to denote the additional time used to compute EOT. We use the same number of timepoints (5), dimensionality (8), and number of cells (8000). We add flow matching as a reference point to STRUCTUREFLOW, where we remove regularization terms, the score model, and replace the NGM backbone with a traditional MLP backbone.

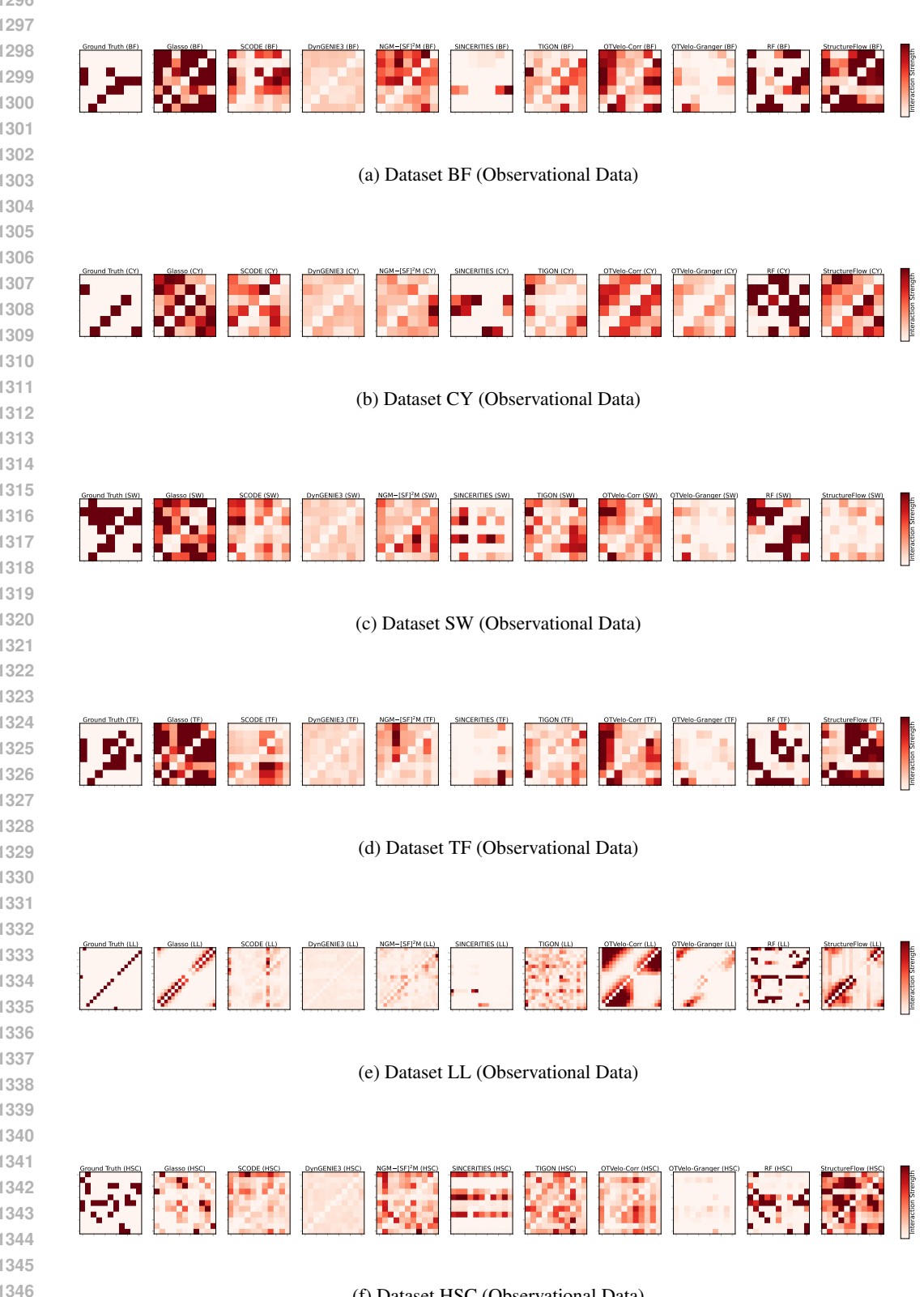

(a) Dataset BF (Observational Data)

(b) Dataset CY (Observational Data)

(c) Dataset SW (Observational Data)

(d) Dataset TF (Observational Data)

(e) Dataset LL (Observational Data)

(f) Dataset HSC (Observational Data)

Figure 9: Six of the synthetic datasets with only observational information. Refer to panels (a)–(f) when discussing individual datasets.

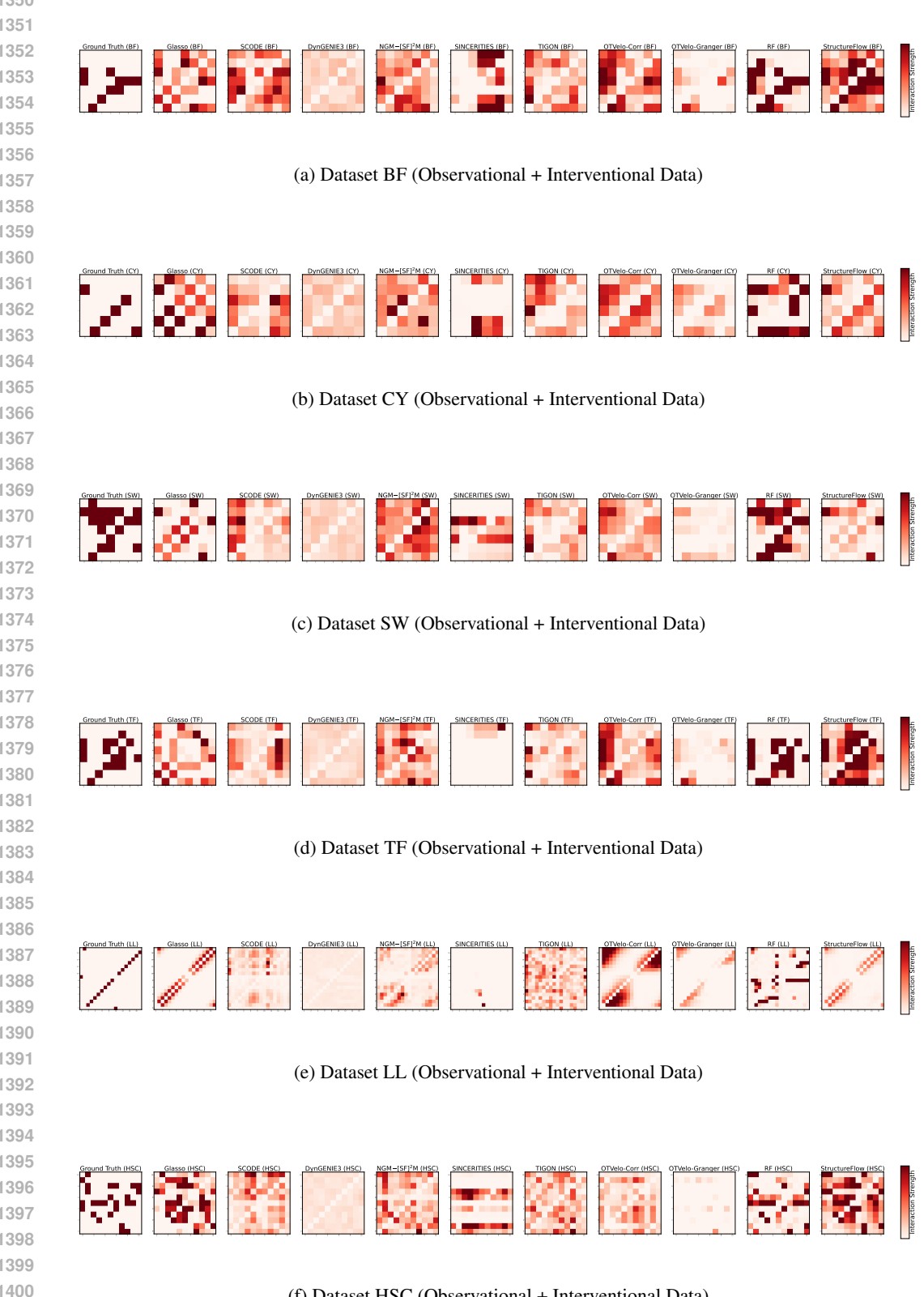

(a) Dataset BF (Observational + Interventional Data)

(b) Dataset CY (Observational + Interventional Data)

(c) Dataset SW (Observational + Interventional Data)

(d) Dataset TF (Observational + Interventional Data)

(e) Dataset LL (Observational + Interventional Data)

(f) Dataset HSC (Observational + Interventional Data)

Figure 10: Six of the synthetic datasets with the addition of interventional information. Refer to panels (a)–(f) when discussing individual datasets.

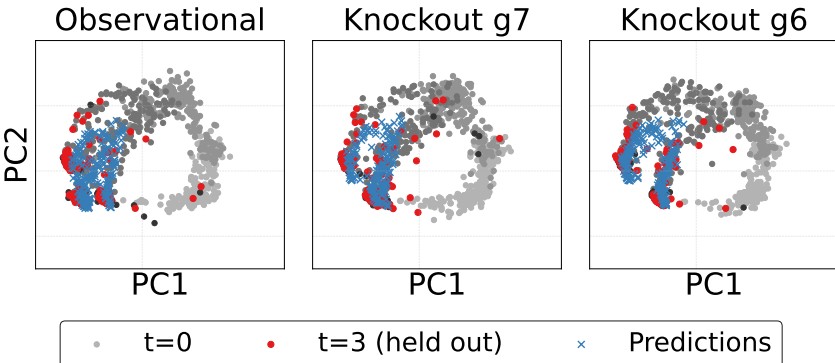

Figure 11: [SF]$^2$M **trajectory inference visualization.** Performance on observational (wild-type) data and two interventions selected for their diverse trifurcating paths. These examples illustrate the trajectory inference problem, where timepoint 3 was excluded during training, and the model simulates with the ODE from timepoint 2 to timepoint 3. Ground truth data is shown in grey, with progressively darker shades as time increases, except for timepoint 3 which is highlighted in red. Blue represent the model's predictions.

Table 9: Model wall-clock time on the Trifurcating (**TF**) BoolODE system. We include parameter counts for all neural network-based methods. We use (·) to denote runtime of EOT.

| Time [s] | Model | Parameter Size |
|---|---|---|
| 263 + (3.3) | STRUCTUREFLOW | 20716 |
| 1511.18 | TIGON | 43140 |
| 82.01 | RF | - |
| 3228.61 | NGM-NODE | 8008 |
| 116.17 + (3.3) | Flow matching | 8008 |
| 208.58 + (3.3) | [SF]$^2$M | 14892 |
| 131.22 | OTVelo | - |
| 5771.41 | SDFL | - |

### E.6 ADDITIONAL ABLATIONS OF HYPERPARAMETERS

In order to assess the stability of our method we vary three hyperparameters that are crucial to the performance of our model: $\alpha$, which varies the weight of the score and flow term; group lasso regularization which determines how sparse the adjacency matrix recovered by the autonomous flow term is; and the size of the hidden dimensions in our flow module (We use one layer in our case).

Table 10: Ablation over the group-lasso regularization term.

| Group Lasso Reg. | 0.0 | 0.0001 | 0.001 | 0.01 | 0.04 | 0.1 | 0.2 | 0.5 |
|---|---|---|---|---|---|---|---|---|
| AUROC | $0.943 \pm 0.004$ | $0.946 \pm 0.004$ | $0.944 \pm 0.004$ | $0.952 \pm 0.002$ | $\mathbf{0.971 \pm 0.008}$ | $0.853 \pm 0.032$ | $0.851 \pm 0.017$ | $0.879 \pm 0.016$ |
| AP | $0.804 \pm 0.014$ | $0.816 \pm 0.015$ | $0.805 \pm 0.014$ | $0.833 \pm 0.008$ | $\mathbf{0.918 \pm 0.026}$ | $0.677 \pm 0.032$ | $0.688 \pm 0.015$ | $0.708 \pm 0.021$ |
| $W_2$ | $0.7623 \pm 0.0047$ | $0.7623 \pm 0.0030$ | $\mathbf{0.7608 \pm 0.0050}$ | $0.7665 \pm 0.0077$ | $0.7886 \pm 0.0048$ | $0.8329 \pm 0.0049$ | $0.8398 \pm 0.0059$ | $0.8414 \pm 0.0058$ |

Table 11: Ablation over $\alpha$. $\alpha = 1$ pertains to only using the score model. $\alpha = 0$ pertains to only using flow model.

| $\alpha$ | 0.1 | 0.2 | 0.5 | 0.8 |
|---|---|---|---|---|
| AUROC | $\mathbf{0.971 \pm 0.008}$ | $0.965 \pm 0.007$ | $\mathbf{0.971 \pm 0.009}$ | $0.937 \pm 0.005$ |
| AP | $0.917 \pm 0.025$ | $0.886 \pm 0.026$ | $\mathbf{0.918 \pm 0.040}$ | $0.851 \pm 0.011$ |
| $W_2$ | $0.7886 \pm 0.0048$ | $\mathbf{0.7814 \pm 0.0007}$ | $0.7887 \pm 0.0035$ | $0.8085 \pm 0.0042$ |

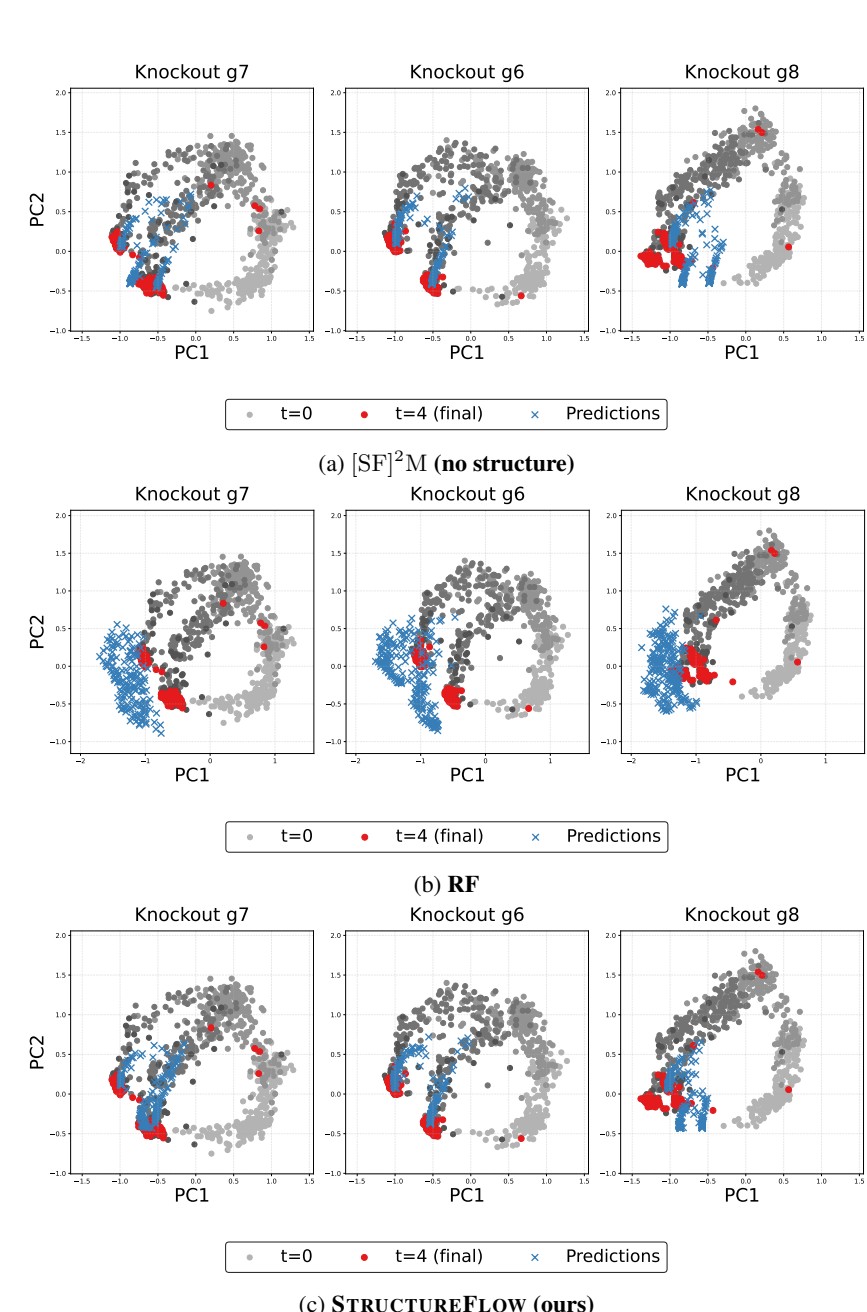

Figure 12: **Comparison of trajectory inference performance for left-out interventions (knockouts).** We show PCA visualizations of model predictions for interventions that were excluded during training. Ground truth data is shown in grey, with model predictions overlaid. This demonstrates each method's ability to generalize to unseen perturbational conditions.

Table 12: Ablation over size of hidden dimension (# of neurons) per layer for the NGM flow model.

| Hidden dimension for NGM | 10 | 50 | 100 | 200 | 256 |
|---|---|---|---|---|---|
| AUROC | $0.925 \pm 0.010$ | $0.964 \pm 0.004$ | $0.971 \pm 0.008$ | $0.969 \pm 0.003$ | $\mathbf{0.975 \pm 0.003}$ |
| AP | $0.832 \pm 0.020$ | $0.910 \pm 0.013$ | $0.917 \pm 0.025$ | $0.916 \pm 0.014$ | $\mathbf{0.932 \pm 0.012}$ |
| $W_2$ | $0.7950 \pm 0.0024$ | $\mathbf{0.7853 \pm 0.0026}$ | $0.7886 \pm 0.0048$ | $0.7972 \pm 0.0055$ | $0.7858 \pm 0.0208$ |

### E.7 ASSESSING THE STABILITY OF THE SCORE AND FLOW NETWORK IN STRUCTUREFLOW

Here we test two settings where we try freezing the weights of one of the modules. We jointly train $s_\phi$ and $v_\theta$ for $K \in \{10000, 15000\}$ iterations, then freeze the parameters of $s_\phi$ and continue training only $v_\theta$ for an additional $M = 5000$ iterations. We do the same experiment but freeze the flow term instead. In both settings our performance stays constant for structure learning, but is more sensitive for dynamics.

Table 13: Ablation over freezing flow and score model post-training when models pre-trained for 10000 iterations.

| Freeze model | AP | AUROC | $W_2$ |
|---|---|---|---|
| score | 0.9011 | 0.9681 | $1.5462 \pm 0.0520$ |
| flow | 0.9109 | 0.9700 | $0.9366 \pm 0.1992$ |

Table 14: Ablation over freezing flow and score model post-training when models pre-trained for 15000 iterations.

| Freeze model | AP | AUROC | $W_2$ |
|---|---|---|---|
| score | 0.9019 | 0.9668 | $1.5200 \pm 0.0569$ |
| flow | 0.8843 | 0.9650 | $0.9337 \pm 0.2181$ |

### E.8 SCALING AND RUNTIME OF VARIOUS BASELINES AND OUR METHOD ON RENGE DATA

Table 15: Model wall-clock time on the Real (Renge) dataset. We include parameter counts for all neural network-based methods. We note that for STRUCTUREFLOW and $[SF]^2M$ the OT pairing was negligble and thus not included.

| Time [s] | Model | Parameter Size |
|---|---|---|
| $319.31 \pm 21.29$ | STRUCTUREFLOW | 1440646 |
| $1542.88 \pm 26.67$ | TIGON (PCA) | 43140 |
| $14112.49$ | TIGON (Ambient) | 409448 |
| $338.12 \pm 0.78$ | RF | - |
| $101.31 \pm 1.43$ | $[SF]^2M$ | 175310 |
| $12.21 \pm .125$ | OTVelo | - |

### E.9 STRUCTURE EXPERIMENTS FOR REAL DATA

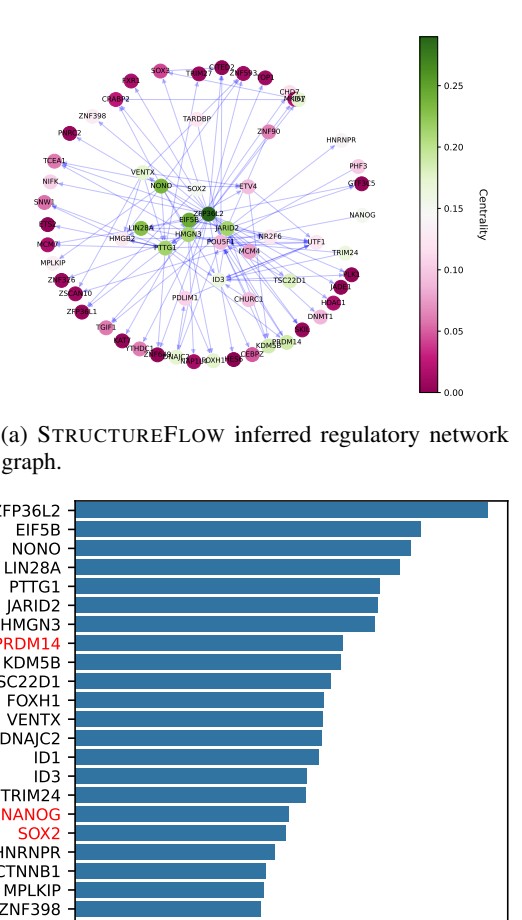

(a) STRUCTUREFLOW inferred regulatory network graph.

(b) Out-edge eigencentrality scores for the STRUCTUREFLOW-inferred network.

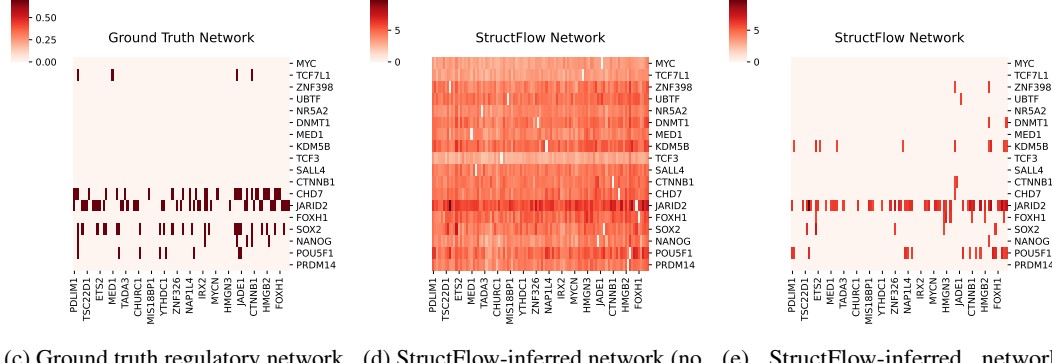

(c) Ground truth regulatory network (heatmap).

(d) StructFlow-inferred network (no threshold).

(e) StructFlow-inferred network (99th quantile threshold).

Figure 13: **Analysis of gene regulatory network inference results. (a)** STRUCTUREFLOW graph representation of the inferred gene regulatory network. **(b)** The top 25 Centrality scores (out-edge eigencentrality) for nodes in the STRUCTUREFLOW-inferred network. **(c)** Ground truth regulatory network shown as a heatmap. This network is from the experimental binding information from the ChIP-atlas. **(d)** STRUCTUREFLOW-inferred network adjacency matrix (no thresholding). **(e)** STRUCTUREFLOW-inferred network thresholded at the 99th quantile to highlight the strongest edges.

## E.10 Evaluating StructureFlow with Irregularly Sampled Time-Points

We evaluate STRUCTUREFLOW and Reference Fitting on the TF BoolODE system using population snapshots from non-uniform timepoints. To accomplish this, we bin the continuous timepoint data into randomly non-uniformly spaced time bins with a standard deviation of 30% of the time bin width (whereas our original experiments bin into discrete equally spaced timebins of $t = 1, 2, 3, \ldots$). We run this experiment over 3 seeds, where the seeding randomizes the time bin spacing (making the data non-deterministic between runs, and thus RF is non-deterministic in this specific setting).

Table 16: Comparison of STRUCTUREFLOW and RF on irregularly sampled time-points for the TF system.

| Metric | STRUCTUREFLOW | RF |
|---|---|---|
| GRN Graph AP | $0.8501 \pm 0.0348$ | $0.9614 \pm 0.0282$ |
| GRN Graph AUROC | $0.9326 \pm 0.0180$ | $0.9812 \pm 0.0099$ |
| Trajectory $W_2$ | $0.8491 \pm 0.0168$ | $1.4060 \pm 0.0665$ |
| Trajectory MMD | $0.1408 \pm 0.0022$ | $0.1977 \pm 0.0152$ |

## E.11 Comparison with Symbolic Distribution Flow Learning (SDFL)

We compare STRUCTUREFLOW to Symbolic Distribution Flow Learning (SDFL) (Dakhmouche et al., 2025), which tackles the population dynamics problem through learning explicit symbolic equations. While SDFL provides interpretable outputs through per-dimension equations, it requires significant computation time to run and cannot explicitly recover a learned matrix structure for the structure learning task. We evaluate SDFL on the trifurcating (TF) system for the trajectory inference task and report results below (we include STRUCTUREFLOW results from Table 1 for easy comparison). Note due to the methods training instability, we were unable to get 3 consecutive successful seeds, and therefore report only 1 seed result.

Table 17: Comparison of dynamical inference performance with SDFL on the TF system.

| Method | $W_2\downarrow$ | MMD$\downarrow$ |
|---|---|---|
| SDFL | 3.4880 | 0.0785 |
| STRUCTUREFLOW | $0.789 \pm 0.012$ | $0.135 \pm 0.004$ |

## APPENDIX REFERENCES

Huynh-Thu, V. A., Irrthum, A., Wehenkel, L., and Geurts, P. (2010). Inferring regulatory networks from expression data using tree-based methods. *PLoS ONE*, 5(9):e12776.

Matsumoto, H., Kiryu, H., Furusawa, C., Ko, M. S. H., Ko, S. B. H., Gouda, N., Hayashi, T., and Nikaido, I. (2017). Scode: an efficient regulatory network inference algorithm from single-cell rna-seq during differentiation. *Bioinformatics*, 33(15):2314–2321.

Zou, Z., Ohta, T., and Oki, S. (2024). ChIP-Atlas 3.0: a data-mining suite to explore chromosome architecture together with large-scale regulome data. *Nucleic Acids Research*, page gkae358.

