# OpenReview forum: "Simulation-Free Structure Learning for Stochastic Dynamics"
_ICLR.cc/2026/Conference — Submitted to ICLR 2026_

### Official Review · Reviewer_6hGC · 2025-10-20

**Soundness:** 3
**Presentation:** 3
**Contribution:** 3
**Rating:** 8
**Confidence:** 3

**Summary:**

The paper proposes STRUCTUREFLOW, a simulation-free framework to jointly learn (i) the network structure of a high-dimensional stochastic dynamical system and (ii) its conditional population dynamics from snapshot data (observational and CRISPR-style interventions). Technically, the method formulates the joint task as a multi-marginal Schrödinger Bridge and trains a pair of neural fields with [SF]²M. Interventions are modeled as ideal knockouts by masking outgoing influences in the graph layer. Empirically, on (a) linear SDE synthetic systems up to d=500, (b) six BoolODE “biological” systems, and (c) a real single-cell CRISPR time-series, the method shows higher AP/AUROC for structure recovery than baselines such as RF, SCODE, dynGENIE3, OTVelo, TIGON, and competitive to [SF]²M/TIGON for dynamic inference; on RENGE left-out timepoints, average W2 5.634 vs 9.718 (RF); on left-out interventions, average W2 5.683 (best among joint methods). Training uses Sinkhorn couplings between adjacent snapshots and Group Lasso for sparsity. Results are reported typically over five seeds.

**Strengths:**

1. This work formulates joint structure+dynamics learning as a multi-marginal SB with [SF]²M, avoiding inner-loop simulation.
2. On BoolODE systems, AP/AUROC for structure recovery are consistently top-tier across six systems.
3. Method assumptions and architecture are well delineated. Besides, results are separated for tasks that baselines can vs cannot handle.
4. Simulation-free training enables scaling as shown in Fig. 2, and the intervention generalization is practically valuable in biology.

**Weaknesses:**

1. The introduction could come with more references in the motivation stage.
2. A line of work is missing in the related works: relational inference / structural inference. These works also work on structure learning and have much to do with the single-cell data.
3. This work comes with strong assumptions. An ideal knockout (zeroing all outgoing edges) may not reflect realistic CRISPR effects (partial/pleiotropic impacts). I would like to suggest testing robustness to imperfect masks and off-target edges.
4. Moreover, the stationary structure between marginals is strong. Please consider adding experiments with time-varying graphs or change-point detection to assess failure modes.

**Questions:**

1. Provide an experiment where you freeze $s_\phi$ after pretraining (or fix $\sigma$) and show that the learned structure in $v_\theta$ remains stable. Conversely, clamp sparsity and let $s_\phi$ adapt, and check how dynamics/structure metrics trade off?
2. It would be good to see more ablations. Please consider sweeping $\alpha$ (flow vs score loss), Group-Lasso $\lambda$, graph hidden width, and knockout mask schedule. If possible, please include CIs over ≥3 seeds.

---

> ### Author Response · Authors · 2025-11-24
> **(1/2)**
>
> We thank the reviewer for their detailed review and constructive feedback. We are delighted to see the reviewer recognizes our work as “practically valuable”, that our method “assumptions and architecture are well delineated”, and that our approach “consistently achieves top-tier” performance across systems. We are also grateful for the reviewer’s valuable suggestions which have helped improve the overall quality of our work. Below, we provided responses to specific questions and comments posed by the reviewer.
>
> > The introduction could come with more references in the motivation stage.
>
> We appreciate the reviewer’s input and have updated the manuscript with additional references in the introduction/motivation. We have additionally added other relevant citations throughout the manuscript.
>
> > A line of work is missing in the related works: relational inference / structural inference. These works also work on structure learning and have much to do with the single-cell data.
>
> We thank the reviewer for highlighting this important line of work, though we note that NRIs rely on trajectory data, whereas StructureFlow considered the setting of population snapshot data where particles (cells) are unpaired between timepoints (Kipf et al., 2018). We have updated the manuscript to mention this related line of work in section 2.1
>
> > This work comes with strong assumptions. An ideal knockout (zeroing all outgoing edges) may not reflect realistic CRISPR effects (partial/pleiotropic impacts). I would like to suggest testing robustness to imperfect masks and off-target edges.
>
> We thank the reviewer for this valuable point. Although we acknowledge that modeling imperfect interventions/knockouts is an important direction, this setting was not the focus of this work. To study this, we need to create (or adjust existing) simulators for biologically meaningful systems to incorporate imperfect knockouts. Since the simulators themselves do not incorporate imperfect knockouts and realistic CRISPR effects, it is difficult to evaluate and study this phenomenon. We believe this is a fruitful direction for future research, and have added a brief discussion of this point to limitations and future work in our updated manuscript.
>
> > Moreover, the stationary structure between marginals is strong. Please consider adding experiments with time-varying graphs …
>
> Thank you for raising this point! In a similar vein to the previous comment, we believe this an interesting and important direction for future research. The problem of learning the network structure of systems, even with the assumption of stationary structure, is challenging and an active area of research. Moreover, jointly learning structure alongside the conditional population dynamics of these systems is an unaddressed problem. Studying structure learning in settings where the underlying network is time-varying, while also in the setting of stochastic population dynamics, requires simulators created to emulate this data generative process. From here, we can develop methods which are tailored to learn time-dependent networks. Our work, StructureFlow, lays out the initial groundwork necessary to further adapt and study the joint inference problem in settings beyond those considered in this work, such as structure learning of non-stationary networks. We believe both curating biologically meaningful simulators with non-stationary network structure for stochastic population dynamics and adapting StructureFlow to time-dependent networks are fruitful directions for future work. We have added discussion of this point in limitations and future work in the revised manuscript.

---

> ### Author Response · Authors · 2025-11-24
> **(2/2)**
>
> > Provide an experiment where you freeze s_\phi after pretraining (or fix \sigma) and show that the learned structure in v_\theta remains stable. Conversely, clamp sparsity and let s_\phi  adapt, and check how dynamics/structure metrics trade off?
>
> Thank you for suggesting this additional ablation. We show results for this ablation in the tables below, and have included them in the revised manuscript (tables 12 and 13, appendix E.7).
>
> We first note that we normally train $s_\phi$ and $v_\theta$ jointly. This coupling ensures that $s_\phi$’s weights update according to the noise component during training, allowing $v_\theta$ to learn the drift and thus recover an accurate structure prediction via NGM, rather than $s_\phi$ serving as a fixed pre-learning component.
>
> Here we test two settings where we try freezing the weights of one of the modules. We jointly train $s_\phi$ and $v_\theta$ for $K \in {10,000, 15000}$ iterations, then freeze the parameters of $s_\phi$ and continue training only v_theta for an additional M = 5000 iterations. We do the same experiment but freeze the flow term instead. In both settings our performance stays constant for structure learning, but is more sensitive for dynamics. This is an unsurprising observation since the score is used to model time-dependent response while the flow is time-independent.
>
> For $K=10000$
> |Freeze model      |AP   |AUROC |$W_2$   |
> |---               |---  |---   |---  |
> |score             |.9011|.9681 |1.5462 +/- 0.0520|
> |flow              |.9109|.9700 |0.9366 +/- 0.1992 |
>
> For $K=15000$
> |Freeze model      |AP   |AUROC |$W_2$   |
> |---               |---  |---   |---  |
> |score             |.9019|.9668 |1.5200 +/- 0.0569|
> |flow              |.8843|.9650| 0.9337 +/- 0.2181 |
>
> > Please consider sweeping \alpha (flow vs score loss), Group-Lasso \lambda, graph hidden width, and knockout mask schedule.
>
> Great point! We have updated the manuscript to include three additional ablation experiments, where we vary the group lasso, graph hidden width (the number of neurons), and \alpha term (see tables below, and appendix E section 6, tables 9,10, and 11). We note that the structure is relatively constant across changes in $\alpha$, but gets worse as the group lasso term grows above 0.1, which makes sense given that we are oversparsifying the system. Additionally, the width of the NGM network plays a role in structure recovery as larger networks, up to 256, tend to outperform smaller ones. Across all three ablations, trajectory inference performance is quite stable, with a slight decrease in performance when oversparsifying the system.
>
> | Group Lasso Reg. | 0.0 | 0.0001 | 0.001 | 0.01 | 0.04 | 0.1| 0.2| 0.5 |
> |---|  ---   |  ---      | ---      |  ---    |  ---    | ---    |  ---   | ---    |
> |AUROC|0.943±0.004|0.946±0.004|0.944±0.004|0.952±0.002|0.971±0.008|0.853±0.032|0.851±0.017|0.879±0.016|
> |AP|0.804±0.014|0.816±0.015|0.805±0.014|0.833±0.008|0.918±0.026|0.677±0.032|0.688±0.015|0.708±0.021|
> |$W_2$|0.7623±0.0047|0.7623±0.0030|0.7608±0.0050|0.7665±0.0077|0.7886±0.0048|0.8329±0.0049|0.8398±0.0059|0.8414±0.0058|
>
> | $\alpha$| 0.1           | 0.2       | 0.5           | 0.8       |
> |---               |  ---          | ---       | ---           |---        |
> |AUROC             |**0.971±0.008**|0.965±0.007|**0.971±0.009**|0.937±0.005|
> |AP                |0.917±0.025    |0.886±0.026|**0.918±0.040**|0.851±0.011|
> |$W_2$                |0.7886±0.0048 |0.7814±0.0007 |0.7887±0.0035|0.8085±0.0042|
>
> |Hidden dim (# neurons)|10         |50         |100        |200        |256            |
> |---                  |---        |---        |---        |---        |---            |
> |AUROC                |0.925±0.010|0.964±0.004|0.971±0.008|0.969±0.003|**0.975±0.003**|
> |AP                   |0.832±0.020|0.910±0.013|0.917±0.025|0.916±0.014|**0.932±0.012**|
> |$W_2$                   |0.7950±0.0024 | 0.7853±0.0026|0.7886±0.0048|0.7972±0.0055|0.7858±0.0208|
>
> We again thank the reviewer for their insightful feedback and valuable comments. Through this rebuttal, we believe we have improved the overall quality of our work, and hope we have addressed all of the reviewer’s comments and questions. If the reviewer is satisfied with our response, we hope the reviewer may consider improving their rating of our paper. We are more than happy to engage in future discussions and answer any salient questions that may arise.

---

> > ### Comment · Reviewer_6hGC · 2025-11-26
> >
> > Dear Authors,
> >
> > I would like to thank you for the rebuttal. The concerns from my review are addressed. But I will pay close attention to the response from other reviewers as some of the points are convincing. For example, weakness 1 and 3 from Reviewer MmjV, scalability issues raised by Reviewer HiCq and Reviewer CZaQ.

---

> > > ### Author Response · Authors · 2025-11-27
> > >
> > > We thank the reviewer for their continued engagement and for confirming that we have addressed their concerns.
> > >
> > > Regarding the points raised by other reviewers that you are monitoring, we would like to briefly highlight the specific evidence we have provided in our detailed responses to them:
> > >
> > > **On Scalability**: We have demonstrated empirically that StructureFlow scales favourably with system dimension (which was already included in the original submission). For instance, **in Figure 2 section 5.1, where we evaluate structure learning performance and computational cost of StructureFlow as we increase the dimensionality of the system**. In this experiment we compare scaling performance to RF (linear method), and NGM-NODE (NeuralODE method, which corresponds to TIGON which also uses a NeuralODE solver).
> > >
> > > We additionally include a computational cost table on the TF BoolODE system (Table 8, Appendix E.5 in the revised manuscript, also in response to reviewer [CZaq]), as well as and additional computational cost Table on the real biological in further discussion with reviewer [HiCq].
> > >
> > > Amongst all these results, we establish that StructureFlow exhibits the most favorable scaling amongst joint structure learning and dynamical inference in both recovering the underlying network structure and computational cost and in recovering the underlying network structure. OTVelo and RF give comparative computational cost, but both methods fall short on yielding competitive performance on the joint task.
> > >
> > > **On Principled Justification (MmjV weakness 1)**. Justification for learning the underlying network structure via the NGM stems from **the theoretical framework provided by Neural Graphical Model (NGM) work** (Bellot et al, ICLR, 2022). The authors provide a rigorous guarantee in Lemma 4 of their paper that the first weight matrix learns the network structure. The authors demonstrate that, when using group lasso solely at the first-layer weights of the NGM trained to learn continuous dynamics, adaptive group lasso recovers the correct sparsity pattern. We provide further details and a brief extension of this theory for our setting in response to the reviewer [CZaQ] (response 2/5).
> > >
> > >
> > > **On Claims of Causal Dependence (MmjV weakness 3)**. We provide a detailed response to MmjV for this point, but we wish to re-iterate here: **we do not make any overstated claims regarding learning causal dependencies**. In fact, on line 476-480 (of revised manuscript) we explicitly state: “We remark that although StructureFlow shows improved generalization performance on the left-out intervention task, implying some plausibly improved learning of underlying mechanisms, we make no claim that StructureFlow truly models the causal dependencies of the underlying data generative process.”
> > >
> > > We hope this summary helps clarify your assessment of our work, and are happy to answer any additional questions that may arise and continue engaging in discussion!

---

### Official Review · Reviewer_CZaQ · 2025-10-26

**Soundness:** 2
**Presentation:** 3
**Contribution:** 1
**Rating:** 2
**Confidence:** 4

**Summary:**

The paper introduces STRUCTUREFLOW, a simulation-free framework that jointly performs dynamical inference and structure learning for stochastic systems from snapshot data (including interventional conditions). The method parameterizes an autonomous neural graphical vector field (NGM) for structure and a time-dependent score; training adopts [SF]^2M losses with entropic OT pairings between adjacent timepoints, and structure is read off via group-lasso on the first NGM layer. Experiments span synthetic systems and a single-cell CRISPR dataset.

**Strengths:**

- Clear, modular parameterization: autonomous NGM for stationary structure + time-varying score; neat and practically convenient.
- Includes interventional modeling via explicit knockout masks; joint training is end-to-end.

**Weaknesses:**

- The core building blocks—[SF]^2M training and NGM structure parameterization—are established; the main additions seem to be the autonomous/score split and intervention masks. This feels incremental.
- The graph is learned purely from data; no biological priors or constraints are incorporated. This limited the identifiability and interpretability.
- Although the paper refers to its formulation as a “multi-marginal Schrödinger Bridge problem” (Intro), the derivations and training procedure operate only on adjacent timepoint couplings.  There is no global multi-marginal coupling or joint entropy minimization across all timepoints, as in formal multi-marginal SB theory (e.g., Chen et al., NeurIPS 2023, Theodoropoulos et al., NeurIPS 2025).  This may suggest a conceptual misunderstanding: the proposed approach is not a true multi-marginal SB formulation in my view.
- Structure is read from the NGM’s first layer via group-lasso norms, but the paper provides no identifiability/consistency guarantees for recovering directed structure. Please formalize the conditions under which edges are recoverable, or position results as a heuristic.
- The model assumes isotropic diffusion and explicitly notes it does not consider unbalanced settings (birth/death/mass change), which are central in single-cell population dynamics. This limits biological fidelity, especially relative unbalanced style approaches that model growth/death.
- Training repeatedly computes Sinkhorn EOT couplings between snapshots; the paper highlights “simulation-free” benefits but does not report wall-clock, memory, or scaling curves in sample size and dimension. Please add complexity analysis and runtime/memory tables for EOT and NGM components.
- The manuscript lists RF, OTVelo, TIGON, and [SF]^2M as baselines, but provides fewer implementation details. Without such detail, it is difficult to assess the fairness of the comparison. Moreover, while the paper emphasizes “simulation-free efficiency,” no runtime or memory analysis is presented relative to baseline methods. Including detailed baseline descriptions and efficiency metrics would strengthen empirical validity.
- I do not find the LLM usage statement, which is required according to the ICLR submission guidelines.

**Questions:**

1.	**Clarification of the “multi-marginal Schrödinger Bridge” formulation**
Could you clearly define your optimization problem? Does it jointly couple all timepoint distributions in a single entropy-regularized objective (as in Chen et al., NeurIPS 2023; Theodoropoulos et al., NeurIPS 2025)?
If not, please justify the terminology “multi-marginal” and explain how your pairwise couplings maintain temporal consistency across all marginals.
2.	**Identifiability and structure recovery**
Under what mathematical or statistical conditions can your readout from the NGM reliably recover directed edges?
3.	**Fairness and transparency of baseline comparisons**
Were the baselines (RF, OTVelo, TIGON, [SF]^2M) retrained in your pipeline using identical preprocessing, data splits, and comparable hyperparameter search budgets (e.g., Optuna sweeps)? Could the authors provide more details on the implementation of the baseline?
4.	**Computational efficiency and scalability**
How does STRUCTUREFLOW’s runtime and GPU memory footprint scale with the number of samples and timepoints compared to other methods?
A wall-clock or asymptotic scaling analysis could support your “simulation-free efficiency” claim.
5.	**Extension to unbalanced dynamics**
Since many biological systems exhibit cell proliferation and death, could your framework be extended to an unbalanced Schrödinger Bridge?
6.	**Use of biological priors** Have you explored incorporating known biological priors to guide sparsity or edge weighting?
If not, do you expect such priors would improve identifiability or stability in the real dataset experiments?
7.	**LLM usage disclosure** Did the team use large language models in code generation, writing, or data preparation? If yes, please include this information as required by the ICLR.


I hope I have not misunderstood some aspects of the work. I am open to revising my score should the authors provide a convincing and well-supported rebuttal.

References
- Tong et al., Simulation-Free Schrödinger Bridges via Score and Flow Matching, AISTATS 2024.
- Chen, T., Liu, G.-H., Tao, M., & Theodorou, E. A. (2023). Deep Momentum Multi-Marginal Schrödinger Bridge. NeurIPS 2023.
- Theodoropoulos, P., Saravanos, A. D., Theodorou, E. A., & Liu, G. H. (2025). Momentum Multi-Marginal Schr\" odinger Bridge Matching. NeurIPS 2025

The reviewer wrote this report independently, with an LLM used only for improving clarity.

---

> ### Author Response · Authors · 2025-11-24
> **(1/5)**
>
> We thank the reviewer for their detailed review of our work, valuable comments, and their insightful questions which have helped improve the overall quality of our paper. We appreciate the reviewer emphasizing the strengths of our work, such as our “modular parameterization”, stating it is “neat and practically convenient,” the usefulness of the joint end-to-end training, and the use of interventional data/masking to model conditional dynamics and learn network structure. Below we respond to the reviewers' specific comments and questions.
>
> > The core building blocks—[SF]^2M training and NGM structure parameterization—are established; the main additions seem to be the autonomous/score split and intervention masks.
>
> The reviewer is suggesting that core building blocks of StructureFlow are already established by NGM-SF2M [4], however, we clearly establish (from our experiments and results, Figure 3 section 5.2) that **NGM-SF2M is inadequate in learning the underlying structure** from interventional population data. In fact, through our experiments, we found **NGM-SF2M to be consistently one of the weakest performers on the structure learning task**. This is unsurprising given NGM-SF2M does not incorporate interventions, does not model conditional dynamics, and was only evaluated in idealized settings. This motivated the need for a better solution. We elaborate below (and for further reference, we point to our response to a similar question to reviewer [MmjV]).
>
> StructureFlow builds significantly on this prior work by incorporating interventional conditions designed to encourage proper learning of the underlying structure (causal dependencies) as well as modeling conditional stochastic population dynamics (NGM-SF2M does not incorporate interventions/conditions), modeling time-dependent dynamics (NGM-SF2M is time-independent), and encouraging sparse structure via group lasso and L1 regularization (NGM-SF2M only uses L1 regularization on structure).
>
> Beyond methodological contributions, our work also adds significantly to the evaluation pipeline of methods akin to NGM-SF2M and beyond. NGM-SF2M was evaluated in only 2 observational (no interventions) synthetic systems (trifurcating and bifurcating) and compared against only trivial correlation-based baselines. Further, the authors of NGM-SF2M considered a data setting of an unrealistic quantity of time-points (> 50 timepoints). This is unrealistic since in practice datasets of this nature (such as the real dataset considered in our work) are severely limited to the number of population snapshots across time (~ 4). Finally, we also demonstrated StructureFlow’s ability to scale to systems of high dimensionality (not done by NGM-SF2M).
>
> To this end, our work contributes both methodologically and in establishing a comprehensive empirical suite of experiments and benchmarks for the problem of joint structure learning and dynamics inference of stochastic dynamics.
>
> **Clarification of the “multi-marginal Schordinger Bridge formulation:**
>
> > Although the paper refers to its formulation as a “multi-marginal Schrödinger Bridge problem” (Intro), the derivations and training procedure operate only on adjacent timepoint couplings. There is no global multi-marginal coupling or joint entropy minimization…
>
> We thank the reviewer for pointing to this distinction. While we do not use a global multi-marginal coupling or jointly compute the couplings across all timepoints simultaneously, we consider the setting of using “multi-timepoint population snapshot” data. In this sense, we label the problem as a multi-marginal Schrödinger Bridge problem. In light of the reviewers comment, we have updated the manuscript to replace “multi-marginal” with “multi-timepoint” to improve clarity in the introduction.
>
> We remark that StructureFlow can easily be extended to the true “multi-marginal SB” setting by swapping out the entropic OT solver (which only uses successive pairs of timepoints) for a global multi-marginal timepoint pairing algorithm without any other modifications. In fact, StructureFlow is directly amenable to multi-marginal flow matching approaches such as [1] for this extension. We believe this is an interesting and fruitful direction for future research, but was out of scope for this work.
>
> We have included an additional (small) paragraph in the revised manuscript in section 3.3 highlighting the difference between our work and multi-marginal SBP works, citing Chen et al. 2023 and other relevant literature, while mentioning extensions to multi-marginal SBP as a direction for future work. We’ve also mentioned these works in Section 2.2 for added context.

---

> ### Author Response · Authors · 2025-11-24
> **(2/5)**
>
> **Identifiability and structure recovery:**
>
> > Structure is read from the NGM’s first layer via group-lasso norms, but the paper provides no identifiability/consistency guarantees for recovering directed structure. Please formalize the conditions under which edges are recoverable, or position results as a heuristic.
>
> In general, the question of identifiability for directed structure is challenging, even in the simplest possible case of linear dynamics (see e.g. [2]) and oftentimes contribute to an entire body of work on their own. This was not within the scope of our work, but nonetheless, we will provide theoretical backing of this nature in this rebuttal. Similarly, the authors of Reference Fitting (RF) were not able to provide an identifiability result but only show well-posedness of their optimization problem.
>
> To our knowledge, identifiability in the joint trajectory inference and structure learning problem is an open problem that is particularly challenging due to the interdependence of the learned structure. We thus leave a general theoretical study of identifiability to future work. In what follows, we reason that the NGM + Group Lasso recovers the true underlying structure in the idealized setting where the drift function and score can be reconstructed exactly.
> **Setup** We consider an autonomous dynamics, where the drift field $v$ is given by a NGM model with parameter $\theta_0$
>
> $$ dX_t = v(X_t; \theta_0) dt + \sigma dB_t.$$
>
> Equivalently, this can be written as a probability flow, with $u_t(x) = v(x) - D \nabla \log p_t$ and $D = \frac{1}{2} \sigma \sigma^\top$:
>
> $$\partial_t p_t = -\nabla \cdot [ p_t(x) u_t(x) ].$$
>
> Consider the case where samples of the flow $u_t(x)$ and score $\nabla \log p_t(x)$ can be accessed exactly, where $(t, x) \sim p_t(x)$.
>
> Writing $\hat{v}_t(x; \theta)$ to be the reconstructed vector field, parameterized as a NGM with parameter $\theta$. The corresponding probability flow is
>
> $$ \hat{u}_t(x) = \hat{v}(x) - D \nabla \log p_t. $$
>
> The flow matching objective with target $u_t(x)$ is thus
>
> $$
> \min\_\theta \mathbb{E}\_{t \sim [0, 1], x \sim p\_t} \Vert \hat{u}\_t(x) - u\_t(x) \Vert\_2^2 \Longrightarrow \min\_\theta \mathbb{E}\_{t} \mathbb{E}\_{x \sim p\_t} \Vert \hat{v}(x; \theta) - v(x; \theta\_0)\Vert^2
> $$
>
> In this idealized setting, flow matching amounts to least-square regression on the target vector field $v(x, \theta_0)$.
>
> Let $\{ x_i \}_{i = 1}^N$ a sample of size $N$ drawn following $t \sim [0, 1]$ and $x \sim p_t$. Then, consider the empirical and population risks $R_N, R$
>
> $$ R_N(\theta) = \min_\theta \frac{1}{N} \sum_{i = 1}^N \| \hat{v}(x_i; \theta) - v(x; \theta_0) \|^2, \tag{1}$$
>
> $$R(\theta) = \mathbb{E}_{t, x} \| \hat{v}(x; \theta) - v(x; \theta_0) \|^2. \tag{2} $$
>
> Our goal is to characterize identifiability of the structural graph from samples. Then this falls into the framework considered by Dinh and Ho [3]. Specifically, the results of [3, Theorem 3.6] guarantees local consistency of the learned graph from minimizing (1) in the limit where $n \to \infty$. We would be happy to accept any suggestion for how to approach this analysis.
>
> **Fairness and transparency of baseline comparisons:**
>
> > The manuscript lists RF, OTVelo, TIGON, and [SF]^2M as baselines, but provides fewer implementation details. Without such detail, it is difficult to assess the fairness of the comparison.
>
> Thank you for pointing this out! Appendix B2 contains descriptions and some implementation details. We have updated the manuscript to include more detailed implementation details for all of the models. In order to be fair, we train all models using the same preprocessing steps, train-test splits, and evaluation metrics. We follow the instructions and tutorials provided by each of the authors per baseline, and verify that each model can successfully use the multi-marginal data relevant to our setting. Additionally, we experimented with the parameters for the baselines and found little variance from the settings the authors provided for their models. Both RF and OTVelo use the BoolODE dataset, so we left the parameters as the authors intended. In response to reviewer [HiCq], we address TIGON as well. In general, StructureFlow has more parameters that require tuning than most of the other models.

---

> ### Author Response · Authors · 2025-11-24
> **(3/5)**
>
> **Computation efficiency and scalability:**
>
> We first aim to clarify the notions of “scaling” which we discuss in this work. Structure learning is inherently understood as a “search” problem over possible sparsity patterns that describe the data generative process of the system. This search problem scales exponentially in the dimension of the system. We evaluate how well StructureFlow scales and recovers the true structure/network when the dimension of the system increases. Moreover, we also wish to evaluate the training efficiency of conducting the structure learning procedure. We provided this experiment in the original submission in Figure 2 (Section 5.1) and observed StructureFlow exhibits favorable scaling compared to a counterpart simulation-based approach. A linear baseline (RF) is more computationally efficient, but falls short in learning the correct structure in high-dimensions (and is not expressive enough for the dynamical inference task, shown in the proceeding sections).
>
> To further back this claim, we have added an additional table comparing training time of methods on the BoolODE TF system (see below, and appendix E.5, table 8 of the revised manuscript). We would like to note that we report this table on the same GPU across methods, but StructureFlow can be very efficiently trained just on a CPU!
>
> We note that in this table we do not report parameter sizes for OTVelo because it is a non-parametric method followed by a linear regression. We also note that the RF estimator class learns an A matrix that is dependent on the number of genes, so we do not include parameter size here either.
>
> Here we show a simple 10 dimensional system (dyn-TF) and compare runtimes. See our scaling experiment for more information on how our system scales to higher dimensions. I add the time to compute EOT in ( ). We use the same number of timepoints (5), number of dimensions and cells (8 and 8000, respectively), and normalize across device (Quadro RTX 6000). Note that we add OT-Flow matching as a baseline, where we remove regularization terms, the score term, and replace the NGM backbone with a traditional MLP backbone. The training time is roughly ½ of the training time for our method.
>
> |Time [s]      |Model        |Parameter Size         |
> |---           |---     |---                    |
> |263 + (3.3)   |StructureFlow|20716 |
> |1511.18       |TIGON        |43140                      |
> |131.22        |OT-Velo      |           -            |
> |82.01         |RF           |      -                 |
> |3228.61     |NGM-NODE     |8008                  |
> |116.17 + (3.3)|Flow-matching|8008              |
> |208.58 + (3.3) |$\text{[SF]}^2\text{M}$| 14892 |
>
> > … the paper highlights “simulation-free” benefits but does not report wall-clock, memory, or scaling curves in sample size and dimension…
>
> We do report wall-clock time in relationship to dimension (see Figure 2 Section 5.1). We do not report memory as the data regime and model size for which these methods operate in are not sufficiently large to produce a bottleneck and suffice this analysis (everything can be trained on CPUs!).
>
> > Training repeatedly computes Sinkhorn EOT couplings between snapshots.
>
> Our implementation only computes the EOT couplings once before training starts. For large data regimes, mini-batch OT can be computed instead, which is an efficient approach that still models optimal paths at inference [4]. At inference no OT solution is needed. We have clarified this in the revised manuscript at line 242.

---

> ### Author Response · Authors · 2025-11-24
> **(4/5)**
>
> **Extension to unbalanced dynamics:**
>
> > The model assumes isotropic diffusion and does not consider unbalanced settings (birth/death/mass change), which are central in single-cell population dynamics. Could the framework be extended to an unbalanced Schrodinger bridge?
>
> We thank the reviewer for raising these points. Firstly, our approach can be adapted to the anisotropic diffusion setting by modification of the cost. Setting the diffusion coefficient to $D = \frac{1}{2}\sigma \sigma^\top$, the appropriate cost is then $c(x, y) = (x - y)^\top D^{-1} (x - y)$. Although interesting, we leave the investigation of adapting StructureFlow to the anisotropic diffusion setting for future work.
>
> Secondly, we agree that the unbalanced setting is an important direction, but was out of scope for this work. In our empirical experiments, we directly compare to a baseline which uses unbalanced OT (TIGON) and have shown that StructureFlow (which does not consider growth/death processes) out performs the unbalanced baseline in synthetic and real data settings.
>
> We remark that our method can be adapted to the unbalanced setting by incorporating existing advancements which consider birth/death processes for stochastic population dynamics [5,6,7]. We note these works do not consider the joint setting of learning structure and dynamics (only consider the inference of dynamics) from population snapshots. Hence, we believe the unbalanced setting is a fruitful direction for future research, but is out of scope for this work.
>
> We note that in this work, we focused on establishing the groundwork for simulation-free joint structure learning and dynamical inference of stochastic dynamics, and an extensive evaluation pipeline which can be adapted and advanced for future investigations to unbalanced setting, anisotropic diffusion, and more. We have revised the manuscript to highlight this in the limitations and future work.
>
> **Use of biological priors:**
>
> > The graph is learned purely from data; no biological priors or constraints are incorporated … Have you explored the use of known biological priors to guide sparsity or edge weighting? Do you expect such prior would improve identifiability of stability in the real dataset?
>
> We thank the reviewer for raising this idea! We believe this is a great direction for future work to further advance the biological meaningfulness of StructureFlow. We would like to re-iterate that “the graph being learned purely from data” is exactly the point of our method, which we view as a strength. In many instances, we do not have great prior knowledge on the underlying biology. In part, StructureFlow can be viewed as a hypothesis generation which can be used to elucidate plausible gene regulatory network structure given data from a system.
>
> While we do not incorporate priors in this work, this is an easy extension of our method. Typical application of prior is done on the network side (which can be easily added as a constraint on NGM structure), However, we believe the most promising direction for using biological priors in StructureFlow to improve identifiability and interpretability is through incorporating biologically meaningful information via the reference process. For example, [8, 9] are recent approaches which explore Schrödinger bridge problems with a non-zero drift reference process. For example [8] constructs reference processes using inferred velocities. The same method can be used to include biological priors or constraints in StructureFlow! While this is not the scope of our work, it is a very fruitful direction for future work that is easily applicable to our setup. We have included a small revision in the manuscript, section 3.3 line 265 to highlight one way biological priors may be included through the choice of reference process, and added a mention of this in limitations and future work.
>
> **LLM usage disclosure:**
>
> Referencing the author guidelines for LLM usage: “If LLMs played a significant role in research ideation and/or writing to the extent that they could be regarded as a contributor, then authors should describe the precise role of the LLM in a separate section on LLM usage.” **We did not use LLMs in either ideation or writing**, hence we checked the box in the open review submission to reflect this. As such, per the guidelines, we did not include a LLM usage statement in the submission. Nonetheless, we have added a LLM Usage Statement at line 512 (after the reproducibility statement).

---

> ### Author Response · Authors · 2025-11-24
> **(5/5) Conclusion + References**
>
> **Concluding remarks:**
>
> We would like to again thank the reviewer for their very detailed evaluation of our work and valuable feedback. We hope that through this rebuttal, we have addressed all of the reviewers comments and questions, and believe we have improved the overall quality of our paper by incorporating the reviewer's feedback. Through the reviewer’s comments regarding unbalanced OT, biological priors, identifiability, and multi-marginal Schrodinger bridges, we feel the reviewer has identified several valuable directions for future work, extendable from the groundwork laid out in our paper. We view this as a reflection that our work is valuable to the community and lays out a foundation for further methodological development and research on the problem of joint structure learning and dynamical inference. In this light, if the reviewer feels we have addressed all of their questions and concerns, we hope the reviewer will consider raising their score. We are more than happy to engage in further discussion and answer any additional questions that may arise!
>
> [1] Rohbeck, Martin, et al. "Modeling complex system dynamics with flow matching across time and conditions." The Thirteenth International Conference on Learning Representations. 2025.
>
> [2] Wang S, Al-Radhawi MA, Lauffenburger DA, Sontag ED. Recovering biomolecular network dynamics from single-cell omics data requires three time points. NPJ Systems Biology and Applications. 2024 Aug 27;10(1):97.
>
> [3] Dinh VC, Ho LS. Consistent feature selection for analytic deep neural networks. Advances in Neural Information Processing Systems. 2020;33:2420-31.
>
> [4] Tong, A., Fatras, K., Malkin, N., Huguet, G., Zhang, Y., Rector-Brooks, J., Wolf, G., & Bengio, Y. (2024). Improving and generalizing flow-based generative models with minibatch optimal transport (arXiv:2302.00482). arXiv.
>
> [5] Zhang Z, Li T, Zhou P. Learning stochastic dynamics from snapshots through regularized unbalanced optimal transport. arXiv preprint arXiv:2410.00844. 2024 Oct 1.
>
> [6] Zhang S, Maddu S, Qiu X, Chardès V. Inferring stochastic dynamics with growth from cross-sectional data. arXiv preprint arXiv:2505.13197. 2025 May 19.
>
> [7] Wang D, Jiang Y, Zhang Z, Gu X, Zhou P, Sun J. Joint Velocity-Growth Flow Matching for Single-Cell Dynamics Modeling. arXiv preprint arXiv:2505.13413. 2025 May 19.
>
> [8] Petrović, K., Atanackovic, L., Moro, V., Kapuśniak, K., Ceylan, İ. İ., Bronstein, M., Bose, A. J., & Tong, A. (2025). Curly Flow Matching for Learning Non-gradient Field Dynamics (arXiv:2510.26645). arXiv.
>
> [9] Zhang, S. Y., & Stumpf, M. P. H. (2025). Learning non-equilibrium diffusions with Schrödinger bridges: from exactly solvable to simulation-free (arXiv:2505.16644). arXiv.

---

> > ### Comment · Reviewer_CZaQ · 2025-11-27
> >
> > Thank you for the detailed response. The manuscript has clearly improved: several imprecise statements have been corrected, and the discussion of limitations and future directions is now more thorough. I am willing to raise my score accordingly. Although in my view, the technical novelty remains relatively limited, I believe the work still has practical value. I also appreciate the authors’ thoughts on the potential future extensions I suggested; incorporating some of these elements would further strengthen the manuscript.
> >
> > I am also curious about how the authors trained TIGON. Was the method re-implemented in this work? TIGON is known to exhibit certain training instabilities, so an explanation of how the authors handled these issues would be informative. Moreover, while the quantitative metrics are useful, it would greatly improve interpretability if the authors could provide visual examples of the structures learned by different methods, e.g., TIGON, OT-velo, and NGM-SF2M, which would help readers gain a more intuitive understanding of how these approaches behave on this task and where their differences lie.

---

> ### Author Response · Authors · 2025-12-01
> **Authors' Response (1/2)**
>
> We thank the reviewer for their continuing engagement and insightful comments. We are delighted to see that the reviewer believes our “manuscript has _clearly_ improved” and that the reviewer is willing to raise their score. During the discussion, the reviewer raised additional comments regarding how we can further improve the quality of our work. We address these comments below, and include additional experiments, as suggested by the reviewer, to further increase the novelty and contributions of our work.
>
> > I also appreciate the authors’ thoughts on the potential future extensions I suggested; incorporating some of these elements would further strengthen the manuscript.
>
> We are grateful to see that the reviewer "appreciated" our thoughts on the numerous amenable future extensions of our work. We have now added two additional contributions (unbalanced OT and identifiability theory) to incorporate additional elements to our work (as suggested by the reviewer) and further increase the novelty of StructureFlow and strengthen the manuscript.
>
> **Unbalanced StructureFlow**.
> We extend StructureFlow to the unbalanced setting and consider a bifurcating system (akin to the BoolODE system in the balanced setting) with growth. To do so, we adopt an Unbalanced Optimal Transport (UOT) formulation by replacing the standard Sinkhorn loss with the Unbalanced Sinkhorn divergence formalized in (Lénaïc et al, 2018; Eyring et al, 2024) in our EOT procedure, and learn growth following the framework of (Wang et al, 2025). To evaluate Unbalanced StructureFlow (USF), we introduce a growth-based bifurcating system. We use the underlying gene regulatory network and boolean logic of the standard BoolODE bifurcation dataset, but we impose lineage specific proliferation rates where one of the attractors has a high proliferation rate and the other has a near-zero growth rate. We compare Unbalanced StructureFlow to TIGON (a baseline unbalanced method), and the base StructureFlow (which does not model growth). We observe that Unbalanced StructureFlow yields the best performance on both structure learning and trajectory inference. We include results in the below, as well as in Table 2 of the main text (and further details of the unbalanced setup in Appendix B.3)
>
> Table for Structure Learning:
> | Metric | Unbalanced StructureFlow | TIGON |StructureFlow
> | :--- | :--- | :--- | :--- |
> |AP    |0.509 ± 0.025|0.427 ± 0.047|0.3575 ± 0.0065   |
> |AUROC |0.838 ± 0.010|0.540 ± 0.014|0.7346 ± 0.0054|
> |W2    |1.670 ± 0.439|2.355 ± 0.591 |1.672 ± 0.532|
> |MMD   |0.027 ±  0.033  |0.038 ± 0.041|0.029 ± 0.031|
>
> **Principled Theoretical Backing for Identifiability/Consistency**. We have added a theoretical backing for Identifiability/Consistency in appendix A.3. Notably, we extend the theoretical frameworks from [2] for our setting of population snapshot data and using a flow matching loss. We believe this addresses the reviewers comment on adding a principled justification for structure recovery of StructureFlow into the manuscript.
>
> > I am also curious about how the authors trained TIGON.
>
> We train TIGON with no algorithmic modifications from the original code in both PCA space (as the original implementation of TIGON) and in ambient space (consistent with StructureFlow and other baselines). We observed that TIGON’s performance does not vary significantly when training in PCA space versus ambient space, other than longer computation times in the ambient space (due to the NeuralODE solver used during training). We refer you to our appendix for further details on TIGON’s implementation and evaluation.
>
> > while the quantitative metrics are useful, it would greatly improve interpretability if the authors could provide visual examples of the structures learned by different methods.
> We thank the reviewer for pointing this out. We have updated all the heatmap/structure plots to show outputs of all methods/baselines. See Figure 4 in the updated manuscript and the corresponding figures in the appendix. We have also added TIGON and OTVelo to the trajectory inference visualization in 2D PCA space (also in Figure 4 of the updated manuscript). We believe these additions will help readers interpret and understand how these approaches differ and improve the overall quality of our work.
>
> We once again thank the reviewer for their valuable feedback and insightful comments. We believe that through this rebuttal and discussion, we have addressed all of the reviewer’s comments and further increased the novelty of StructureFlow, strengthened the results and experiments, and improved the overall quality of our manuscript.

---

> > ### Author Response · Authors · 2025-12-01
> > **Authors' Response (2/2)**
> >
> > Additional references:
> >
> > Lénaïc Chizat, Gabriel Peyré, Bernhard Schmitzer, François-Xavier Vialard, Unbalanced optimal transport: Dynamic and Kantorovich formulations, Journal of Functional Analysis, Volume 274, Issue 11, 2018, Pages 3090-3123, ISSN 0022-1236,
> >
> > Eyring, L., Klein, D., Uscidda, T., Palla, G., Kilbertus, N., Akata, Z., and Theis, F. (2024). Unbalancedness in neural monge maps improves unpaired domain translation.
> >
> > Wang, D., Jiang, Y., Zhang, Z., Gu, X., Zhou, P., & Sun, J. (2025). Joint Velocity-Growth Flow Matching for Single-Cell Dynamics Modeling. arXiv preprint arXiv:2505.13413.

---

### Official Review · Reviewer_MmjV · 2025-10-28

**Soundness:** 2
**Presentation:** 3
**Contribution:** 2
**Rating:** 2
**Confidence:** 4

**Summary:**

The paper proposes a new approach to learn stochastic dynamical systems with a network structure. It considers a setting where empirical distributions are observed at given time points, and the goal is to construct a SDE that models the underlying data-generating process. The authors formulate this goal as a multi-marginal Schrödinger Bridge problem with Wiener process as a prior. The modelling is based on the estimation of the vector field defining the SDE by a sparse neural network. Extensive numerical experiments are conducted on synthetic and real-world genomics data.

**Strengths:**

- The paper addresses an import problem in dynamical system learning
- It is clearly written and motivated
- It evaluates the proposed method extensively in various settings

**Weaknesses:**

- It lacks a principled justification for learning the underlying network of the dynamical system (See questions below)
- It would benefit from a more elaborate description of the novelties compared to [SF]2M. Is it a slight modification in the learning objective ?
- Overstated claim on learning causal dependencies

**Questions:**

1/ Could you elaborate on the claim of deducing properties such as bifurcations or attractors from a neural dynamical system such as the one learned here ?

Neural dynamical systems significantly contrast with learning closed-form dynamical systems like in [1, 2, 3] and analysing qualitative properties such as bifurcations and attractors for neural dynamical systems is very challenging and largely an open question. The paper might benefit from clarifying this contrast.

2/ Given that the vector field is modelled with a highly non-linear transformation (a neural net), why would you expect the first (sparse) weight matrix to encode the right network structure ?

3/ How sensitive is the learned structure to the regularisation strength in the Group Lasso term?

4/ Is the method stable when the time marginals are unevenly spaced or noisy?


[1] Brunton, Steven L., Joshua L. Proctor, and J. Nathan Kutz. "Discovering governing equations from data by sparse identification of nonlinear dynamical systems." Proceedings of the national academy of sciences 113.15 (2016): 3932-3937.

[2] Dakhmouche, Ramzi, Ivan Lunati, and Hossein Gorji. "Robust Symbolic Regression for Dynamical System Identification." Transactions on Machine Learning Research. 2025.

[3] Sun, Fangzheng, et al. "Symbolic Physics Learner: Discovering governing equations via Monte Carlo tree search." The Eleventh International Conference on Learning Representations. 2022.

**Details Of Ethics Concerns:**

The authors refer to one of the cited papers (in line 481) as their own, which seems to reveal their identity, and hence violate ICLR policy.

---

> ### Author Response · Authors · 2025-11-24
> **(1/3)**
>
> We thank the reviewer for their detailed assessment. We are encouraged to see that the reviewer found our work to “address an important problem in dynamical system learning”, that our paper is “clearly written and motivated”, and that the method is “evaluated extensively across various settings”. We address your specific concerns below, particularly regarding the principled justification for structure learning and the novelty relative to [SF]2M.
>
> > It would benefit from a more elaborate description of the novelties compared to [SF]2M. Is it a slight modification in the learning objective?
>
> The reviewer asks if StructureFlow is merely a "slight modification" of NGM-SF2M [6]. It is not; the differences are fundamental to why StructureFlow works where NGM-SF2M fails (as shown in Figure 3 of our paper, where NGM-SF2M is one of the lowest performers).
>
> **Decomposition of Dynamics**: NGM-SF2M attempts to regress a time-independent Neural Graphical Model (NGM) directly against a time-dependent probability flow target. This is a mismatch. StructureFlow decomposes the probability flow $u_t$ into two components:
>
> An Autonomous Drift $v(x)$, a time-independent component capturing system structure.
>
> A Score Function, which is time-dependent and captures the stochasticity.
>
> This decomposition aligns with biophysical reality (where gene regulation laws are generally time-invariant) and allows us to learn the structure accurately.
>
> **Handling Interventions**: Unlike NGM-SF2M, StructureFlow incorporates a **knockout conditioning mask** at both training and inference. This is critical for disentangling correlation from causation in biological data and allows us to perform interventional trajectory inference.
>
> > why would you expect the first (sparse) weight matrix to encode the right network structure?
>
> This is an important point of clarification. The **Neural Graphical Model (NGM)** theory [7], which provides a rigorous guarantee, not just a heuristic, that the first weight matrix encodes the network structure.
>
> **Function structure**: In a continuous-time system, a directed edge $x_k \rightarrow x_j$ exists if and only if the drift $f_j$ has a **functional dependence** on $x_k$, i.e. $\partial_k f_j \neq 0$.
>
> **Sparsity Constraint**: It was proven in [7] that for a neural network parameterizing $f_j$, any functional dependence on input $x_k$ must propagate through the first layer. Therefore, if the k-th column of the first (sparse) weight matrix is zero, the network output cannot depend on $x_k$.
>
> **Identifiability**: This justifies the application group lasso solely to the first-layer weights, because zeros in that layer correspond exactly to absent edges in the underlying continuous system. [7] shows local consistency: with enough data, adaptive group lasso recovers the correct sparsity pattern.
>
> > Overstated claim on learning causal dependencies
>
> We do not claim that we learn causal dependencies. In fact, on line 457 we explicitly state:
>
> “We remark that although STRUCTUREFLOW shows improved generalization performance on the left-out intervention task, implying some plausibly improved learning of underlying mechanisms, we make no claim that STRUCTUREFLOW truly models the causal dependencies of the underlying data generative process.”
>
> Beyond this, we remark that the term “causality” has no single agreed-upon meaning and is used differently across philosophical, statistical, and modelling literatures. Mechanistic ideas of causal dependency consider two entities causally linked only when connected by an underlying physical mechanism or process, often formalized through structural differential equations or through conserved-quantity exchanges. On the other hand, there are probabilistic, counterfactual, and interventional notions of causation. Although these perspectives are often presented as distinct, they are not mutually exclusive, i.e. mechanisms can be given counterfactual interpretations, and structural causal models (SCMs) can serve as an intermediate representation connecting underlying dynamical laws to statistical associations.
>
> Importantly, our use of directed temporal dependencies does not rely on any particular metaphysical stance about causation; as long as the system can be represented as a Neural Dynamic Structural Model [7], directed edges simply encode how derivatives induce changes over time, with all other dependencies interpreted contemporaneously. This is fully consistent with widely accepted causal interpretations in dynamical systems.

---

> > ### Author Response · Authors · 2025-11-24
> > **(2/3)**
> >
> > > Could you elaborate on the claim of deducing properties such as bifurcations or attractors?
> >
> > We thank the reviewer for this insightful question. We agree that rigorously analyzing qualitative properties (bifurcations, attractors, etc.) of neural dynamical systems remains an open research question. We remark that answering this question explicitly is not the focus of our work. Our discussion of these concepts was intended as high-level motivation for why recovering structure in dynamical systems is valuable, similar to what is done in [8]. We are not claiming that StructureFlow enables full qualitative analysis of learned dynamics and we have revised the updated manuscript to better reflect this (mentioned below).
> >
> > We would like to point out that symbolic regression methods similar to SINDy [3] assume that trajectories are directly observed and typically assume deterministic dynamics. Our setting is markedly different, assuming only population snapshots are available and in the stochastic setting.
> >
> > Our approach does not assume any specific underlying structure; it can learn any (first-order) structure given multi-timepoint population snapshot data that adheres from the respective system. We show through synthetic experiments StructureFlow’s flexibility to learn models of systems (from data) which exhibit trifurcations, bifurcations, and many others. We have expanded the related works (section 4) to cite the suggested references ([1, 2, 3]) while differentiating our setting (line 266) and updated line 56 to clearly state we do not claim to solve the problem of deducing qualitative properties.
> >
> > > How sensitive is the learned structure to the regularisation strength in the Group Lasso term?
> >
> > We thank the reviewer for raising this valuable question. We have provided an additional ablation in response to reviewer [6hGC] and likewise included this table in the updated manuscript (table 9, appendix E.5). We observe some sensitivity to this parameter, but StructureFlow still achieves competitive structure learning and dynamical inference performance across regularization strengths.
> >
> > > Is the method stable when the time marginals are unevenly spaced or noisy?
> >
> > Yes, as the vector field is autonomous (time-independent), unevenly spaced or noisy timepoints simply change the magnitude of the velocity for the flow field during training. While we did not systematically measure this robustness, we do find it to be robust. As an example, on the trajectory inference task we leave out an intermediate timepoint $t$ during training, and couple data at $t-1$ to $t+1$, leading to the unevenly spaced timepoints you asked about (so $\Delta t$ is replaced with $\Delta t_i = t_{i+1} - t_i$i) . In this setting, even with a data point missing, the regulatory network is recovered with almost equal accuracy to that of the full data regime (with the small loss in quality attributable to training on less data). Therefore yes, StructureFlow can be trained on uneven time marginals and tends to remain stable! We leave more detailed exploration into unevenly spaced timepoints for future work.

---

> > > ### Author Response · Authors · 2025-11-24
> > > **(3/3)**
> > >
> > > > The authors refer to one of the cited papers (in line 481) as their own, which seems to reveal their identity, and hence violate ICLR policy.
> > >
> > > The sentence was not meant to imply the cited work belongs to us (and it does not!). This is a grammatical ambiguity. Therefore, we did not de-anonymize ourselves. We thank the reviewer for pointing to this ambiguity in grammatical sentence structure, and we have fixed the sentence in the updated manuscript to reduce confusion.
> > >
> > > **Concluding remarks:**
> > >
> > > We would like to again thank the reviewer for their valuable comments and time in reviewing our work. We are delighted to see the numerous strengths pointed out by the reviewer: the importance of the problem addressed in our work, clarity and presentation of our manuscript, well motivated, and our extensive experimental evaluation.
> > >
> > > We believe that through addressing the valuable comments and questions raised by the reviewer we have improved the overall quality of our manuscript. We hope that if our response addresses all of the reviewer’s comments and questions, the reviewer will consider raising their score. We are more than happy to engage in further discussion and answer any remaining questions the reviewer may have.
> > >
> > > [1] Brunton, Steven L., Joshua L. Proctor, and J. Nathan Kutz. "Discovering governing equations from data by sparse identification of nonlinear dynamical systems." Proceedings of the national academy of sciences 113.15 (2016): 3932-3937.
> > >
> > > [2] Dakhmouche, Ramzi, Ivan Lunati, and Hossein Gorji. "Robust Symbolic Regression for Dynamical System Identification." Transactions on Machine Learning Research. 2025.
> > >
> > > [3] Sun, Fangzheng, et al. "Symbolic Physics Learner: Discovering governing equations via Monte Carlo tree search." The Eleventh International Conference on Learning Representations. 2022.
> > >
> > > [4] Vastola JJ, Holmes WR. Chemical Langevin equation: A path-integral view of Gillespie's derivation. Physical Review E. 2020 Mar;101(3):032417.
> > >
> > > [5] Ventre E, Espinasse T, Bréhier CE, Calvez V, Lepoutre T, Gandrillon O. Reduction of a stochastic model of gene expression: Lagrangian dynamics gives access to basins of attraction as cell types and metastabilty. Journal of Mathematical Biology. 2021 Nov;83(5):59.
> > >
> > > [6] Tong A, Malkin N, Fatras K, Atanackovic L, Zhang Y, Huguet G, Wolf G, Bengio Y. Simulation-free schr\" odinger bridges via score and flow matching. arXiv preprint arXiv:2307.03672. 2023 Jul 7.
> > >
> > > [7] Bellot, A., Branson, K., and van der Schaar, M. (2022). Neural graphical modelling in continuous time: Consistency guarantees and algorithms. International Conference on Learning Representations.
> > >
> > > [8] Qiu X, Zhang Y, Martin-Rufino JD, Weng C, Hosseinzadeh S, Yang D, Pogson AN, Hein MY, Min KH, Wang L, Grody EI. Mapping transcriptomic vector fields of single cells. Cell. 2022 Feb 17;185(4):690-711.

---

> > > > ### Comment · Reviewer_MmjV · 2025-11-27
> > > >
> > > > Thank you for the detailed response. I am willing to raise my score, after couple of further clarifications.
> > > >
> > > >
> > > > [Q1] Since deducing qualitative properties of neural dynamical systems is not possible, it does not make sense to have an unfeasible goal as a motivation. Note that reference [2] actually tackles the population dynamics problem with explicit symbolic fields.
> > > >
> > > > [Q4] The fact that the model is a continuous-time one can in principle handle irregularly spaced screen-shots, however, in practice it can struggle. So, could you evaluate (for e.g. in synthetic settings) the models ability to handle unevenly distributed screenshots at test time ?

---

> > > > > ### Author Response · Authors · 2025-12-01
> > > > > **Authors' Response**
> > > > >
> > > > > Thank you for your continuing engagement! We are happy to see that the reviewer is satisfied with our rebuttal and will raise their score. Below, we provide responses and details to the additional clarifying questions posed by the reviewer.
> > > > >
> > > > > > [Q1] On deducing qualitative properties of dynamics and reference [2].
> > > > >
> > > > > **On deducing qualitative properties of dynamics**. We thank the reviewer for this important point. We are happy to revise this sentence in the manuscript, and reduce this claim. Specifically, we now state that “From a good estimate of the vector fields, qualitative properties of dynamics can be interpreted …”. We have also updated lines 55-58 in the revised manuscript.
> > > > >
> > > > > **On reference [2]**. Thank you for pointing this out! We have revised the related works section (see updated manuscript lines 284) to reflect this. Namely, we note that [2] does solve population dynamics, and provides an interpretable output through per-dimension equations, but takes significant computation time to run (5760.4s runtime). We demonstrate this through an additional experiment on the trifurcating (TF) system for the trajectory inference task, where we observe poor performance. Note that the method cannot explicitly recover a learned matrix structure, and thus we cannot assert its performance on the structure learning task. We include results for this in the Table below as well as in appendix section E.11, table 16. Note due to the methods training instability, we were unable to get 3 consecutive successful seeds, and therefore report only 1 seed result.
> > > > >
> > > > > | Method | W2 | MMD |
> > > > > | :--- | :--- | :--- |
> > > > > | SDFL | 3.4880 | 0.0785 |
> > > > >
> > > > >
> > > > > > [Q4] Evaluating StructureFlow in irregularly sampled time-points.
> > > > >
> > > > > To back this claim, we evaluated StructureFlow on the TF BoolODE system, using population snapshots from non-uniform timepoints. To accomplish this, we bin the continuous timepoint data into randomly non-uniformly spaced time bins with a standard deviation of 30% of the time bin width (whereas our original experiments bin into discrete equally spaced timebins of t = 1, 2, 3, ...). We run this experiment over 3 seeds, and note that the seeding randomizes the time bin spacing (making the data non-deterministic between runs, and thus RF is non-deterministic in this specific setting).
> > > > >
> > > > > We compare to RF and observe that StructureFlow yields comparative performance on the structure learning task, while performing favourably on the dynamical inference task (see table below). We do observe structure learning performance drops slightly compared to the uniform time setting, but we note that, in interest of time, we chose a single reasonable hyperparameter setting for the model, and there is room for further improvement in adapting StructureFlow to this setting. We have also included this experiment in appendix E.10 table 15 in the updated manuscript.
> > > > >
> > > > > | Metric | StructureFlow | RF |
> > > > > | :--- | :--- | :--- |
> > > > > | GRN Graph AP | 0.8501 ± 0.0348 | 0.9614 ± 0.0282 |
> > > > > | GRN Graph AUROC | 0.9326 ± 0.0180 | 0.9812 ± 0.0099 |
> > > > > | Trajectory W2 | 0.8491 ± 0.0168 | 1.4060 ± 0.0665 |
> > > > > | Trajectory MMD | 0.1408 ± 0.0022 | 0.1977 ± 0.0152 |
> > > > >
> > > > > We again thank the reviewer for their insightful questions and valuable feedback. We believe that through this rebuttal we have addressed all of the reviewer’s concerns, and have improved the overall quality of the manuscript.

---

### Official Review · Reviewer_HiCq · 2025-10-29

**Soundness:** 4
**Presentation:** 3
**Contribution:** 3
**Rating:** 4
**Confidence:** 3

**Summary:**

This paper presents StructureFlow, a novel simulation-free flow matching method for performing structure learning and modelling dynamics of dynamical systems simultaneously. The method is nicely designed and is based on interesting ideas: using neural graphical model to capture the system structure and a time-dependent score function to capture evolving stochastic dynamics. The method is clearly presented and comprehensive experimental evaluation shows that StructureFlow achieves its objetives.

**Strengths:**

S1: The paper addresses a very important problem of joint dynamical inference and structure learning.
S2: It introduces an elegant design of StructureFlow, and the novelty is very high.
S3: The paper has a comprehensive evaluation of StructureFlow through synthetic systems, biologically simulated systems, and a real-life single-cell dataset.
S4: The paper is clearly written and easy to follow.

**Weaknesses:**

W1: The scalability of the StructureFlow is not fully addressed.
W2: The evaluation of StructureFlow for structure learning is not fully satisfactory.

Minor comments:
1. Page 3, lines 148-150 have the same information as lines 143-146.
2. Page 4, line 165: I am confused about q(z), which does not appear in Eq. 8 or before.

**Questions:**

Q1: Can you please discuss about the scalability of StructureFlow? I understand simulation-free flow match itself is quite scalable, but I am curious about StructureFlow.
Q2: For the evaluation of structure learning, only limited methods are considered and compared with StructureFlow. There is a benchmark for structural inference of dynamical systems with synthetic data, covering more SOTA methods such as NRI-based ones. It will be interesting to see whether StructureFlow is still competitive with those methods.
Q3: I am curious about the details of how dynGENIES is applied and evaluated on the BoolODE data. Do you use the simulated trajectories directly?
Q4: I am also curious about the details of how TIGON is applied and evaluated. According to previous experience, TIGON does not scale well and needs to perform dimensionality reduction first to reduce the number of variables. It is a bit surprising to me that it works for the real biological dataset to infer a 103*103 network.

---

> ### Author Response · Authors · 2025-11-24
> **(1/2)**
>
> We’d like to thank the reviewer for their time and effort in reviewing our work. We are happy to see the numerous positive comments reflective of our contributions, namely: that the novelty of our work is “very high” while introducing an "elegant design/method”, that our paper “addresses a very important problem of joint dynamical inference and structure learning” with “comprehensive experimental evaluations on both synthetic and real datasets” which validates the utility of our method, and that the “paper is clearly written and easy to follow”. Below, we provide detailed responses and answers to the reviewer’s questions.
>
> > scalability of StructureFlow (W1/Q1)
>
> StructureFlow relies on exactly the same training paradigm as flow matching and is therefore simulation free, i.e. the training procedure does not rely on numerical integration, and is as scalable as other flow matching methods.
> The novelty that we introduce is in (i) the parameterization of the flow model using an interpretable architecture from which a graph can be extracted and (ii) the added inductive bias of an time-independent dynamics.
>
> A practical demonstration of the scalability of our approach is in Figure 2 (Section 5.1) of our original submission. We show how StructureFlow scales as the number of variables increases. We compare against Reference Fitting (RF) and NGM NeuralODE (NGM-NODE) baselines, measuring training time in seconds varying the dimension up to $d = 500$.
>
> While RF incurs the lowest computational cost, it assumes **linear** dynamics. This is a serious limitation for the trajectory inference task, since it can only model Gaussian distributions (further results on this are shown in section 5.2). Furthermore, its performance on the structure learning task also degrades in higher dimensions.
> NGM-NODE requires backpropagation through ODE solves at training time, and is an order of magnitude slower than either StructureFlow or RF.
>
> To further illustrate scalability, we include a new table for the BoolODE TF system where we show StructureFlow training time, structure learning performance, and dynamical inference performance compared to the baselines (see response to [CZaQ] and Table 8 in E.5 of the appendix in the revised manuscript).
>
> Overall, our results demonstrate that StructureFlow exhibits favourable scaling both in terms of computational cost and ability to learn the underlying system structure, while remaining able to model non-linear dynamics.
>
> > evaluation of StructureFlow (W2/Q2)
>
> It is important to clarify a fundamental distinction in problem setting. StructureFlow is designed for dynamics and structure learning in the setting of **stochastic** dynamics where **trajectories/longitudinal information are unobserved**, and one has only access to a series of population snapshots.
>
> Methods such as those in NRI [1], which the reviewer mentions, and a similar stochastic method SFI [2] are designed for the setting where the trajectories of the underlying systems are **fully observed**. Furthermore, [1] assumes that dynamics are **deterministic**.
> Both of these methods are therefore inapplicable to the settings we consider, such as single cell time-series experiments, where the dynamics are intrinsically stochastic and longitudinal, and trajectory-wise observations are impossible (since the observation process in cells is destructive). Thus, [1, 2] are orthogonal to the setting we consider, where trajectories are not available, only population snapshots. Nevertheless, as the works are related, we have added them to section 2.1 to provide the reader with additional background.
>
> Existing methods in the literature that perform joint trajectory inference and structure learning are few. We compare already to a comprehensive set of baseline methods for each of the tasks – 9 competing methods for structure learning/network inference, and 4 methods for trajectory inference.

---

> > ### Author Response · Authors · 2025-11-24
> > **(2/2)**
> >
> > > details of how dynGENIES is applied and evaluated on the BoolODE data (Q3)
> >
> > BoolODE outputs are just single-cell snapshots + timestamps. dynGENIE3 is designed for bulk time series and does not natively work on per-cell single-cell snapshots. We therefore take the average values of gene-expression count concentrations across all cells at each time point So for each unique time $t$, we average across all cells at that time to create pseudo-bulk expressions. We then stack those per-timepoint means into a matrix of (num_times, num_genes) and give it to dynGENIE3 as one trajectory. We found an existing work that performs the same operation for dynGENIE3 which corroborates our approach as reasonable [3]. We have added these details to the revised manuscript in Appendix B.2.
> >
> > > details of how TIGON is applied and evaluated (Q4)
> >
> > Thank you for pointing this out! We too found that TIGON does not scale well, mostly because it involves integrating an ODE at each training step. TIGON uses either a PCA layer or an autoencoder to first encode the ambient data. This is a step for the “dimensionless-solver'' used within TIGON, which is optimized to operate in 10 dimensional space. All of the other baselines, and StructureFlow are trained directly in the ambient space. Thus, in order to provide a comparative setting, we train TIGON in the ambient space as well and forgo the PCA/autoencoder step. We did not encounter challenges when applying TIGON to the high dimensional real (Renge) biological system. However, because TIGON uses Gaussian mixture models to represent their density functions at the given discrete timepoints and [4] show that parameter estimation with high dimensionality can be challenging because of the large number of parameters that need to be estimated, we also train TIGON in PCA space, and then convert to the ambient space using the following formula: $J_X = W^T J_z W$ where $J_z$ is the latent Jacobian. We note no difference in performance for structure learning. We include further details in our updated appendix section B.2.
> >
> > > Minor comments (line 148, line 165)
> >
> > Thank you for pointing these out! For 1. We have updated the manuscript to remove the duplicate text. For 2. You are correct that $q(z)$ was never used, we have replaced it with $z := (x_0, x_1) \sim \pi(x_0, x_1)$.
> >
> > We’d like to once again thank the reviewer for their time and effort in reviewing our paper. We are happy to see your positive overview of our work. We believe that through this rebuttal, we have addressed the reviewers comments and questions. If the reviewer is satisfied with our response, we’d greatly appreciate it if they'd consider raising their score, reflective of the generally positive review of our paper. We are happy to answer any questions or additional comments that may arise.
> >
> > [1] Neural relational inference for interacting systems, Kipf et al.
> >
> > [2] Learning Force Fields from Stochastic Trajectories, Frishman and Roncera
> >
> > [3] Moscardó García, M., Aalto, A., Montanari, A.N. et al. Multi-omic network inference from time-series data. npj Syst Biol Appl 11, 114 (2025).
> >
> > [4] Ruan L, Yuan M, Zou H. Regularized parameter estimation in high-dimensional gaussian mixture models. Neural Comput. 2011 Jun;23(6):1605-22. doi: 10.1162/NECO_a_00128. Epub 2011 Mar 11. PMID: 21395439; PMCID: PMC5638044.

---

> > > ### Comment · Reviewer_HiCq · 2025-11-26
> > >
> > > 1. Can you provide further details how to make TIGON scalable by dropping the PCA layer?
> > > 2. Thanks for discussing the differences between StructureFlow and NGM+SF2M, somehow I feel that the novelty of StructureFlow is limited when compared to NGM+SF2M -- for NGM+SF2M, just adding a mask on the input to adopt interventions and an extra time dimension to make it time-dependent. What are the other novel contributions of StructureFlow?

---

> ### Author Response · Authors · 2025-11-27
> **(1/2)**
>
> > Can you provide further details how to make TIGON scalable by dropping the PCA layer?
>
> We thank the reviewer for their continued engagement. Below, we address the reviewers additional questions.
>
> 1. **Regarding details for making TIGON scalable.** We agree that TIGON exhibits unfavourable scaling in the input dimension, and we observe this as well through our experiments. We were able to run TIGON in both Ambient space (gene space, d = 103) and PC space (10 PCs) in our real biological setting, despite the increase in computational cost when running in ambient dimension, and notice no difference in structure recovery performance. We provide a table showing the relative run-times between TIGON and StructureFlow on the real biological dataset and include this in appendix E.8, Table 14 in our updated manuscript. We also refer the reviewer to Table 8 in our updated manuscript, and our response on scalability (response 3/5) to reviewer [CZaq] for further reference.
>
> In regards to running TIGON in ambient space: we simply drop the PCA step, and run directly on the high dimensional space (103 dims/genes in the real biological setting). As mentioned, this is significantly slower than StructureFlow, but the dataset is quite small (~4500 cells), so it does not require a significant amount of iterations to find a competitive solution.
>
> |Time [s]                         |Model        |Parameter Size         |
> |---                              |---          |---                    |
> |319.31+(4.0) $\pm$ 21.29+(0.08)  |StructureFlow|1440646                |
> |1542.88 $\pm$ 26.67              |TIGON (PCA) |       43140                  |
> |14112.49                                 |TIGON (Ambient) |  409448               |
> |338.12 $\pm$ 0.78                |RF           |       --                |
> |101.31 + (4.30) $\pm$ 1.43+(0.29)|$[SF]^2M$    |175310                 |
> |12.21 $\pm$ .125                 |OT-Velo      |     --                  |

---

> ### Author Response · Authors · 2025-11-27
> **(2/2)**
>
> > Thanks for discussing the differences between StructureFlow and NGM+SF2M, somehow I feel that the novelty of StructureFlow is limited when compared to NGM+SF2M -- for NGM+SF2M, just adding a mask on the input to adopt interventions and an extra time dimension to make it time-dependent. What are the other novel contributions of StructureFlow?
>
> **Regarding points on differences with NGM-$[SF]^2M$.** We would like to emphasize here, that **NGM-$[SF]^2M$ is one-of the worst performers on the structure learning task (by a large margin), as shown in Figure 3 section 5.2, and does not adequately recover the underlying network structure**. Through this, we establish that NGM-$[SF]^2M$ is not a practical solution to this problem. Below we highlight the several novel contributions of StructureFlow that enable state-of-the-art performance on the joint task:
>
> **Decomposition of Dynamics**: First, we will clarify, **we _do not_ make the model time-dependent by just adding an extra time dimension**. We argue that modeling time-dependence in this way is unprincipled, in that the NGM treats this extra dimension as a system variable, which is incorrect (we do not want to model causal dependencies between t and $\mathbf{x}$’s, only between $\mathbf{x}$’s). Rather, we decompose the flow into two components: an autonomous (time-independent) drift $\mathbf{v}(\mathbf{x})$, which is parameterized via the NGM and models structure, and a time-dependent score $\mathbf{s}(\mathbf{x}, t)$. We describe this in detail in Section 3.1 of our original submission. This aligns more closely with priors on underlying biology (where gene regulatory relationships are generally time-invariant) as well as the SDE model we use. Overall this leads to better performance, demonstrated through our empirical results.
>
> **Intervention model and evaluation on unseen knockouts**: Unlike NGM-$[SF]^2M$, StructureFlow is designed to both handle interventions, and predict conditional population dynamics. Modeling interventions is critical for disentangling correlation from causation, and allows us to incorporate information from interventional data. Moreover, because of this, we are able to effectively predict/generate conditional population dynamics (NGM-$[SF]^2M$ cannot). Moreover, we demonstrate that StructureFlow is able to predict response of _unseen_ interventions (Table 3 Section 5.3), a longstanding problem in perturbational biology.
>
> **Stochastic dynamics**: StructureFlow models stochastic dynamics (SDE), while the best performing method of NGM-$[SF]^2M$ does not; i.e. Tong et al, 2024, find that OT-CFM (ODE), using no score, performs best. Thus, NGM-$[SF]^2M$ does not adequately capture the stochastic nature of the biological dynamics. We additionally show the necessity of modeling stochasticity in Table 7 Appendix E.4 in our manuscript. Without stochasticity, the model's structure learning performance drops significantly, emphasizing the importance of StructureFlow.
>
> **Comprehensive suite of empirical experiments / benchmarks.** Beyond methodological contributions, our work also adds significantly to the evaluation pipeline of methods akin to StructureFlow and beyond. NGM-$[SF]^2M$ was evaluated in only 2 observational (no interventions) synthetic systems (trifurcating and bifurcating) and compared against only trivial correlation-based baselines. Further, the authors of NGM-$[SF]^2M$ considered a data setting of an unrealistic quantity of time-points (> 50 timepoints). This is unrealistic since in practice datasets of this nature (such as the real dataset considered in our work) are severely limited to the number of population snapshots across time (~ 4). Finally, we also provide an experiment to demonstrate StructureFlow’s ability to scale to high dimensional systems (not done by NGM-$[SF]^2M$).
>
> We hope these additional details help clarify the remaining questions raised by the reviewer. If the reviewer feels we have addressed their concerns, we kindly ask that they consider raising their score. We are more than happy to continue engaging in discussion and addressing any salient points/questions that may arise!

---

### Author Response · Authors · 2025-11-24
**Summary of Reviews and Our Response (1/2)**

We’d like to begin by thanking all the reviewers for their valuable time and effort in reviewing our paper, for their constructive feedback, and for their insightful questions, which together have helped improve the overall quality of our work.

In this work, we tackled the problem of **jointly learning** the underlying network structure of systems (_structure learning_) while simultaneously inferring the respective system dynamics (_trajectory/dynamical inference_), from partially observed, noisy, and unpaired empirical samples/snapshots (_population data_). **This is generally a very hard problem** as we do not observe entire trajectories (only distributions over time, samples are unpaired), we have limited timepoints (e.g. the real biological system considered here has only 4 timepoints), and systems tend to be high-dimensional (100s - 1000s of variables). **There are currently minimal-to-no adequate solutions to this problem**, motivating our work.

To address this problem, we (i) introduced **StructureFlow** a novel simulation-free method for addressing the joint problem from population snapshot data, and (ii) provided a comprehensive suite of experiments on synthetic systems, biologically meaningful simulated systems, and on a real biological data system to evaluate StructureFlow and akin methods. In the following, we provide a summary of the feedback and the rebuttal, overviewing the numerous positives highlighted by the reviewers and summarizing the central comments/questions raised and how we addressed them (which were primarily clarifying questions).

**Positive highlights.** We were delighted to see the many positive comments made by the reviewers:

- [HiCq] and [MmjV] pointed to the “importance” of the problem of joint structure learning and dynamical inference, which is addressed in our work and by our method (StructureFlow).

- [HiCq, CZaQ, 6hGC] all commented positively on the design choices of our method. For example, [HiCq] pointed out the “elegant design” of StructureFlow and that our method is of “very high novelty”, [CZaQ] states StructureFlow is “neat and practically convenient" while pointing to the “clear and modular parameterization, and [6hGC] states our “method assumptions and architecture are well delineated”. [CZaQ] and [6hGC] both pointed to the importance and practical value of modeling interventions for biological applications and the problem setting considered in our work.

- [HiCq, MmjV, 6hGC] all appreciated the breadth of our evaluations, which demonstrated StructureFlow’s scalability and performance on real and synthetic data experiments. [HiCq, MmjV] pointed to the “comprehensive” nature of our empirical experiments, while [6hGC] pointed to our method’s ability to reach “top tier performance” across systems.

- Lastly, all reviewers gave either positive comments, or positive evaluation (presentation = 3) regarding the writing and presentation of our paper.

**Clarifications and elaboration.** At the same time, reviewers provided valuable feedback through clarifying questions and comments on how we can improve our work. Below we outline how we addressed these comments through the rebuttal and revisions in the manuscript. (We have updated the manuscript where we included relevant revisions in blue text.)

- **Elaboration on novelties:** Reviewers [MmjV] and [CZaQ] asked for clarifications about the methodological advancements of StructureFlow in relation to NGM-SF2M. We clarify that **the advances in StructureFlow are fundamental to the improved performance** and our results demonstrate the shortcomings of NGM-SF2M, which fails to recover underlying structure (Figure 3 section 5.2). We’ve included detailed responses to the individual reviewers regarding this point and added a discussion in a new appendix section C (novelty statement).

- **Additional discussion of related works and future directions:** Reviewers asked about StructureFlow’s relation to some existing literature (relational inference [MmjV, 6hGC], multi-marginal Schrodinger bridges [CZaQ]). We have added discussion and references of this literature in sections 2.1 and 2.2. Regarding comments on extension and advancements of StructureFlow to amenable settings (e.g. unbalanced optimal transport [CZaQ], use of biological priors [CZaQ], considering imperfect interventions [6hGC], theoretical guarantees on identifiability [CZaQ], unevenly spaced temporal samples [MmjV], and time-varying networks [6hGC]), we provide detailed rebuttals in the responses to individual reviewers and added a discussion of these directions to our conclusion (future work).

- **Additional ablations:** Reviewers [6hGC], [CZaQ], and [MmjV] suggested additional informative ablations. We conducted all of the suggested ablations and added the results to the updated manuscript in Appendix E Tables 8-11 (as well as within the responses below).

---

> ### Author Response · Authors · 2025-11-24
> **(2/2)**
>
> **Concluding remarks on feedback summary:**
>
> We view the numerous positive comments, alongside the questions/comments pertaining to advancements and extensions of StructureFlow, as an overall positive evaluation of our work. While the positive feedback demonstrates the uniqueness and importance of our paper and method, comments regarding extensions of StructureFlow show that our work is easily amenable to future research and advancement. This signifies the value our work brings to the community, as many of these suggested directions are oftentimes sufficient to hold as their own bodies of work.
>
> Through this rebuttal, we hope we have answered all of the reviewers’ questions and addressed their concerns. In light of the numerous positive comments and identification of the various future directions our work can be taken, we kindly ask the reviewers’ to consider raising their scores to reflect, what we perceive, as a positive evaluation.

---

### Author Response · Authors · 2025-12-01
**Summary of Discussion (For AC, SAC, and PCs)**

Dear AC, SAC, and PCs,

Thank you for your time and effort in evaluating our work and navigating these difficult circumstances. We believe that this process has improved the overall quality of our work, strengthened our contributions, and further clarified the novelty of our work. We strongly believe that the original scores did not accurately reflect the evaluation of our work, which was predominantly positive (across all reviewers). Through this rebuttal and discussion, we further enhanced this positive evaluation.

Below, we provide a summary of the discussion phase, reviewer responses, and how we addressed all of the reviewers' concerns during this phase. We refer you to the Feedback Summary (“Summary of Reviews and Our Response”) for a detailed overview of the original reviews and rebuttal. We provided an updated manuscript (revisions in blue text) incorporating all suggested changes, which were predominantly additions to the appendix, with one new experiment (unbalanced setting) added to the main text. The remaining changes in the main text are all additional clarifications.

**Summary of discussion, reviewer follow ups, and score raises**.

**Reviewer [HiCq]**: HiCq asked two clarifying follow up questions. We addressed both questions in detail during the discussion phase by providing an additional experiment (also included in the revised manuscript) and further clarified the novelty (and _need_) of our method. Any further comments regarding novelty would have been addressed by our follow up responses to CZaQ, where we provided additional functionality and theoretical backing in the updated manuscript, but unfortunately we did not get the opportunity to continue the discussion with the reviewer.

Since we addressed all of the concerns raised by HiCq, we are optimistic that the reviewer would have raised their score.

**Reviewer [MmjV]**: MmjV acknowledged our “detailed response”, raised their score, and explicitly stated that they were “willing to raise their score after a couple further clarifications”, all prior to our follow up response. We addressed the reviewer’s follow-up clarifying questions by adding two additional experiments (in response and in the manuscript), and clarified two sentences in the revised manuscript.

We addressed all of the reviewer’s original concerns, as well as follow up questions, hence we are optimistic that MmjV would have further raised their score.

**Reviewer [CZaQ]**: CZaQ acknowledged our “detailed response”, raised their score, stated that our “manuscript has clearly improved”, and stated that our “work has practical value”. In parallel, the reviewer asked clarifying follow up questions, suggested we incorporate some of the potential future directions cited in our response and their rebuttal, and inquired about additional visualizations. We addressed all of these comments in our follow up response. Namely, we added an extension to our method per the reviewer’s suggestions (Unbalanced StructureFlow) and demonstrated its effectiveness empirically (we added the experiment to updated manuscript), we added principled theoretical justification for our method to the manuscript, we added all requested additional visualizations/figures into the updated manuscript, and we clarified the reviewer’s question regarding the implementation of a baseline.

Prior to our follow up response, the reviewer gave an initial score raise. We believe that our follow up response addressed all of the reviewer’s additional concerns, while further increasing the novelty of our work and strengthening our contributions. With this, we strongly believe CZaQ would have further raised their score.

**Reviewer [6hGC]**: 6hGC acknowledged our rebuttal and stated that the “concerns from their review have been addressed”. The reviewer also stated that they would monitor the responses from other reviewers in the discussion phase for specific points. We addressed all of these specific points either in the original rebuttal (to other reviewers) or in follow up questions during the discussion phase.

We thank you for consideration for ICLR 2026! We strongly believe we have addressed all of the reviewers' concerns throughout the rebuttal and discussion period and feel that reviewers were in the process of raising their scores, reflective of the additions we made to the manuscript and their positive evaluations. We hope that you will take into account all of these factors in your evaluation and that you will strongly consider our work for acceptance at ICLR 2026.

---

### Meta-Review · Area_Chair_9aXW · 2026-01-05

**Summary:**

The authors introduce a simulation-free approach to jointly perform structure learning and inference in dynamical systems. The method uses a neural graphical model/vector field to apture the structure and a time-dependent score function for the stochastic dynamics. The approach is inspired by the multi-marginal SB, but after discussion does not directly optimise a multi-marginal SB. Nevertheless, the method has been described as clearly presented and interesting by several reviewers with an extensive evaluation.

Reviewers have initially been rather split about this submission, with positive assessments [6hGC] and negative assessments [MmjV, CZaQ]. Negative reviews dominated but the more critical reviewers seem to have been partially satisfied with the responses during the rebuttal. However, due to the low initial scores only a mild improvement is to be expected.

**Reviewer Concerns:**

One of the remaining concerns was with regard to the scalability of the method [HiCo, CZaQ], which has been partially addressed by the authors. Further remaining concerns are about the limited novelty of the method [CZaQ].

**Reviewer Scores:**

Potential change of the scores is as follows:

- [HiCq]: Likely to have increased. => 5/6
- [MmjV]: Likely to have increased. => 4/5
- [CZaQ]: Likely to have increased. => 4/5
- [6hGC]: Likely kept or reduced score. => 7/8

---

### Decision · Program_Chairs · 2026-01-26

Reject